# Genetic instability from a single S phase after whole-genome duplication

Simon Gemble[1✉], René Wardenaar[2,8], Kristina Keuper[3,8], Nishit Srivastava[4], Maddalena Nano[1,7], Anne-Sophie Macé[5], Andréa E. Tijhuis[2], Sara Vanessa Bernhard[3], Diana C. J. Spierings[2], Anthony Simon[1], Oumou Goundiam[1], Helfrid Hochegger[6], Matthieu Piel[4], Floris Foijer[2], Zuzana Storchová[3] & Renata Basto[1✉]

Diploid and stable karyotypes are associated with health and fitness in animals. By contrast, whole-genome duplications—doublings of the entire complement of chromosomes—are linked to genetic instability and frequently found in human cancers[1–3]. It has been established that whole-genome duplications fuel chromosome instability through abnormal mitosis[4–8]; however, the immediate consequences of tetraploidy in the first interphase are not known. This is a key question because single whole-genome duplication events such as cytokinesis failure can promote tumorigenesis[9] and DNA double-strand breaks[10]. Here we find that human cells undergo high rates of DNA damage during DNA replication in the first S phase following induction of tetraploidy. Using DNA combing and single-cell sequencing, we show that DNA replication dynamics is perturbed, generating under- and over-replicated regions. Mechanistically, we find that these defects result from a shortage of proteins during the G1/S transition, which impairs the fidelity of DNA replication. This work shows that within a single interphase, unscheduled tetraploid cells can acquire highly abnormal karyotypes. These findings provide an explanation for the genetic instability landscape that favours tumorigenesis after tetraploidization.

As whole-genome duplications (WGDs) can have different origins[11,12], we developed several approaches to induce tetraploidization through either mitotic slippage, cytokinesis failure or endoreplication in the diploid and genetically stable RPE-1 human cell line. Most cells resulting from cytokinesis failure contained two nuclei, whereas endoreplication or mitotic slippage generated mononucleated tetraploid cells. Cell size, cell number, nucleus size and centrosome number were considered to distinguish diploid cells from tetraploid cells (Fig. 1a, b, Extended Data Fig. 1a–i). For each approach, a mix of diploid and tetraploid cells was obtained, enabling the comparison of internal diploid controls and tetraploids. In all conditions, most tetraploid cells continued to cycle throughout the first interphase, allowing us to probe the consequences of tetraploidy within the first cell cycle.

Using γH2AX, an early marker of DNA damage, we found high levels of DNA damage in tetraploid cells (but not in controls) independently of how they were generated (Fig. 1c–h, Extended Data Figs. 1a–i, 2a–f, Methods). Moreover, whereas more than 10 γH2AX foci were present in only 5–9% of diploid cells, this proportion reaches 34–54% in tetraploid cells (Fig. 1c–h). The number of γH2AX foci correlated with fluorescence intensity (Extended Data Fig. 1j). We excluded the possibility that the increase in tetraploid cells was simply owing to increased nuclear size by normalizing the number of γH2AX foci to the nuclear area or nuclear

fluorescence intensity (Extended Data Fig. 1k–l). High levels of DNA damage were also found in tetraploid BJ fibroblast and HCT116 cells upon WGD (Extended Data Fig. 2g, h).

To evaluate levels of DNA damage after WGD, we compared DNA damage between tetraploid and diploid cells with replication stress. Replication stress results from the slowing or stalling of replication forks, which can be induced by high doses of aphidicolin (APH; a DNA polymerase inhibitor) or hydroxyurea[13,14] (a ribonucleotide reductase inhibitor). APH or hydroxyurea generated similar levels of DNA damage in diploid cells, when compared with untreated tetraploid cells (Extended Data Fig. 2i). In addition to γH2AX, we also observed a significant increase in the number of foci containing the double strand break repair factors FANCD2 and 53PB1[15] in the first interphase following WGD (Fig. 1i–l). Further, tetraploid cells showed an increased olive tail moment in alkaline comet assays, indicating single and double strand breaks (Extended Data Fig. 2j, k).

We next tested whether DNA damage is also generated in the subsequent cell cycles. A high proportion of tetraploid RPE-1 cells arrests after the first cell cycle in a LATS2–p53-dependent manner[16]. We thus analysed DNA damage levels in p53-depleted cells (Extended Data Fig. 2l). During the second and third interphases following tetraploidization, we observed a considerable decrease in DNA damage levels

[1]Institut Curie, PSL Research University, CNRS, UMR144, Biology of Centrosomes and Genetic Instability Laboratory, Paris, France. [2]European Research Institute for the Biology of Ageing, University of Groningen, University Medical Center Groningen, Groningen, The Netherlands. [3]Department of Molecular Genetics, TU Kaiserslautern, Kaiserslautern, Germany. [4]Institut Curie and Institut Pierre Gilles de Gennes, PSL Research University, CNRS, UMR 144, Systems Biology of Cell Polarity and Cell Division, Paris, France. [5]Cell and Tissue Imaging Facility (PICT-IBiSA), Institut Curie, PSL Research University, Centre National de la Recherche Scientifique, Paris, France. [6]Genome Damage and Stability Centre, School of Life Sciences, University of Sussex, Brighton, UK. [7]Present address: Molecular, Cellular, and Developmental Biology Department, University of California, Santa Barbara, CA, USA. [8]These authors contributed equally: René Wardenaar, Kristina Keuper. ✉e-mail: simon.gemble@curie.fr; renata.basto@curie.fr

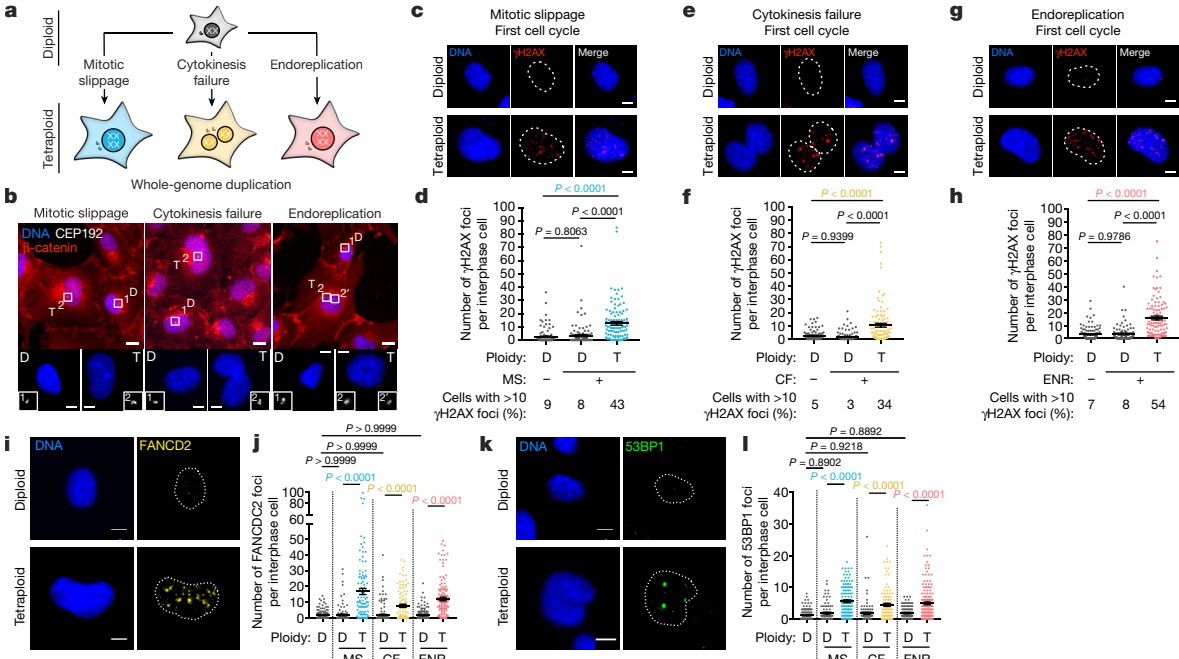

**Fig. 1 | High levels of DNA damage in the first interphase following unscheduled WGD. a**, Schematic of the methods used to generate tetraploid cells. **b**, Top, images of diploid (D) and tetraploid (T) RPE-1 cells generated by mitotic slippage, cytokinesis failure or endoreplication. Centrosomes labelled with anti-CEP192 and cell membranes labelled with anti-β-catenin. Bottom, outlined regions are shown at higher magnification. **c, e, g**, Images showing DNA damage caused by mitotic slippage (**c**), cytokinesis failure (**e**) or endoreplication (**g**) revealed by anti-γH2AX in diploid and tetraploid RPE-1 cells as indicated. **d, f, h**, The number of γH2AX foci following mitotic slippage (**d**), cytokinesis failure (**f**) or endoreplication (**h**) per interphase cell in diploid and

tetraploid RPE-1 cells. Data are mean ± s.e.m.; >100 interphase cells, 3 independent experiments. The percentage of interphase cells with at least ten γH2AX foci for each condition is indicated under the graph. **i, k**, Images of diploid and tetraploid RPE-1 cells generated by mitotic slippage labelled with anti-FANCD2 (**i**) or anti-53BP1 (**k**) antibodies. **j, l**, The number of FANCD2 (**j**) or 53BP1 (**l**) foci per interphase cell in diploid and tetraploid RPE-1 cells. Data are mean ± s.e.m.; >100 interphase cells, 3 independent experiments. Dotted lines indicate the nuclear region. CF, cytokinesis failure; ENR, endoreplication; MS, mitotic slippage. **d, f, h, j, l**, One-sided analysis of variance (ANOVA) test. Scale bars, 10 µm.

(Extended Data Fig. 2m–o). As most animal cells are normally organized in tissues with cell–cell adhesions, we tested the consequences of WGD in 3D cultures. Spheroids containing tetraploid cells displayed a higher γH2AX index (Methods) compared with diploid cells (Extended Data Fig. 3a–d).

Collectively, our results show that a transition from a diploid to a tetraploid status after unscheduled WGD is accompanied by high levels of DNA damage within the first cell cycle.

## DNA replication-dependent DNA damage

We determined the cell cycle stage when the DNA damage occurs using the fluorescence ubiquitination cell cycle indicator (FUCCI). During G1, the number of γH2AX foci was quite low and similar to that found in controls. As tetraploid cells entered S phase, we observed a slight increase in the number of foci, which increased substantially at the end of S phase (Fig. 2a, b, Extended Data Fig. 3e, f). These results were further confirmed by time-lapse imaging using H2B–GFP to visualize DNA and 53BP1–RFP (Extended Data Fig. 3g, h, Supplementary Videos 1, 2). To confirm that DNA damage in tetraploid cells appeared during S phase, we blocked cells at the G1/S transition using high doses of inhibitors of CDK4/6 or CDK2 for 16 h (Extended Data Fig. 3i, j). We chose a 16-h period because this corresponds to the end of S phase in the cycling population (Fig. 2a, b) and thus enables us to distinguish whether DNA damage accumulates in a specific cell cycle phase or, alternatively, after a certain period of time. G1-arrested tetraploid cells showed low levels of DNA damage, whereas cells released in S phase exhibited high levels of DNA damage (Extended Data Fig. 3i–o). Of note, we observed a significant increase in the percentage of γH2AX foci co-localizing with markers of active DNA replication sites visualized by proliferating

cell nuclear antigen (PCNA) and EdU incorporation in tetraploid cells compared with diploid cells (31% versus 7%) (Extended Data Fig. 3p, q).

By evaluating markers of DNA damage signalling and repair pathways we observed that the number of foci containing KU80 and XRCC1—proteins involved in non-homologous end joining[17]—remained low in tetraploid cells. By contrast, the number of foci containing the homologous recombination (HR) factor RAD51 was increased. Moreover, the percentage of RAD51 foci co-localizing with γH2AX foci was significantly increased in tetraploid cells compared with diploid cells (14% versus 3%). Foci containing the replication stress markers replication protein A (RPA) and FANCD2 were also increased in number, and we observed a significant increase in their colocalization with γH2AX foci in tetraploid cells compared with diploid cells (40% versus 14%) (Extended Data Fig. 4a–k). Together, these results demonstrate that tetraploid cells experience high levels of DNA damage during S phase, indicated by markers of DNA damage and HR.

We hypothesized that DNA damage in tetraploid cells arises from errors occurring during DNA replication. To test this possibility, cells were arrested in G1 (Extended Data Fig. 3k). We then released them in the presence of very low doses of APH or PHA-767491 (PHA; a Cdc7 inhibitor) to inhibit DNA replication (detected by absence of EdU) without generating DNA damage (Methods). This leads to inhibition of DNA replication while maintaining the biochemical activity typical of the S phase nucleus. DNA damage levels were markedly decreased in tetraploid cells treated with APH or PHA (Fig. 2c, d, Extended Data Fig. 5a–f). Of note, in the few tetraploid cells that escaped DNA replication inhibition (revealed by high EdU incorporation) there was still a large number of γH2AX foci (Extended Data Fig. 5g, h). Together, these results establish that WGD generates DNA replication-dependent DNA damage. Deoxyribonucleoside triphosphate (dNTP) exhaustion leads to

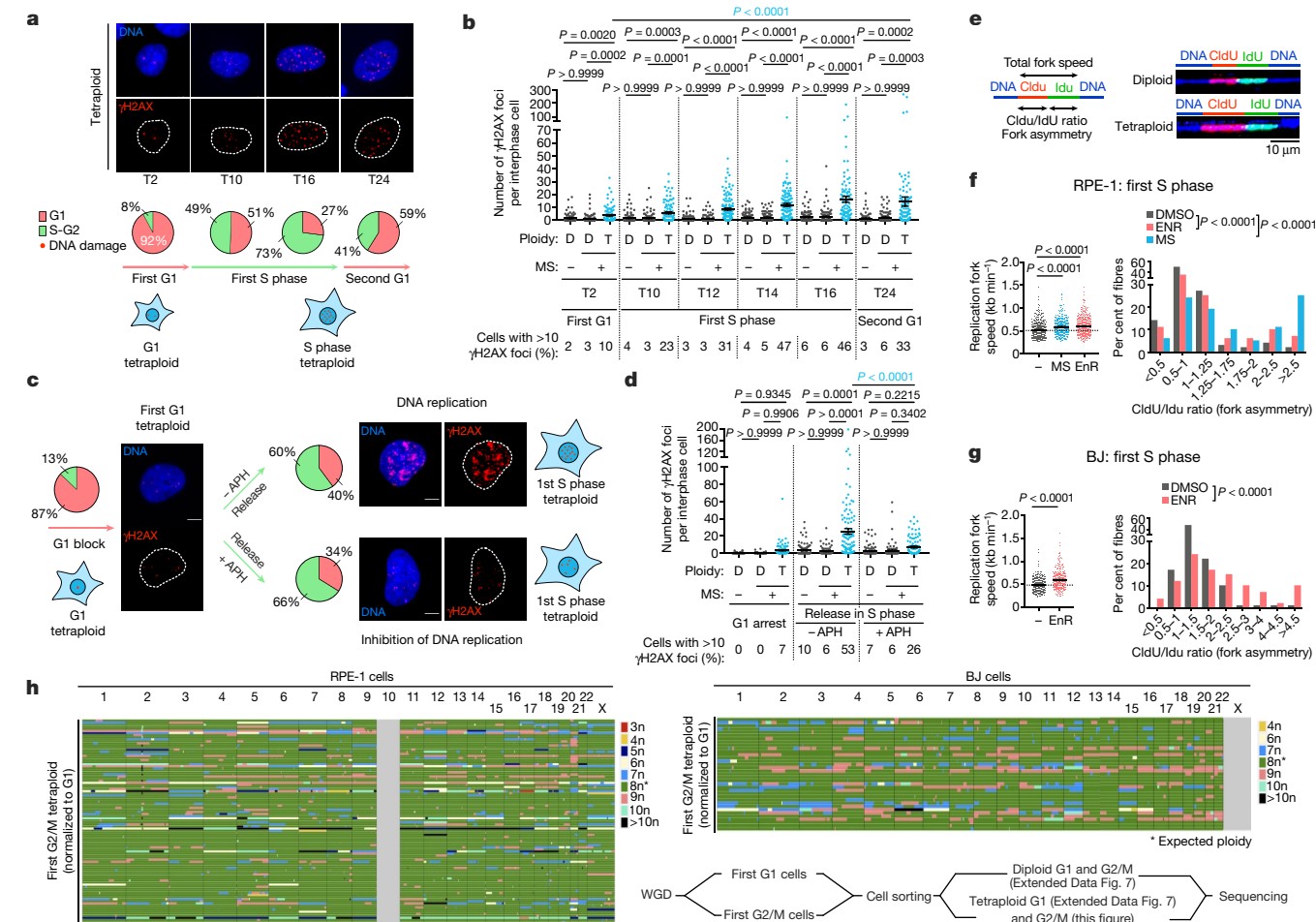

**Fig. 2 | Genetic instability in tetraploid cells is generated during S phase in a DNA replication-dependent manner. a**, Top, DNA damage visualized using γH2AX in RPE-1 tetraploid cells. Bottom, percentage of RPE-1 tetraploid cells in G1 or in S–G2. Data are mean >100 interphase cells, 3 independent experiments. **b**, The number of γH2AX foci per interphase cell in diploid (D) and tetraploid (T) RPE-1 cells. Data are mean ± s.e.m.; >100 interphase cells, 3 independent experiments. **c**, Percentage of RPE-1 tetraploid cells in G1 or in S–G2 and representative images showing DNA damage (anti-γH2AX) in tetraploid cells synchronized in G1 using 1 μM palbociclib or released in S phase with or without 400 nM APH. Data are mean ± s.e.m.; >100 interphase cells, 3 independent experiments. **d**, The number of γH2AX foci per interphase cell in diploid and tetraploid RPE-1 cells from **c**. Data are mean ± s.e.m.; >100 interphase cells, 3 independent experiments. **e**, Left, scheme for replication fork analysis. Right, immunofluorescence of DNA fibres obtained from diploid and tetraploid RPE-1 cells. **f**, **g**, Left, the replication fork speed in diploid and tetraploid RPE-1 (**f**) or BJ cells (**g**). Right, the CldU/IdU ratio in diploid and tetraploid RPE-1 (**f**) or BJ cells (**g**). Data are mean ± s.e.m.; >330 replication forks (**f**), >295 replication forks (**g**). **h**, Genome-wide copy number plots for G2/M tetraploid RPE-1 or BJ cells induced by mitotic slippage. Each row represents a cell. Bottom right, workflow showing the method used to sort the cells. **b**, **d**, **f**, One-sided ANOVA test. **g**, Two-sided *t*-test. Scale bars, 10 μm.

replication stress and genetic instability[18]. We tested whether supplying nucleosides rescued the DNA damage defects described above. This was however not the case in cells or in an in vivo model of polyploidy generation (Extended Data Figs. 5i, j, 10g). These results suggest that unscheduled WGD does not induce exhaustion of nucleoside levels as described in other oncogenic conditions[18].

We characterized DNA replication using RPE-1 cell lines stably expressing PCNA chromobodies (Supplementary Information, Methods). Quantitative 4D live imaging of DNA replication in diploid and tetraploid cells revealed marked decreases in the total number of PCNA foci and their volume and a similar effect on the number of EdU foci (Extended Data Fig. 6a–f). This suggests a lack of scaling up with DNA content and fewer active replication sites in tetraploid cells. Time-lapse analysis of PCNA and fluorescence intensity was used as a readout of early and late S phase[19], revealing a longer early S phase period in tetraploid cells (Supplementary Information, Extended Data Fig. 6g–i, Supplementary Videos 3, 4). We next performed DNA combing, which enables visualization of replication fork behaviour in single DNA fibres[14,20]. Median fork speed and fork asymmetry (a readout of stalled or collapse forks) were

increased in tetraploid cells (Fig. 2e–g, Extended Data Fig. 6j, k). We attempted to analyse inter-origin distance (IOD), as the number of active regions can influence fork speed[21]. We noted a trend for increased IOD in tetraploid cells; however, it did not reach the threshold for significance (a possible explanation is provided in Methods).

To assess the type of karyotype generated in a single S phase after WGD, we used single-cell DNA sequencing (Methods, Supplementary Information, Supplementary Methods). We identified over-duplicated chromosomes (more than 10) in addition to frequent over- and under-replicated regions (9n, 7n and 4n) in G2/M tetraploid cells (Fig. 2h, Extended Data Fig. 7a, b). Both aneuploidy and heterogeneity scores and the proportion of the genome affected by aneuploidies were increased in G2/M tetraploid cells (Fig. 2h, Extended Data Fig. 7a–d, Methods). Our data establish that WGD generates abnormal karyotypes within a single S phase.

## Non-optimal S phase in tetraploid cells

Tetraploid cells would be expected to 'scale up' RNA and protein content by a factor of two. However, we found no evidence of such an increase

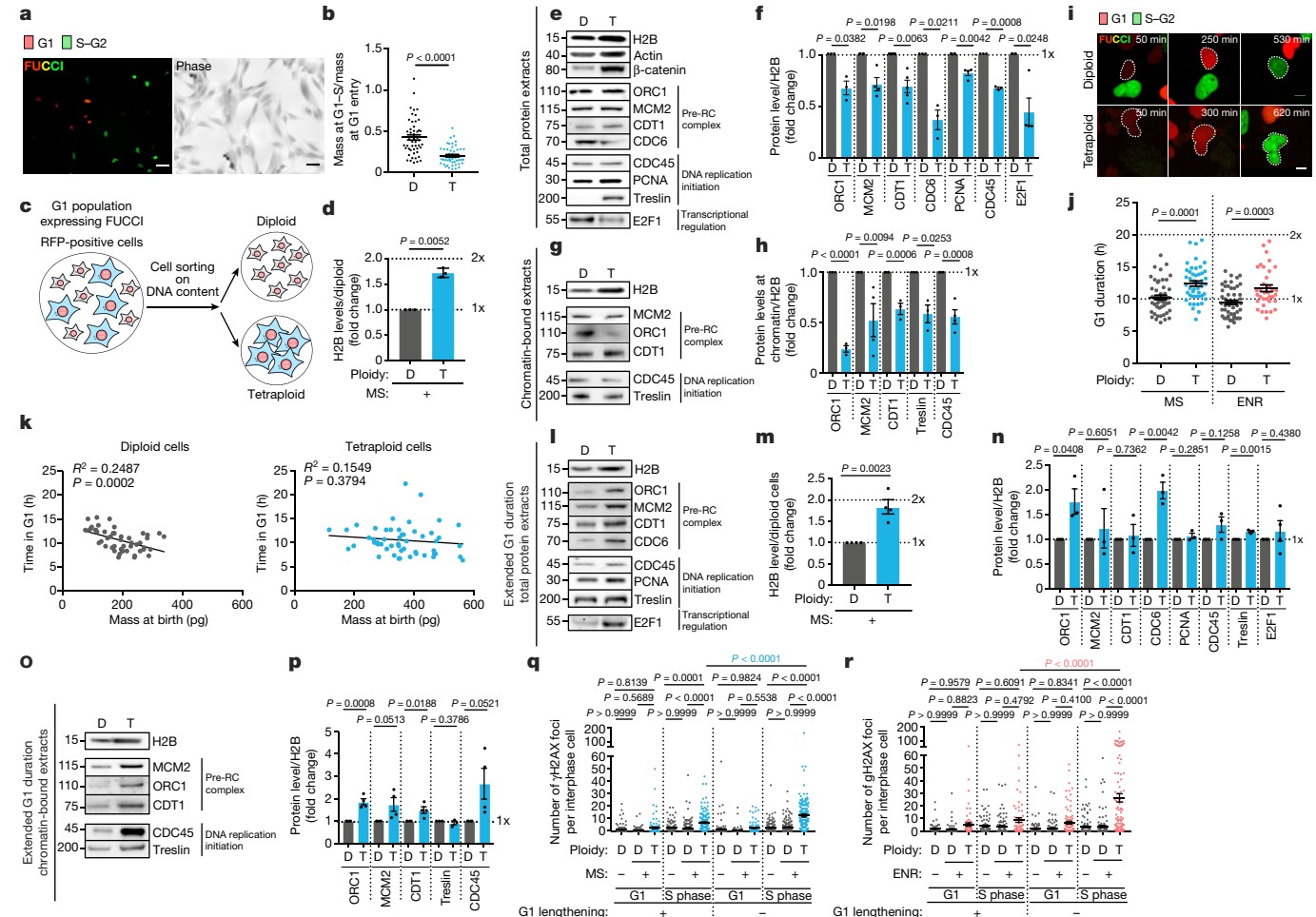

**Fig. 3 | Key replication factors do not scale up in tetraploid cells.**
**a**, Tetraploid cells expressing FUCCI and the corresponding image under phase microscopy. **b**, The ratio of protein produced during G1 in diploid (D) and tetraploid (T) cells. Data are mean ± s.e.m.; >50 G1 cells, 2 experiments. **c**, Schematic of fluorescence-activated cell sorting. **d**, Relative H2B levels in RPE-1 cells. Data are mean ± s.e.m.; three experiments. **e**, **g**, Western blots of total protein extracts (**e**) or chromatin-bound extracts (**g**) obtained from RPE-1 cells. **f**, **h**, The protein levels from total protein extracts in **e** (**f**) and chromatin-bound extracts in **g** (**h**). Data are mean ± s.e.m.; three independent experiments. **i**, Stills from time-lapse videos of RPE-1 cells expressing FUCCI. **j**, Graph showing the duration of G1 in RPE-1 cells. Data are mean ± s.e.m.; >35 interphase cells, 2 independent experiments. **k**, Graphs showing the time in G1 and the mass at

birth of RPE-1 cells. More than 50 interphase cells, 2 independent experiments. **l**, **o**, Western blots of (**l**) or chromatin-bound extracts (**o**) obtained from RPE-1 cells with extended G1 duration. **m**, Relative H2B levels in RPE-1 cells with extended G1 duration. Data are mean ± s.e.m.; four experiments. **n**, **p**, Protein concentration in total protein extracts from **l** (**n**) and chromatin-bound extracts from **o** (**p**). Data are mean ± s.e.m.; three experiments. **q**, **r**, The number of γH2AX foci in RPE-1 cells with G1 lengthening or G1 arrest using 160 nM or 1 μM palbociclib and released in S phase. Tetraploidy induced by mitotic slippage (**q**) or endoreplication (**r**). Data are mean ± s.e.m.; >100 interphase cells, 3 independent experiments. **e**, **g**, **l**, **o**, The same number of cells was loaded for each condition. **j**, **q**, **r**, One-sided ANOVA test. **d**, **f**, **h**, **m**–**o**, Two-sided *t*-test. **k**, Two-sided Pearson test. Scale bars, 50 μm (**a**), 10 μm (**i**).

in total RNA and protein content in newly born tetraploid cells using pyronin Y staining and quantitative phase imaging (Fig. 3a, b, Extended Data Fig. 8a–c). We next tested the levels of key DNA replication factors. We developed protocols to sort tetraploids from diploids on the basis of FUCCI and DNA content from a common cell population (Fig. 3c, Extended Data Fig. 8d, e, Methods). The same number of cells was loaded for diploid and tetraploid conditions and total protein extracts and chromatin-bound extracts were probed by western blot. The chromatin-associated H2B variant, the cytoskeleton component actin and the membrane component β-catenin showed increases consistent with tetraploidization. By contrast, using H2B as a read-out of DNA content, there was no similar increase in G1 and S phase DNA replication factors in tetraploid cells (Fig. 3d–f, Extended Data Fig. 8f, g). We analysed the origin recognition complex 1[22] (ORC1), the minichromosome maintenance 2 helicase[23] (MCM2), Cdc10-dependent transcript 1 protein[24] (CDT1) and CDC6[25]. These proteins are key members of pre-replication complexes and are normally loaded in G1 during

origin licensing. We also tested PCNA, CDC45[26] and treslin[27], which are required for the initiation of DNA replication. We further probed the levels of E2F1, a transcription factor that activates the expression of S phase genes[28–30]. With the exception of treslin, the total levels of these proteins did not show the expected increase in tetraploid cells (Fig. 3e, f). Furthermore, levels of pre-replication complexes, treslin and CDC45 also did not increase in the chromatin-bound fractions from tetraploid cells (Fig. 3g, h, Extended Data Fig. 8h).

In normal proliferative cell cycles, growth occurring during G1 phase prepares cells for DNA replication, increasing the expression and accumulation of key S phase regulators[29,31]. We measured G1 duration in tetraploid cells and found only a slight increase compared with diploid cells (Fig. 3i, j, Extended Data Fig. 8i, j). Further, although there was a significant correlation between cell mass and G1 duration in diploid cells, as described previously[32], this was not the case in tetraploid cells (Fig. 3k). We then tested whether G1 lengthening favoured error-free DNA replication in tetraploid cells. We delayed S phase entry using very

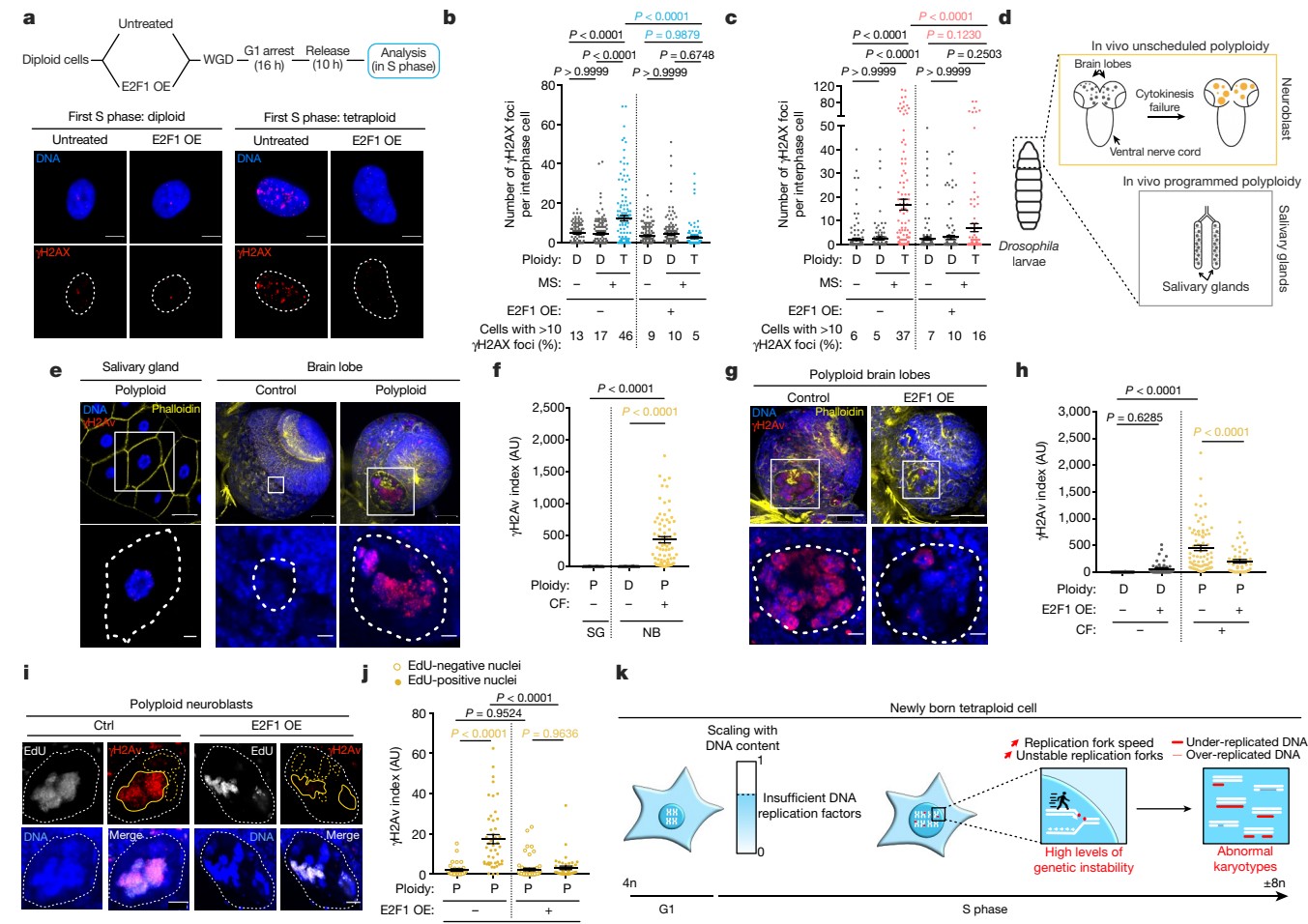

**Fig. 4 | Increased E2F1 levels are sufficient to rescue genetic instability in both tetraploid cells and in unscheduled polyploid cells in vivo. a**, Top, workflow showing the method used to overexpress E2F1 (E2F1 OE). Bottom, γH2AX immunofluorescence in cells overexpressing E2F1. **b**, **c**, Graphs showing the number of γH2AX foci per interphase cell in diploid (D) and tetraploid (T) RPE-1 cells released in S phase with and without E2F1 overexpression. Tetraploidy induced by mitotic slippage (**b**) or endoreplication (**c**). Data are mean ± s.e.m.; >100 interphase cells, three experiments. **d**, Experimental scheme to show the brain and the salivary glands of *Drosophila* larva. **e**, Representative images of salivary glands from wild-type larvae and brain lobes of control or *sqh*-mutant larvae. **f**, γH2Av index in salivary glands (SG) and in diploid (D) and polyploid (P) neural stem cells from the *Drosophila* larvae brain. NB, neuroblast. Data are mean ± s.e.m.; >60 interphase cells,

3 experiments. **g**, γH2Av in brain lobes of control or *sqh*-mutant larvae with or without E2F1 overexpression. **h**, γH2Av index in neuroblasts with or without E2F1 overexpression. Data are mean ± s.e.m.; >30 interphase cells, 3 experiments. **i**, γH2Av in neuroblasts derived from *sqh*-mutant larvae with or without E2F1 overexpression. The yellow dotted lines indicate EdU-negative nuclei, the solied yellow line indicates EdU-positive nuclei. **j**, γH2Av index in EdU-negative and EdU-positive nuclei with or without E2F1 overexpression. Data are mean ± s.e.m.; >30 interphase cells, 3 experiments. **k**, Model in which a single S phase generates genetic instability in tetraploid cells. The white dotted lines indicate the nuclear (**a**) or cell area (**e**, **g**, **i**). **b**, **c**, **f**, **h**, **j**, One-sided ANOVA test. Scale bars, 10 μm (**a**, **e** bottom right, **g** bottom), 20 μm (**e** bottom middle, **i**), 50 μm (**e** top, **e** bottom left, **g** top).

low doses of inhibitors of CDK4/6 or CDK2 (Extended Data Fig. 9a–c, Supplementary Information, Methods). In this condition, the levels of DNA replication factors from total cell or chromatin extracts scaled up with DNA content (comparing Fig. 3l–p with Fig. 3e–h and Extended Data Fig. 9k). Further, the number and volume of active replication sites in S phase scaled up with DNA content in tetraploid cells and the dynamic behaviour of PCNA in tetraploid cells was similar to that in diploid cells (Extended Data Fig. 9d–h, Supplementary Videos 5, 6). The time spent in S phase was not altered, but the ratio between early and late S phase in tetraploid cells was restored (Extended Data Fig. 9i, j). In all cell lines, G1 lengthening was sufficient to reduce the number of γH2AX, FANCD2 and 53BP1 foci in tetraploid S phase cells (Fig. 3q–r, Extended Data Fig. 9l–r).

Our data show that tetraploid cells transition from G1 to S phase prematurely without undergoing scaling of global protein mass. They enter S phase with insufficient DNA replication factors, which can be compensated for by G1 lengthening.

## E2F1 rescues genetic instability in tetraploid cells

As the time spent in G1 does not prepare tetraploid cells for S phase, we reasoned that increased E2F1 levels might compensate for defects in G1 length scaling up. E2F1 is a transcription factor that promotes proliferation and cell cycle progression by regulating S phase and DNA replication factors[29,30]. We over-expressed E2F1 in diploid cells, enabling us to increase the expression of DNA replication proteins just before generating tetraploid cells. This was sufficient to rescue the levels of DNA damage in tetraploid cells (Fig. 4a–c, Extended Data Fig. 10a–c).

A key prediction of our findings is that unscheduled polyploid *Drosophila* interphase neuroblasts[33] should also accumulate high levels of DNA damage in vivo. Indeed, the γH2Av index (Methods) was higher in polyploid neuroblasts compared with diploid neuroblasts or programmed polyploid salivary gland cells, which normally accumulate very high ploidies[34] (Fig. 4d–f). We tested the effect of E2F1OE in

polyploid neuroblasts and found that this was sufficient to decrease substantially DNA damage levels in vivo. Further, DNA damage was mainly restricted to EdU⁺ nuclei (Fig. 4g–j, Extended Data Fig. 10d–f). Together, these data show that in vivo unscheduled polyploidy is a source of DNA damage and genetic instability in replicating cells, which can be inhibited by increased E2F1 levels.

As WGDs are quite frequent in human tumours, which have high levels of genetic instability[1,2,35], our findings predict that these tumours must cope with increased DNA damage levels and therefore upregulate the DNA damage response pathway. We performed gene set enrichment analysis (GSEA) using cohorts of tetraploid and diploid lung, bladder and ovarian tumours[36]. This revealed an enrichment for DNA repair pathways in all tetraploid tumours when compared with diploid tumours (Extended Data Fig. 10h). These results suggest an increased requirement for the DNA damage response in tumours with WGD.

## Discussion

Here we analysed the initial defects following WGD and identified a very early window of high genetic instability that could promote acquisitions of multiple mutations, making it possible to bypass cell cycle controls while promoting survival of tetraploid cells. Our results are consistent with a model in which tetraploid cells transit through the first cell cycle while lacking the capacity to support faithful replication of increased DNA content (Fig. 4k, Supplementary Discussion).

In non-physiological conditions, such as those studied here, newly born tetraploids might not sense the increase in DNA content and may therefore be unable to adapt G1 duration or protein content to replicate a 4n genome. Further research is needed to identify the molecular mechanisms that promote ploidy increase while maintaining genetic stability and cell homeostasis to understand how tetraploid cancers and tetraploids arising during evolution adapted to the new cellular state.

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

## Methods

### Cell culture

Cells were maintained at 37 °C in a 5% $CO_2$ atmosphere. hTERT RPE-1 cells (ATCC cat. no. CRL-4000, RRID:CVCL 4388) and HEK 293 cells (ATCC cat. no. CRL-1573, RRID:CVCL 0045) were grown in Dulbecco's modified medium (DMEM) F12 (11320-033 from Gibco) containing 10% fetal bovine serum (GE Healthcare), 100 U ml$^{-1}$ penicillin, 100 U ml$^{-1}$ streptomycin (15140-122 from Gibco). BJ cells (ATCC cat. no. CRL-4001, RRID:CVCL 6573) and HCT116 cells (ATCC cat. no. CCL-247, RRID:CVCL 0291) were grown in Dulbecco's modified medium + GlutaMAX (61965-026 from Gibco) containing 10% fetal bovine serum (GE Healthcare), 100 U ml$^{-1}$ penicillin, 100 U ml$^{-1}$ streptomycin (15140-122 from Gibco).

All cells were routinely checked for mycoplasma infection and are negative for mycoplasma infection. Identity and purity of the human cell lines used in this study were tested and confirmed using STR authentication.

### Generation of an RPE-1 PCNA^chromo stable cell line

RPE-1 cells were transfected with 10 μg Cell Cycle-Chromobody plasmid (TagRFP) (from Chromotek) using JET PRIME kit (Polyplus Transfection, 114-07) according to the manufacturer's protocol. After 24 h, 500 μg ml$^{-1}$ G418 (4727878001 from Sigma Aldrich) was added to the cell culture medium and then a mixed population of clones expressing PCNA chromobodies were selected.

### Generation of an RPE-1 FUCCI or RPE-1 CCNB1^AID FUCCI stable cell line

To produce lentiviral particles, HEK 293 cells were transfected with 4 μg pBOB-EF1-FastFUCCI-Puro (Addgene 86849) + 4 μg pMD2.G (Addgene 12259) + 4 μg psPAX2 (Addgene 12260) using a FuGENE HD Transfection Reagent (Promega E2311) in OptiMEM medium (ThermoFisher 51985034). Cells were incubated at 37 °C in a 5% $CO_2$ atmosphere for 16 h and then growth media were removed and replaced by 6 ml fresh OptiMEM. The following day, viral particles were isolated by filtering the medium containing them through a 0.45-μm filter (Sartorius Stedim Biotech 16537). Then, RPE-1 or RPE-1 CCNB1^AID cells[37] were incubated with viral particles in the presence of 8 μg ml$^{-1}$ polybrene (Santa Cruz sc-134220) at 37 °C in a 5% $CO_2$ atmosphere for 24 h. RPE-1 GFP and RFP-positive cells were then collected using Sony SH800 FACS (BD FACSDiva Software Version 8.0.1). RPE-1 or RPE-1 CCNB1^AID clones expressing FUCCI were selected and the cell lines were established from one single clone.

pBOB-EF1-FastFUCCI-Puro[38] was a gift from K. Brindle and D. Jodrell (Addgene 86849).

### Generation of an RPE-1 GFP-53BP1 RFP-H2B stable cell line

This cell line was obtained as described below. In brief, to produce lentiviral particles, HEK 293 cells were transfected with 4 μg pSMPUW-IRIS-Neo-H2BmRFP (Fachinetti laboratory) + 4 μg pMD2.G (Addgene 12259) + 4 μg psPAX2 (Addgene 12260). Then, RPE-1 cells were incubated with viral particles and RPE-1 RFP-positive cells were collected using Sony SH800 FACS (BD FACSDiva Software Version 8.0.1). RPE-1 clones expressing RFP-H2B were selected, and the cell line was established from one single clone.

Then, new lentiviral particles were produced by transfecting HEK 293 cells with 4μg Apple-53BP1trunc (Addgene 69531) + 4 μg pMD2.G (Addgene 12259) + 4 μg psPAX2 (Addgene 12260). RPE-1 RFP-H2B cells were incubated with viral particles, and RPE-1 clones expressing both RFP-H2B and GFP-53BP1 were selected using flow cytometry (Sony SH800 FACS). The cell line was established from one single clone.

Apple-53BP1trunc was a gift from R. Weissleder[39] (Addgene).

### Generation of an RPE-1 shp53 stable cell lines

This cell line was obtained as described below. In brief, to produce lentiviral particles, HEK 293 cells were transfected with 4 μg short hairpin RNA (shRNA) p53-puromycin (Fachinetti laboratory) + 4 μg pMD2.G (Addgene 12259) + 4 μg psPAX2 (Addgene 12260). Then, RPE-1 cells were incubated with viral particles. After 24 h, 5 μg ml$^{-1}$ puromycin (A1113803 from Gibco) was added to the cell culture medium and then a mixed population of clones expressing p53 shRNA was selected.

### Induction of tetraploidy in human cell lines

To induce mitotic slippage, cells were incubated with DMSO (D8418 from Sigma Aldrich) or with 50 μM monastrol (S8439 from Selleckchem) + 1 μM MPI-0479605 (S7488 from Selleckchem) for at least 2 h. Alternatively, CCNB1 depletion in RPE CCNB1^AID cells was induced as described[37]. In brief, cells were treated with 2 μg ml$^{-1}$ doxycycline (D3447 from Sigma Aldrich) + 3 μM asunaprevir (S4935 from Selleckchem) for 2 h. Then, 500 μM auxin (I5148 from Sigma Aldrich) was added to the cell culture medium for at least 4 h. In the figures, mitotic slippage was induced by the combination of monastrol + MPI-0479605 treatment except for the following figures: Figs. 2i, 3a–h, j–o, Extended Data Figs. 2a, b, 7a, d, 8d–h, 9k, in which mitotic slippage was induced by CCNB1 depletion.

To induce cytokinesis failure, cells were incubated with 10 μM genistein (G6649 from Sigma Aldrich) for at least 2 h. Alternatively, cell were incubated with 0.75 μM dihydrocytochalasin D (DCD; D1641 from Sigma-Aldrich) or with 5 μM latrunculin (L5288 from Sigma-Aldrich) for 1 h. In the figures, cytokinesis failure was induced by genistein treatment except for the following figures: Extended Data Fig. 6j, k, in which cytokinesis failure was induced by DCD treatment and Extended Data Fig. 2c, d, in which cytokinesis failure was induced by latrunculin treatment.

To induce endoreplication, cells were incubated with 10 μM SP600125 (S1460 from Selleckchem) for at least 2 h. Alternatively, CCNA2 depletion in RPE CCNA2^AID cells was induced as described[37]. In brief, cells were treated with 2 μg ml$^{-1}$ doxycycline (Sigma Aldrich D3447) for 2 h. Then, 500 μM auxin (Sigma Aldrich I5148) + 3 μM asunaprevir (Selleckchem S4935) was added to the cell culture medium for at least 4 h. In the figures, endoreplication was induced by SP600125 treatment except for Figs. 3q, 4c, Extended Data Figs. 2e, f, 3f, 5d, j, 8c, in which endoreplication was induced through CCNA2 depletion.

### Cell cycle synchronization and DNA replication inhibition

Cells were treated with 1 μM palbociclib (Cdk4/6 inhibitor, Selleckchem S1579), or with 0.5 μM abemaciclib (Cdk4/6 inhibitor, Selleckchem S5716) or with 1 μM K03861 (Cdk2 inhibitor, Selleckchem S8100) for 16 h to synchronize cells at G1/S transition, and were collected (indicated by 'G1 arrest' in the figures). Alternatively, cells were then washed five times with PBS and released in S phase for 10 h before being collected. To extend G1 duration cells were treated with 160 nM palbociclib or with 50 nM abemaciclib or with 400 nM K03861 for 16 h and were collected (indicated by 'G1 lengthening' in the figures). Alternatively, cells were then washed 5 times in PBS and released in S phase for 10 h before being collected.

To inhibit DNA replication, cells were released in S phase in the presence of low doses of Aphidicolin (APH, A0781 from Sigma-Aldrich), a DNA replication polymerase inhibitor, or of PHA767491 (PZ0178 from Sigma-Aldrich), a Cdc7 inhibitor (indicated by 'release in S phase + APH' or 'release in S phase + PHA', respectively, in the figures). Doses were chosen to significantly decrease EdU incorporation without affecting the levels of DNA damage.

### Nucleoside supplementation

Cells were synchronized in G1 using 1 μM palbociclib and then released in S phase (see 'Cell cycle synchronization and DNA replication inhibition') in the presence of nucleosides at the following concentrations: dC 7.3 mg l$^{-1}$ (Sigma Aldrich D0776); dG 8.5 mg l$^{-1}$ (Sigma Aldrich D0901); dU 7.3 mg l$^{-1}$ (Sigma Aldrich D5412); dA 8 mg l$^{-1}$ (Sigma Aldrich D8668)

and dT 2.4 mg l$^{-1}$ (Sigma Aldrich T1895) (+ in the figures) or dC 14.6 mg l$^{-1}$; dG 17 mg l$^{-1}$; dU 14,6 mg l$^{-1}$; dA 16 mg l$^{-1}$ and dT 4,8 mg l$^{-1}$ (++ in the figures).

### Treatments

The drugs were used at the following concentrations: Auxin (Sigma I5148), 500 μM; doxycycline (Sigma D3447), 2 μg ml$^{-1}$; asunaprevir (Selleckchem S4935), 3 μM; monastrol (Selleckchem S8439), 50 μM; MPI-0479605 (Selleckchem S7488), 1 μM; genistein (Sigma G6649), 10 μM; SP600125 (Selleckchem S1460), 10 μM; abemaciclib (Selleckchem S5716), 50 nM or 0.5 μM; KO3861 (Selleckchem S8100), 400 nM or 1 μM; palbociclib (Selleckchem S1579), 120 nM or 1 μM; aphidicolin (Sigma A0781), 0,4 μM or 1 μM; hydroxyurea (Selleckchem S1896), 2 mM; PHA767491 (Sigma PZ0178), 1 μM; RO3306 (Calbiochem 217699), 10 μM; dihydrocytochalasin D (Sigma D1641), 0,75 μM; latrunculin B (Sigma L5288), 5 μM; 5′-chloro-2′-deoxyuridine (CIdU) (Sigma C6891), 100 μM; 5′-iodo-2′-deoxyuridine (IdU) (Sigma I7125), 100 μM.

### Fly husbandry and fly stocks

Flies were raised on cornmeal medium (0.75% agar, 3.5% organic wheat flour, 5.0% yeast, 5.5% sugar, 2.5% nipagin, 1.0% penicillin-streptomycin and 0.4% propionic acid). Fly stocks were maintained at 18 °C. Crosses were carried out in plastic vials and maintained at 25 °C. Stocks were maintained using balancer inverted chromosomes to prevent recombination. Stocks used in this study: *sqh*[140], *pavarotti* RNAi (Pav$^{RNAi}$) (Bloomington *Drosophila* Stock Center BL#42573)[33], UAS-E2F1 (FlyORF F001065) and UAS-Rb (Bloomington Drosophila Stock Center BL#50746).

In all experiments, larvae were staged to obtain comparable stages of development. Egg collection was performed at 25 °C for 24 h. After development at 25 °C, third instar larvae were used for dissection.

### Preparation and imaging of human cells

Cells were plated on cover slips in 12-well plates and treated with the indicated drugs. To label cells, they were fixed using 4% of paraformaldehyde (Electron Microscopy Sciences 15710) + Triton X-100 (2000-C from Euromedex) 0.1% in PBS (20 min at 4 °C). Then, cells were washed three times using PBS-T (PBS + 0.1% Triton X-100 + 0.02% Sodium Azide) and incubated with PBS-T + BSA (Euromedex 04-100-812-C) 1% for 30 min at room temperature. After 3 washes with PBS-T + BSA, primary and secondary antibodies were incubated in PBS-T + BSA 1% for 1 h and 30 min at room temperature, respectively. After 2 washes with PBS, cells were incubated with 3 μg ml$^{-1}$ DAPI (Sigma Aldrich D8417) for 15 min at room temperature. After two washes with PBS, slides were mounted using 1.25% n-propyl gallate (Sigma P3130), 75% glycerol (bidistilled, 99.5%, VWR 24388-295), 23.75% H$_2$O.

Images were acquired on an upright widefield microscope (DM6B, Leica Systems, Germany) equipped with a motorized *xy* stage and a 40× objective (HCX PL APO 40×/1.40−0.70 Oil from Leica). Acquisitions were performed using Metamorph 7.10.1 software (Molecular Devices) and a sCMOS camera (Flash 4V2, Hamamatsu). Stacks of conventional fluorescence images were collected automatically at a z-distance of 0.5 μm (Metamorph 7.10.1 software; Molecular Devices, SCR 002368). Images are presented as maximum intensity projections generated with ImageJ software (SCR 002285).

### Whole-mount tissue preparation and imaging of *Drosophila* larval brains

Brains or salivary glands from third instar larvae were dissected in PBS and fixed for 30 min in 4% paraformaldehyde in PBS. They were washed 3 times in PBST 0.3% (PBS, 0.3% Triton X-100 (Sigma T9284), 10 min for each wash) and incubated for several hours in agitation at room temperature and overnight at 4 °C with primary antibodies at the appropriate dilution in PBST 0.3%. Tissues were washed three times in PBST 0.3% (10 min for each wash) and incubated overnight at 4 °C with secondary antibodies diluted in PBST 0.3%. Brains and salivary glands

were then washed 2 times in PBST 0.3% (30 min for each wash), rinsed in PBS and incubated with 3 μg ml$^{-1}$ DAPI (4′,6-diamidino-2-phenylindole; Sigma Aldrich D8417) at room temperature for 30 min. Brains and salivary glands were then washed in PBST 0.3% at room temperature for 30 min and mounted on mounting media. A standard mounting medium was prepared with 1.25% n-propyl gallate (Sigma P3130), 75% glycerol (bidistilled, 99.5%, VWR 24388-295), 23.75% H$_2$O.

Images were acquired on a spinning disk microscope (Gataca Systems). Based on a CSU-W1 (Yokogawa), the spinning head was mounted on an inverted Eclipse Ti2 microscope equipped with a motorized *xy* stage (Nikon). Images were acquired through a 40× NA 1.3 oil objective with a sCMOS camera (Prime95B, Photometrics). Optical sectioning was achieved using a piezo stage (Nano-z series, Mad City Lab). The Gataca Systems' laser bench was equipped with 405, 491 and 561 nm laser diodes, delivering 150 mW each, coupled to the spinning disk head through a single mode fibre. Multi-dimensional acquisitions were performed using Metamorph 7.10.1 software (Molecular Devices). Stacks of conventional fluorescence images were collected automatically at a z-distance of 1.5 μm (Metamorph 7.10.1 software; Molecular Devices SCR 002368). Images are presented as maximum intensity projections generated with ImageJ software (SCR 002285).

Primary and secondary antibodies were used at the following concentrations: guinea pig anti-CEP192 antibody[41] (1:500; R.B. laboratory), rabbit anti-β catenin (1:250; Sigma-Aldrich C2206, RRID AB 476831), mouse anti-γH2A.X phospho S139 (1:1,000; Abcam ab22551, RRID AB 447150), mouse anti-XRCC1 (1:500; Abcam ab1838, RRID AB 302636), rabbit anti-Rad51 (1:500; Abcam ab133534, RRID AB 2722613), mouse anti-KU80 (1:200; ThermoFisher MA5-12933, RRID AB 10983840), rabbit anti-FANCD2 (1:150; Novusbio NB100-182SS, RRID AB 1108397), mouse anti-53BP1 (1:250; Millipore MAB3802, RRID AB 2206767), rabbit anti-γH2Av (1:500; Rockland 600-401-914, RRID AB 11183655), Alexa Fluor 647 Phalloidin (1:250; ThermoFisher Scientific A22287, RRID AB 2620155), goat anti-rabbit IgG (H+L) Highly Cross-Adsorbed Secondary Antibody, Alexa Fluor 647 (1:250; ThermoFisher A21245, RRID AB 2535813), goat anti-guinea pig IgG (H+L) Highly Cross-Adsorbed Secondary Antibody, Alexa Fluor 488 (1:250; ThermoFisher A11073, RRID AB 253411), goat anti-mouse IgG (H+L) Cross-Adsorbed Secondary Antibody, Alexa Fluor 546 (1:250; ThermoFisher A11003, RRID AB 2534071), goat anti-rabbit IgG (H+L) Highly Cross-Adsorbed Secondary Antibody, Alexa Fluor 546 (1:250; Thermo Fisher Scientific A-11035, RRID AB 2534093).

### Quantitative analysis of DNA damage

***Drosophila* neuroblasts and 3D spheroids.** Quantitative analysis of DNA damage was carried out as previously described[33]. In brief, DNA damage was assessed in *Drosophila* using a γH2Av primary antibody and in 3D spheroids with a γH2AX antibody, and detected with an Alexa Fluor secondary antibody. Confocal volumes were obtained with optical sections at 1.5-μm intervals. Image analysis was performed using Fiji and a custom plugin developed by QUANTACELL. After manual segmentation of the nuclei, a thresholding operation was used to determine the percentage of γH2Av- or γH2AX-positive pixels (coverage) and their average intensity in a single projection. Coverage and intensity were multiplied to obtain the γH2Av or γH2AX index. The threshold used to detect and quantify the γH2Av index in polyploid neuroblasts does not detect any damage in salivary glands. However, it is important to mention that in a fraction of these cells, γH2Av dots (small and of low fluorescence intensity) can be occasionally seen.

**2D human cell lines.** For DNA damage quantification, the signals obtained in cultured cells were different from the signals found in *Drosophila* neuroblasts. To asses DNA damage in human cells, we used an ImageJ software-based plugin developed by QUANTACELL, where γH2AX signals were measured using z-projection stacks after thresholding. Nuclear size, DAPI intensity, the number of γH2AX foci,

γH2AX fluorescence intensity and the percentage of nuclear coverage by γH2AX signal were obtained for each nucleus.

## Time-lapse microscopy

Cells were plated on a dish (627870 from Dutscher) and treated with the indicated drugs. Images were acquired on a spinning disc microscope (Gataca Systems). Based on a CSU-W1 (Yokogawa), the spinning head was mounted on an inverted Eclipse Ti2 microscope equipped with a motorized $xy$ stage (Nikon). Images were acquired through a 40× NA 1.3 oil objective with a sCMOS camera (Prime95B, Photometrics). Optical sectioning was achieved using a piezo stage (Nano-z series, Mad City Lab). Gataca Systems' laser bench was equipped with 405-, 491- and 561-nm laser diodes, delivering 150 mW each, coupled to the spinning disk head through a single mode fibre. Laser power was chosen to obtain the best ratio of signal/background while avoiding phototoxicity. Multi-dimensional acquisitions were performed using Metamorph 7.10.1 software (Molecular Devices). Stacks of conventional fluorescence images were collected automatically at a $z$-distance of 0.5 μm (Metamorph 7.10.1 software; Molecular Devices, RRID SCR 002368). Images are presented as maximum intensity projections generated with ImageJ software (RRID SCR 002285), from stacks deconvolved with an extension of Metamorph 7.10.1 software.

## 3D cultures

**Mitotic slippage on 3D cultures.** To generate spheroids, 500 cells per well were seeded into 96 ultra-low-attachment well plates (Corning 7007) in presence of DMSO (Sigma Aldrich D8418) or with 50 μM monastrol (Selleckchem S8439) and 1 μM MPI-0479605 (Selleckchem S7488). Plates were spin down at 200$g$ for 3 min, to allow spheroid formation, and incubated for 24 h at 37 °C.

**Immunostaining.** Spheroids were collected and washed quickly with PBS before fixation using 4% paraformaldehyde (Electron Microscopy Sciences 15710) in PBS for 40 min. Then, spheroids were permeabilized for 5 min using Triton X-100 (Euromedex 2000-C) 0.3% in PBS and blocked for 30 min using blocking buffer (PBS + 0.3% Triton X-100 + 0.02% sodium azide + 3% BSA). Aggregates were incubated with primary antibodies diluted into blocking buffer overnight. After 3 washes using blocking buffer, spheroids were incubated with secondary antibodies in blocking buffer for 3 h. Cells were then washed several times for 2 h in blocking buffer and mounted on glass with EverBrite (Biotium). For primary and secondary antibodies see 'Immunofluorescence microscopy and antibodies'.

**Imaging and DNA damage analysis.** Spheroids were imaged using an inverted scanning laser confocal (Nikon A1RHD25) equipped with a 100× CFI Plan Apo Lambda S Sil objective (NA 1.35). $z$-stacks were acquired every 0.3 μm. Diploid and tetraploid cells were distinguished using cell and nuclear size and centrosome number. Then, quantitative analysis of DNA damage was carried out (see 'Quantitative analysis of DNA damage').

## EdU staining

EdU incorporation into DNA was visualized with the Click-it EdU imaging kit (Life Technologies C10338), according to the manufacturer's instructions. For human cell lines, EdU was used at a concentration of 1 μM (Extended Data Figs. 6e, 9h) or 10 μM (Extended Data Fig. 5g, h) for the indicated time. Cells were incubated with the Click-it reaction cocktail for 15 min. EdU incorporation in polyploid neuroblasts was done as previously described[33] with a pulse of 2 h before fixation.

## Comet assay

Comet assays were performed using Single Cell Gel Electrophoresis Assay kit (4250-050-ES from Trevigen) according to the manufacturer's instructions. Comets were then imaged using an inverted Eclipse Ti-E Nikon videomicroscope equipped with a 40× CFI Plan Fluor objective. Images were analysed with OpenComet plugin on Fiji. Based on the comet DNA content of DMSO treated cells, a manual threshold was applied to identify diploid from tetraploid cells. The same threshold was applied on the cells treated for mitotic slippage.

## FACS of diploid and tetraploid cells

A mix of diploid and tetraploid cells (see 'Induction of tetraploidy in human cell lines') were incubated with 2 μg ml$^{-1}$ Hoescht 33342 (Sigma Aldrich 94403) for 1 h at 37 °C, 5% $CO_2$. Then, a single cell suspension was generated. Cells were washed using PBS, the supernatant was removed and cells were resuspended in a cold cell culture medium at $1 \times 10^7$ cell per ml and kept at 4 °C during all the experiments. Fluorescence-activated cell sorting (FACS) was performed using Sony SH800 FACS (BD FACSDiva Software Version 8.0.1). Compensation was performed using the appropriate negative control samples. Experimental samples were then recorded and sorted using gating tools to select the populations of interest. RFP$^+$GFP$^-$ cells (G1 cells) were first selected. Then, in this population, DNA content was used to segregate diploid (2n) and tetraploid (4n) G1 cells (Extended Data Fig. 8d). Once gates have been determined, the same number of diploid and tetraploid G1 cells were sorted into external collection tubes. The number of cells was then checked using a cell counter and the same number of diploid an tetraploid cells were collected for western blot analysis. In parallel, post-sort analysis was performed to determine the purity of the sorted populations (Extended Data Fig. 8e).

## Cell cycle analysis and measure of RNA levels by flow cytometry

Cells were detached by treatment with Accutase (Sigma), immediately washed in PBS, fixed in 2 ml 70% ethanol and stored at −20 °C overnight. They were then washed in PBS and staining buffer (BD Pharmingen 554656).

For cell cycle analysis, DNA content was visualized by incubating the cells with 2 μg ml$^{-1}$ Hoescht 33342 (Sigma Aldrich 94403) in staining buffer for 15 min at room temperature. Alternatively, to measure RNA levels, cells were incubated with 2 μg ml$^{-1}$ Hoescht 33342 + pyronin 4 μg ml$^{-1}$ (Santa Cruz sc-203755A) in a staining buffer for 20 min at room temperature. Flow cytometry analysis was done using LSRII (BD Biosciences), by analysing 10,000 cells per condition. Data were then analysed with FlowJo 10.6.0 software (Tree Star).

## E2F1 overexpression

RPE-1 cells were transfected using 0.25 μg pCMVHA E2F1 (Addgene 24225) with a JET PRIME kit (Polyplus Transfection 114-07) according to the manufacturer's protocol. Five hours later, cells were incubated with DMSO (D8418 from Sigma Aldrich) or with 50 μM monastrol (Selleckchem S8439) + 1 μM MPI-0479605 (Selleckchem S7488) to generate tetraploid cells. After 2 h, DMSO or 1 μM palbociclib (Selleckhem S1579) were added to the cell culture medium for 16 h. Cells were then fixed in G1 (T$_0$) or washed five times using PBS and released in S phase and fixed after 10 h (T$_{10}$). The immunofluorescence protocol is described in the corresponding section.

pCMVHA E2F1 was a gift from K. Helin[42] (Addgene plasmid 24225).

## Western blot

For a whole-cell extract, cells were lysed in 8 M urea, 50 mM Tris HCl, pH 7.5 and 150 mM β-mercaptoethanol (Bio-Rad 161-0710), sonicated and heated at 95 °C for 10 min. For chromatin-bound fractions, cells were prepared using the Subcellular Protein Fractionation Kit for Cultured Cells (ThermoFisher Scientific 78840), according to the manufacturer's instructions. Then, samples (equivalent of $2 \times 10^5$ cells) were subjected to electrophoresis in NuPAGE Novex 4–12% Bis-Tris pre-cast gels (Life Technologies NP0321). The same number of cells (see 'FACS sorting of diploid and tetraploid cells') were loaded for diploid and tetraploid conditions, allowing us to compare one diploid cell with

one tetraploid cell. Protein fractions from the gel were electrophoretically transferred to PVDF membranes (PVDF transfer membrane; GE Healthcare RPN303F). After 1 h saturation in PBS containing 5% dry non-fat milk and 0.5% Tween 20, the membranes were incubated for 1 h with a primary antibody diluted in PBS containing 5% dry non-fat milk and 0.5% Tween 20. After three 10-min washes with PBS containing 0.5% Tween 20, the membranes were incubated for 45 min with a 1:2,500 dilution of peroxidase-conjugated antibody. Membranes were then washed three times with PBS containing 0.5% Tween 20, and the reaction was developed according to the manufacturer's specifications using ECL reagent (SuperSignal West Pico Chemiluminescent Substrate; Thermo Scientific 34080).

The background-adjusted volume intensity was calculated and normalized using a H2B signal (H2B was used as a readout of DNA content) for each protein, using Image Lab software version 6.0.1, Bio-Rad Laboratories. All the original uncropped blots (gel source data) are presented in Supplementary Fig. 1.

Primary and secondary antibodies were used at the following concentrations. Mouse anti-α-tubulin (1:5,000; Sigma T9026, RRID AB 477593), mouse anti-CDC45 (1:100; Santa Cruz Biotechnology sc-55569, RRID AB 831146), rabbit anti-PCNA (1:500; Santa Cruz sc56, RRID AB 628110), rabbit anti-actin (1:2,000; Sigma-Aldrich A5060, RRID AB 476738), mouse anti-H2B (1:1,000; Santa Cruz Biotechnology sc-515808), mouse anti-ORC1 (1:100; Santa Cruz Biotechnology sc-398734), mouse anti-MCM2 (1:500; BD Biosciences 610701, RRID AB 398024), mouse anti-E2F1 (1:2,000; Santa Cruz sc251, RRID AB 627476), mouse anti-CDC6 (1:500; Santa Cruz sc-9964, RRID AB 627236), rabbit anti-CDT1 (1:500; Cell Signaling 8064S, RRID AB 10896851), rabbit anti-treslin (1:500; Betyl A303-472A, RRID AB 10953949), goat anti-rabbit IgG (H+L) Cross-Adsorbed Secondary Antibody, HRP (1:2,500; ThermoFisher G21234, RRID AB 2536530), Peroxidase AffiniPure goat anti-mouse IgG (H+L) (1:2500; Jackson ImmunoResearch 115-035-003, RRID AB 10015289).

### 3D reconstruction and analysis
3D videos (see 'Time-lapse microscopy') were imported into Imaris software v.9.6.0 (Bitplane, RRID SCR 007370). For chosen cells, the module 'Spot tracking' of Imaris v.9.6.0 was used to detect the foci, as spots of diameter 0.5 μm in the $xy$-direction and 1 μm in $z$-direction (modelling PSF elongation). Because the volume of the foci changes in time, the option 'Enable growing regions' was used. In each video, the threshold was chosen on the brightest frame (to detect a maximum of the correct spots) and then applied to the whole video. For each cell, at each time point, the number of spots and volumes were recorded. To determine DNA replication timing, we quantified the signal of PCNA fluorescence intensity in the nucleus. This replication timing was characterized independently of any particular behaviour of PCNA. As soon as PCNA fluorescence intensity was detected in the nucleus, $t = 0$ (beginning of S phase) was defined, and when PCNA fluorescence intensity was not detected anymore the last time point was defined (end of S phase). For each condition, at least ten cells (Supplementary Data 1) were studied and the statistics from Imaris v.9.6.0 were averaged at each time point using a MATLAB script.

### Molecular combing
Tetraploid HCT116 were generated by cytokinesis inhibition using 0.75 μM dihydrocytochalasin D (DCD, inhibitor of actin polymerization, Sigma-Aldrich D1641) for 18 h overnight. Afterwards, the cells were washed 3 times with PBS and cultured in DMEM supplemented with 10% FBS and 1% penicillin-streptomycin for additional 20 h. Tetraploid RPE-1 and BJ cells were generated by mitotic slippage or endoreplication (see 'Induction of tetraploidy in human cell lines'). Then, the cells were washed three times with PBS and cultured in DMEM supplemented with 10% FBS and 1% penicillin-streptomycin for an additional 20 h. For each method, we determined that the proportion of tetraploid cells in the treated population is about 40–60%. Due to the presence of diploid cells in the treated population, the consequences of tetraploidization on replication fork speed, fork asymmetry and IOD are most probably underestimated.

Diploid controls and the tetraploid-enriched population were then pulse-labelled with 0.1 mM CldU and 0.1 mM IdU for 30 min and 100,000–300,000 cells per condition were collected for further analysis. The DNA was extracted from cells and prepped following the manufacturer's instructions using the FiberPrep DNA Extraction Kit (Genomic Vision). Subsequently, the prepped DNA was stretched onto coated glass coverslips (CombiCoverslips, Genomic Vision) by using the FiberComb Molecular Combing System (Genomic Vision). The labelling was performed with antibodies against ssDNA, IdU and CldU using the Replication Combing Assay (RCA) (Genomic Vision). The imaging of the prepared cover slips was carried out by Genomic Vision and analysed using the FiberStudio 2.0.1 Analysis Software by Genomic Vision. Replication speed was determined by measuring the combined length of the CldU and IdU tracks. Fork asymmetry was determined by measuring symmetry of the CldU and IdU incorporation by the forks (the length of the first track (CldU) is compared to the length of the second track (IdU)). IOD was determined by measuring distance between two origins on the same fibres.

Antibodies were used at the following concentrations. Rabbit anti-ssDNA (1:5; IBL International 18731, RRID AB 494649), rat anti-CldU (1:10; Abcam Ab6326, RRID AB 2313786), mouse anti-IdU (1:10; BD Biosciences 555627, RRID AB 10015222), mouse Alexa Fluor 647 donkey (1:25; Biozol JIM-715-605-151), rat Alexa Fluor 594 donkey (1:25; Biozol JIM-712-585-153), rabbit Brilliant Violet 480 donkey (1:25; Jackson Immuno Research 711-685-152, RRID AB 2651109).

### Quantitative phase imaging and measurements
Cells were plated on glass-bottom dishes coated with 50 μg ml$^{-1}$ fibronectin for 1 h and rinsed, and trypsinized cells were plated at a concentration of $1.5 \times 10^6$ cells per ml. The cells used for the experiments were seeded in T-25 dishes at a concentration of $0.7 \times 10^6$ cells per ml 2 days before the actual experiment. On the day of the experiment, the cells were detached with EDTA (versene), and plated at a concentration of $1.5 \times 10^6$ cells per ml. For inducing tetraploidy, cells were treated with 2 μg ml$^{-1}$ doxycycline (Sigma Aldrich D3447) for 2 h. Then, 500 μM auxin (Sigma Aldrich I5148) + 3 μM asunaprevir (Selleckchem S4935) was added to the cell culture medium for at least 4 h. The cells were then imaged for 35 h every 20 min to track them throughout their cell cycle.

The cell cycle state was indicated by the FUCCI system; G1 cells express Cdt1–RFP while S/G2 cells express geminin–GFP and mitosis was indicated by the nuclear envelope break down with geminin being present through the cells[43]. To quantify the fluorescence of geminin in the nucleus, first a background subtraction was performed on the images. A region of interest (ROI) was used to define an area containing the background fluorescence in the image. An average value of the ROI was then subtracted from all the frames. Subsequently, a ROI was drawn as close as possible to the cell, and then the mean gray value was measured across all the frames. This helped identify the frames of birth and G1/S transition during the cell cycle.

A detailed protocol for the mass measurement with phasics camera is available in refs. [44,45]. Images were acquired by a Phasics camera every 20 min for 35 h for the duration of the experiment. To obtain the reference image, 32 empty fields were acquired on the dish and a median image was calculated. This reference image was subtracted from the interferograms (images acquired by phasics) by custom written MATLAB scripts to measure the optical path difference. They were then processed to calculate the phase, intensity and phase cleaned images (the background set to 1,000 and the field cropped to remove edges). Background normalization was performed using a gridfit method, and a watershed algorithm was used to separate cells which came in contact with each other. Mass was calculated by integrating the intensity of the whole cell.

## Sequencing and AneuFinder analysis

A mixed population of diploid and tetraploid RPE-1 CCNB1[AID] FUCCI cells were synchronized in G1 using 1 μM palbociclib (Selleckchem S1579) for 16 h or released in S phase for 20 h in the presence of 10 μM RO3306 (Calbiochem 217699) in order to block cells in the subsequent G2/M. G1 and G2/M diploid and tetraploid cells were then isolated using cell sorting (see 'FACS sorting of diploid and tetraploid cells') and collected in a 96-well plate.

Sequencing was performed using a NextSeq 500 (Illumina; up to 77 cycles; single end). The generated data were subsequently demultiplexed using sample-specific barcodes and changed into fastq files using bcl2fastq (Illumina; version 1.8.4). Reads were afterwards aligned to the human reference genome (GRCh38/hg38) using Bowtie2 (version 2.2.4; ref. [46]. Duplicate reads were marked with BamUtil (version 1.0.3; ref. [47]. The aligned read data (bam files) were analysed with the copy number calling algorithm AneuFinder[48] (https://github.com/ataudt/aneufinder). Following GC correction and blacklisting of artefact-prone regions (extreme low or high coverage in control samples), libraries were analysed using the dnacopy and edivisive copy number calling algorithms with variable width bins (average bin size = 1 Mb; step size = 500 kb). The G1 samples were analysed with an euploid reference[49]. The G1 samples were used as a reference for the analysis of the G2/M samples (G1 diploid for G2/M diploid and G1 polyploid for G2/M polyploid). Aneuploid libraries were not used as a reference and blacklists were constructed using the example from Bioconductor as a guideline. The RPE-1 diploid G1 sample (2n) was analysed with the standard version of AneuFinder (from Bioconductor) while the other samples were analysed with the developer version of AneuFinder (from GitHub; 4n and 8n samples). The ground ploidy for these samples was constrained between 3.5 and 4.5 (4n samples) or between 7.5 and 8.5 (8n samples; parameters: min.ground.ploidy and max.ground.ploidy). Results were afterwards curated by requiring a minimum concordance of 95 % (2n sample) or 90 % (4n and 8n samples) between the results of the two algorithms. Libraries with on average less than 10 reads per chromosome copy of each bin (2-somy: 20 reads, 3-somy: 30 reads, etc.) were discarded. This minimum number of reads comes down to roughly 60,000 for a diploid genome in G1 phase (2n) up to 240,000 for a polyploid genome in G2/M phase (8n). Analysis of the BJ samples showed aberrations (wavy patterns) that resulted in wrongly called segments with a copy number which is either one higher or one lower than the expected state (when euploid). The means of the read counts (read counts of the bins) of these states were too close to the mean of the expected state (for example, mean 5-somy too close to mean 4-somy; 4n sample; Supplementary Methods 1). When more than 1 % of the genome was classified as such (for example, more than 1 % 5-somy), a non-rounded version of the copy number of the state was calculated using the mean of the expected state (ploidy of euploid sample) as a reference:

Non-rounded copy number.state = Mean state/(mean.expected state/copy number.expected state)

Example 5-somy (4n sample):

Non-rounded copy number.5-somy = Mean.5-somy/(Mean.4-somy/4)

This was done to quantify the distance between the two states. The values are typically between −0.5 and +0.5 of the state under consideration (for example, 5-somy; between 4.5 and 5.5), which will result in a rounded value equal to the state. The libraries with aberrations have typically a deviation of 0.25 and more from the expected value (Supplementary Methods 1). Libraries that showed a deviation of more than 0.25 were therefore discarded (For 5-somy; a value lower than 4.75 or higher than 5.25). By applying this cut-off, we eliminated libraries that clearly showed this aberration (Supplementary Methods 1) while preserving true aneuploid libraries (Supplementary Methods 1). This specific method was only used for the BJ samples.

## GSEA with TCGA PanCancer data

GSEA was performed using GSEA software v.4.2.1[50,51]. The normalized mRNA expression (Illumina HiSeq_RNASeqV2, RSEM) from pan cancer studies were downloaded from https://www.cbioportal.org/: detailed information about RNA sequencing experiment and tools used can be found at the NCI's Genomic Data Commons (GDC) portal https://gdc.cancer.gov. The ploidy status for bladder urothelial carcinoma (156 near-diploid and 200 near-tetraploid samples), Lung adenocarcinoma (205 near-diploid and 240 near-tetraploid samples), and ovarian serous cystadenocarcinoma (116 near-diploid and 130 near-tetraploid samples) were extracted from[36]. In addition to ranked list of genes and ploidy status, we use gene sets derived from the GO Biological Process ontology to assess significant pathway enrichment between near-diploid and near tetraploid tumors in GSEA tool. GSEA is a computational method that determines whether a defined set of genes shows statistically significant concordant differences between two biological states (for example, two distinct phenotypes), using the algorithm based on the calculation of an enrichment score (ES), the estimation of significance level of ES (nominal *P* value) and adjustment for multiple hypothesis testing (ES normalization and FDR calculation)[50].

## Quantifications

Image analysis and quantifications were performed using Image J software V2.1.0/1.53c, https://imagej.net/software/fiji/downloads. To quantify the colocalizations between two signals (Extended Data Figs. 3i, m, 4g, j) we calculated the Manders coefficient using the JACOP plugin with Image J V2.1.0/1.53c software. We determined that the colocalizations between γH2AX signal and EdU, FANCD2 or RAD51 signals are not random using an home-made based Costes randomization on nuclear area with Image J software. 1000 randomizations of the pixel positions were performed for each condition (Supplementary data 2). 3D videos (Extended Data Figs. 3c, 6c, 9c, d) were corrected using the 3D correct drift plugin with Image J V2.1.0/1.53c software to keep the cell of interest at the centre of the region of interest. The nuclear area and DAPI intensity were measured using the wand tool with Image J V2.1.0/1.53c software. For the figures, images were processed on Image J V2.1.0/1.53c software, and mounted using Affinity Designer (https://affinity.serif.com/fr/designer/).

## Statistics and reproducibility

At least two (*n*) independent experiments were carried out to generate each dataset, and the statistical significance of differences was calculated using GraphPad Prism (RRID SCR 002798) version 7.00 for Mac (GraphPad Software). The statistical test used for each experiment is indicated in the figure legends. Each representative image (Figs. 2a, c, 3a, e, g, k, n, 4a, Extended Data Figs. 2a, c, e, l, 3a, g, 4c, 5g, 6c, 9c, d, 10a) originates from a dataset composed of at least two (*n*) independent experiments.

## Reporting summary

Further information on research design is available in the Nature Research Reporting Summary linked to this paper.

## Data availability

Source data are available at https://doi.org/10.6084/m9.figshare.19137323.v1. Source data are provided with this paper.

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

**Acknowledgements** We authors acknowledge the Cell and Tissue Imaging platform (PICT-IBiSA), member of the French National Research Infrastructure France-BioImaging (ANR10-INBS-04) and the Nikon Imaging center from Institut Curie for microscopy. We thank L. Guyonnet, A. Chipont, C. Guerrin and C. Roulin from the Cytometry and RADEXP platforms of Institut Curie; and V. Marthiens, S. Lambert, E. Schwob, W. Marshall, G. Almouzni, J. S. Hoffmann, D. Fachinetti, M. Budzyk, F. Edwards, G. Fantozzi, R. Salamé, A. Goupil, C. Chen, F. Perez and N. Ganem for helpful discussions and/or comments on the manuscript. This work was supported by FOR2800/STO918-7 to Z.S., ERC CoG (ChromoNumber-725907) to R.B., Institut Curie and the CNRS. The Basto laboratory is a member of the Cell(n)Scale Labex.

**Author contributions** S.G. and R.B. conceived the project and wrote the manuscript. DNA damage characterization and the analysis of its origins were conceived together with Z.S. S.G. did most of the experiments and data analysis presented here. M.N. did the initial observations of high levels of DNA damage in *Drosophila* polyploid neuroblasts. R.W., A.E.T., D.C.J.S. and F.F. did the scDNA-seq and bio-informatical analysis. K.K., S.V.B. and Z.S. contributed to DNA combing. A.-S.M. helped with image quantifications and analysis. A.S. performed the 3D spheroids experiments and helped with image quantification. O.G. performed the bio-informatical analysis of ovarian TCGA tumours. N.S. and M.P. performed the quantitative phase imaging experiments and analysis and H.H. contributed with unpublished cell lines. All authors read and comment on the manuscript.

**Competing interests** The authors declare no competing interests.

**Additional information**
**Correspondence and requests for materials** should be addressed to Simon Gemble or Renata Basto.

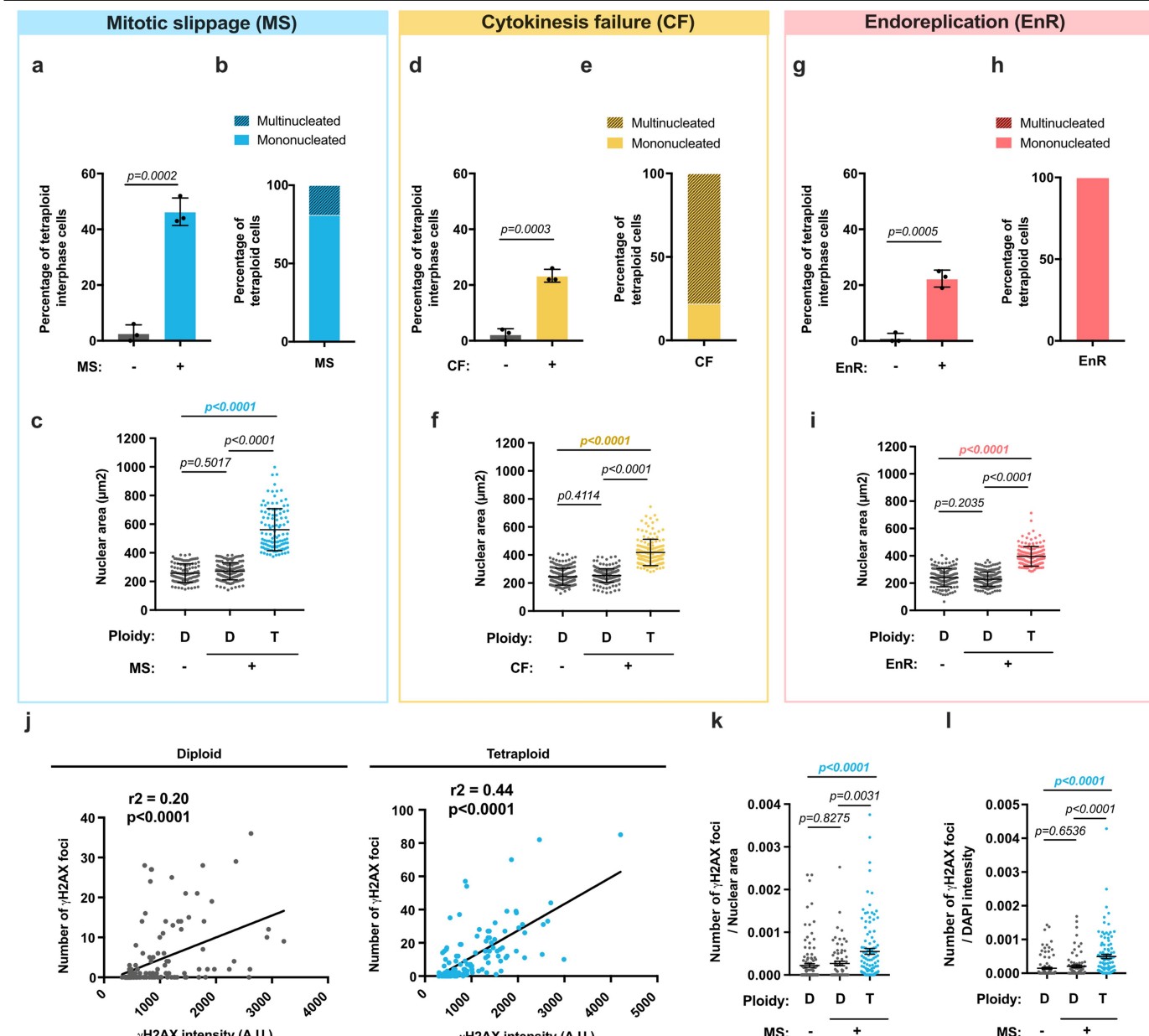

**Extended Data Fig. 1 | Characterization of RPE-1 cells upon WGD. (a, d** and **g)** Graphs showing the percentage of tetraploid interphase RPE-1 cells in the indicated experimental conditions. Mean ± SEM, >100 interphase cells were analyzed from three independent experiments. **(b, e** and **h)** Graphs showing the percentage of mono- and multinucleated RPE-1 tetraploid cells in the indicated experimental conditions. Mean ± SEM, >100 interphase cells were analyzed from three independent experiments. **(c, f** and **i)** Graphs representing the nuclear area in diploid (D) and tetraploid (T) RPE-1 cells. Mean ± SEM, >100 interphase cells were analyzed from three independent experiments. **(j)** Graph showing the correlation between the number of γH2AX foci and γH2AX foci intensity in diploid (left panel, gray) and tetraploid (right panel, blue) RPE-1 cells induced through MS. >100 interphase cells were analyzed from three independent experiments. **(k–l)** Graphs showing the number of γH2AX foci relative to nuclear area **(k)** or DAPI fluorescence intensity (FI) **(l)** in diploid (gray) and tetraploid (blue) RPE-1 cells induced through MS. t-test (two-sided) **(a, d** and **g)**. ANOVA test (one-sided) **(c, f, i, k** and **l)**. Pearson test (two-sided) **(j)**.

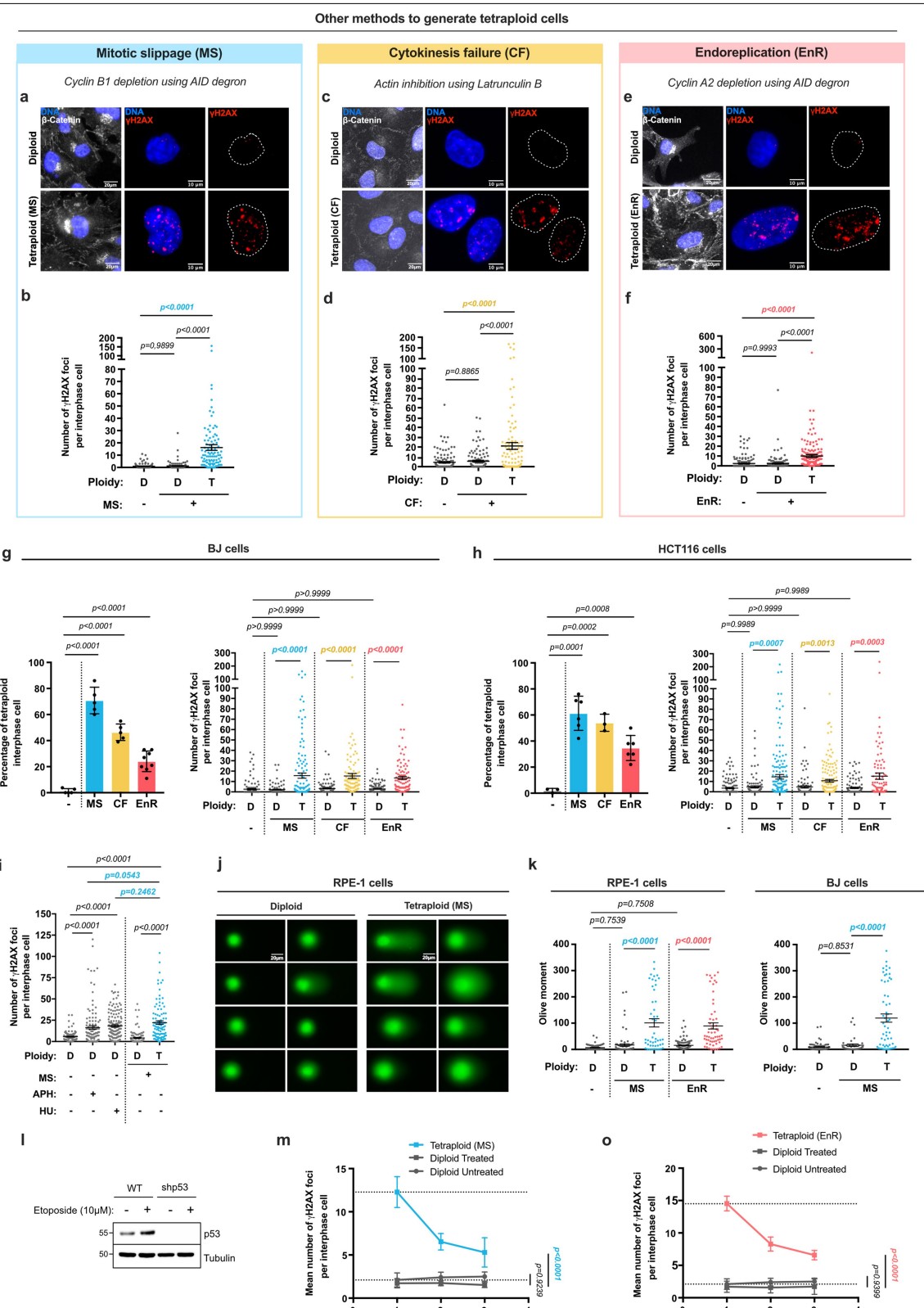

**Extended Data Fig. 2** | See next page for caption.

**Extended Data Fig. 2 | Additional methods and cell lines confirm that WGD generates high levels of DNA damage within the first interphase. (a, c** and **e)** Images showing diploid and tetraploid (generated as indicated) RPE-1 cells labeled with γH2AX (red) and β-Catenin (gray) antibodies. DNA in blue. **(b, d** and **f)** Graphs showing the number of γH2AX foci in diploid (D) and tetraploid (T) RPE-1. Mean ± SEM, >100 interphase cells were analyzed from at least three independent experiments. **(g–h)** Left - Graph showing the percentage of tetraploid interphase cells in BJ **(g)** or HCT116 **(h)** cell lines. Mean ± SEM, >100 interphase cells were analyzed from at least three independent experiments. Right - Graph representing the number of γH2AX foci in diploid and tetraploid BJ **(g)** or HCT116 **(h)** cells. Mean ± SEM, >100 interphase cells were analyzed from at least three independent experiments. **(i)** Graph showing the number of γH2AX foci in diploid (gray) or tetraploid (blue) RPE-1 cells treated with 1 µM APH or 2 mM HU. Mean ± SEM, >100 interphase cells were analyzed from at least three independent experiments. **(j)** Comet images from diploid (left) and tetraploid (right) RPE-1 cells. **(k)** Graph showing the olive moment in diploid and tetraploid RPE-1 (left) or BJ (right) cell lines. Mean ± SEM, >100 comets were analyzed from two independent experiments. **(l)** p53 and tubulin levels assessed by western blot. Etoposide was added as a control for the increased p53 levels. **(m** and **o)** Graphs representing the mean number of γH2AX foci per interphase cell over time (days in culture) in diploid and tetraploid RPE-1 cells. Mean ± SEM, >100 interphase cells were analyzed from two independent experiments. The dotted lines indicate nuclear area. ANOVA test (one-sided) **(b, d, f, g, h, i, k, m** and **o)**.

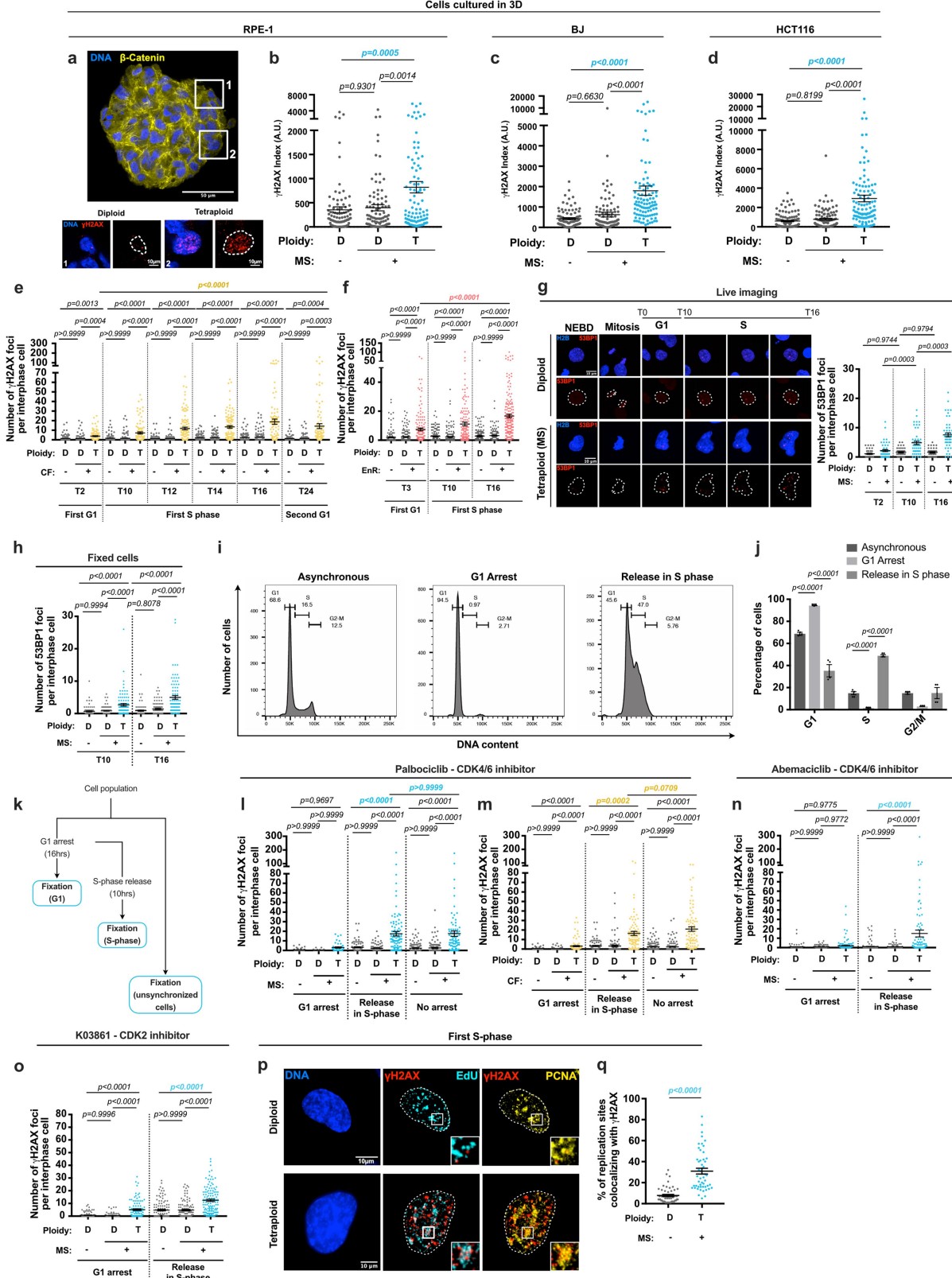

**Extended Data Fig. 3** | See next page for caption.

**Extended Data Fig. 3 | DNA damage is generated during the first S-phase upon WGD. (a)** 3D RPE-1 spheroid low magnification (top) and insets of two cells showing diploid and tetraploid nuclei (bottom) induced through MS labeled with γH2AX (red) and β-Catenin (yellow) antibodies. DNA in blue. **(b–d)** γH2AX index in diploid and tetraploid RPE-1 **(b)**, BJ **(c)** and HCT116 **(d)** spheroids. Mean ± SEM, > 95 interphase cells were analyzed from at least two independent experiments. **(e–f)** γH2AX foci in diploid and tetraploid RPE-1 cells over time. **(g)** Left - Stills of RPE-1 cells expressing RFP-H2B and GFP-53BP1 time lapse videos. Right- 53BP1 foci number in diploid and tetraploid cells. Mean ± SEM, > 40 interphase cells were analyzed from three independent experiments. **(h)** 53BP1 foci number in fixed diploid and tetraploid RPE-1. **(i)** Cell cycle distribution of RPE-1 cells in the indicated conditions. **(j)** Percentage of RPE-1 cells in G1, S and G2-M in the indicated conditions. Mean ± SEM, >30 000 cells from at least three independent experiments. **(k)** Workflow used to analyze G1 or S-phase cells. **(l and m)** γH2AX foci number in diploid and tetraploid RPE-1 cells as indicated. Experiments **(l and m)** share the same reference control. **(n, o)** γH2AX foci number in diploid and tetraploid RPE-1 cells synchronized in G1 using 0,5 μM abemaciclib **(n)** or 1 μM K03861 **(o)** or released in S-phase. **(p)** Images of γH2AX (red) and EdU (cyan)/ PCNA (yellow) foci co-localization in S-phase in diploid and tetraploid RPE-1 cells. DNA in blue. White squares highlight higher magnifications. **(q)** Percentage of replication sites (EdU) colocalizing with γH2AX foci. Mean ± SEM, >50 interphase cells were analyzed from at least three independent experiments. For **(e, f, h, l–o)** Mean ± SEM, >100 interphase cells were analyzed from at least three independent experiments. The dotted lines indicate nuclear area. ANOVA test (one-sided) **(b, c, d, e, f, g, h, j, l, m, n** and **o)**. t-test (two-sided) **(q)**.

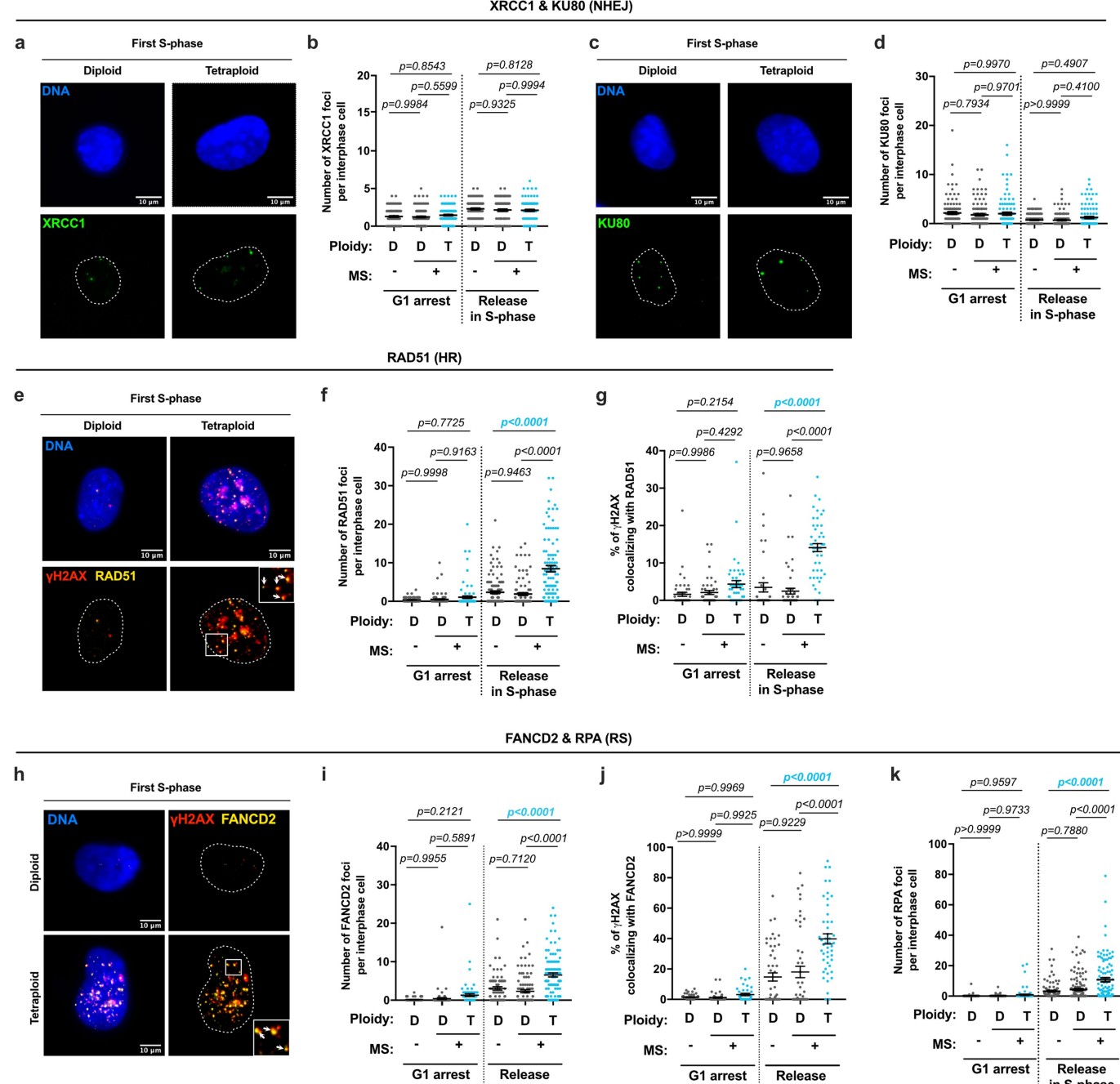

**Extended Data Fig. 4 | DNA damage in newly born tetraploid S-phase cells is associated with HR and RS-associated markers. (a** and **c)** Diploid and tetraploid RPE-1 S-phase cells labeled with XRCC1 (**a**, in green) or KU80 antibodies (**c**, in green). DNA in blue. (**b** and **d)** Number of XRCC1 (**b)** or KU80 (**d)** foci in diploid and tetraploid RPE-1 cells synchronized in G1 using 1μM palbociclib or released in S-phase. Mean ± SEM, >100 interphase cells were analyzed from at least three independent experiments. (**e)** Images showing γH2AX (red) and RAD51 (yellow) foci colocalization (white arrows) in diploid and tetraploid RPE-1 cells. DNA in blue. The white squares correspond to higher magnification regions. (**f)** RAD51 foci number in diploid and tetraploid RPE-1 cells arrested in G1 using 1 μM palbociclib or released in S-phase. Mean ± SEM, >100 interphase cells analyzed from at least three independent experiments. (**g)** Percentage of colocalizing γH2AX and RAD51 signals in diploid and

tetraploid RPE-1 cells arrested in G1 using 1 μM palbociclib or released in S-phase. Mean ± SEM, >50 interphase cells were analyzed from at least three independent experiments. (**h)** Images showing the colocalization (white arrows) of γH2AX (red) and FANCD2 (yellow). DNA in blue. (**i)** FANCD2 foci number in diploid and tetraploid RPE-1 cells. Mean ± SEM, >80 interphase cells were analyzed from at least three independent experiments. (**j)** Graph representing the percentage of γH2AX signal colocalizing with FANCD2 foci in diploid and tetraploid RPE-1 interphase cells. Mean ± SEM, >50 interphase cells were analyzed from at least three independent experiments**. (k)** Graph showing RPA number foci in diploid and tetraploid RPE-1 cells arrested in G1 using 1 μM palbociclib or released in S-phase. Mean ± SEM, >100 interphase cells were analyzed from at least three independent experiments. The dotted lines indicate the nuclear area. ANOVA test (two sided) (**b, d, f, g, i, j** and **k)**.

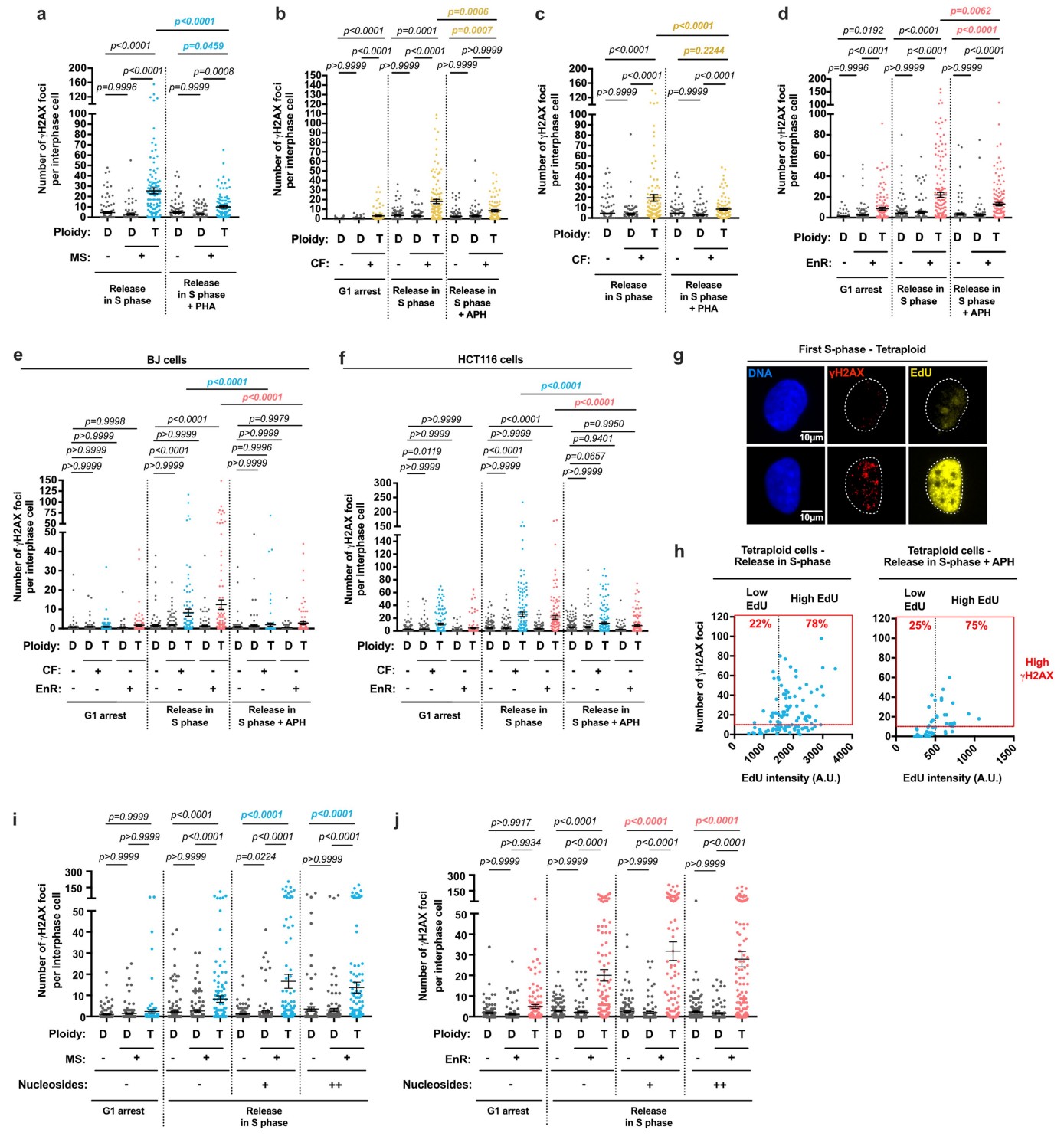

**Extended Data Fig. 5 | DNA damage in newly born tetraploid cells is generated in a DNA replication-dependent manner. (a)** γH2AX foci number in diploid and tetraploid RPE-1 cells released in S-phase ± 1 μM PHA. Mean ± SEM, >100 interphase cells were analyzed from at least three independent experiments. **(b–c)** γH2AX foci number in diploid and tetraploid RPE-1 cells, arrested in G1 using 1 μM palbociclib or released in S-phase ± 400 nM aphidicolin (APH) **(b)** or 1 μM PHA **(c)**. Mean ± SEM, >100 interphase cells were analyzed from at least three independent experiments. **(d)** γH2AX foci number in diploid and tetraploid RPE-1 cells released in S-phase ± 400 nM APH. Mean ± SEM, >100 interphase cells were analyzed from at least three independent experiments. **(e–f)** γH2AX foci number in diploid and tetraploid BJ **(e)** or HCT116 **(f)** cells, released in S-phase ± 400 nM APH. Mean ± SEM, >100 interphase cells were analyzed from at least three independent experiments. **(g)** Images showing EdU ± tetraploid RPE-1 cells. γH2AX antibodies in red, EdU in yellow and DNA in blue. **(h)** γH2AX foci number relative to EdU intensity in RPE-1 tetraploid cells released in S-phase untreated (left panel) or treated (right panel) with 400 nM aphidicolin (APH). Mean ± SEM, >100 interphase cells were analyzed from at least three independent experiments. **(i, j)** γH2AX foci number in diploid and tetraploid cells (**i**, blue) or EnR (**j**, red), synchronized in G1 using 1 μM palbociclib or released in S-phase ± nucleosides at two different concentrations (methods). Mean ± SEM, >100 interphase cells were analyzed from at least three independent experiments. The dotted lines indicate the nuclear area. ANOVA test (one-sided) (**a, b, c, d, e, f, i** and **j**). Pearson test (two-sided) (**h**).

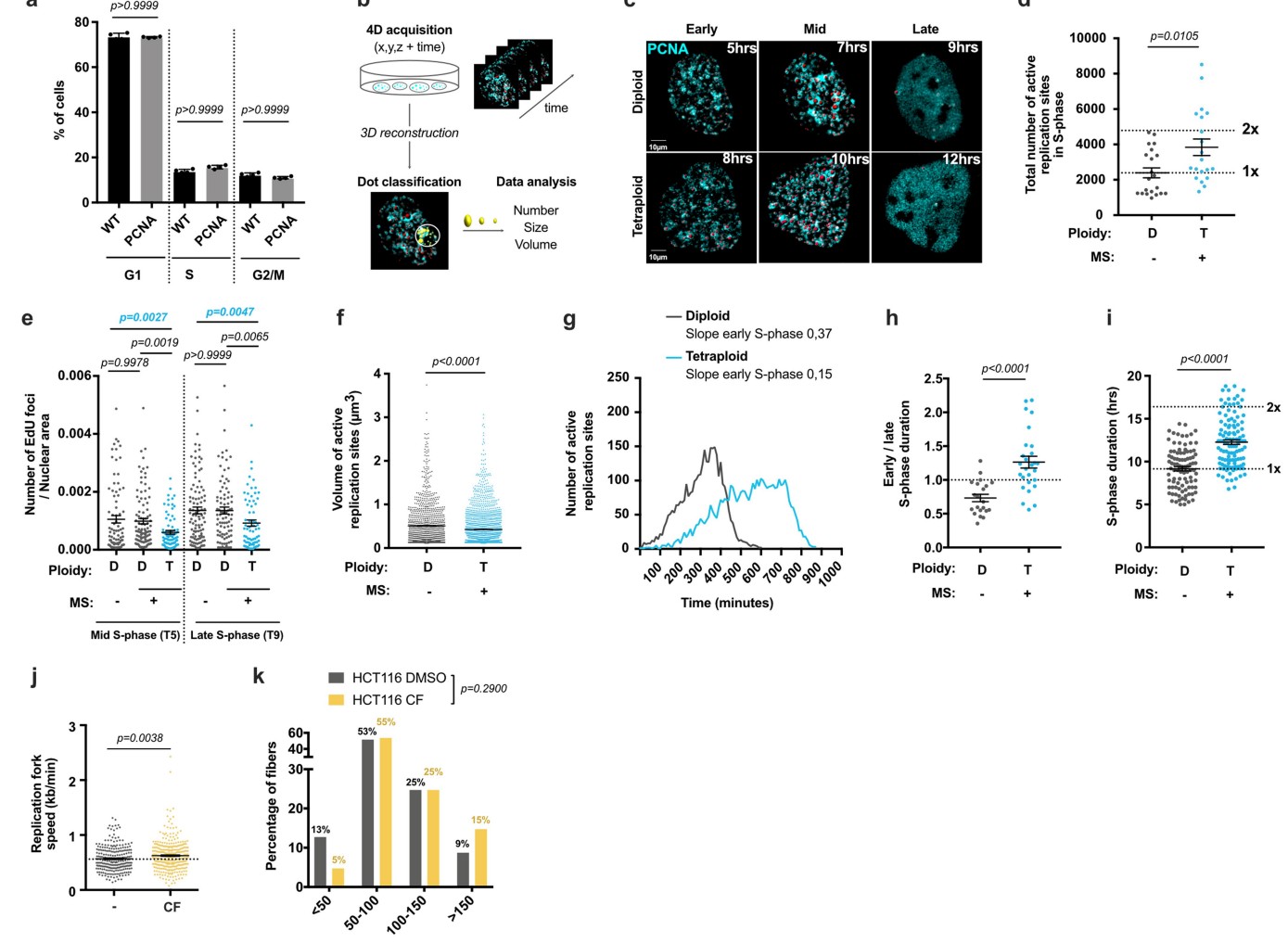

**Extended Data Fig. 6 | DNA replication dynamics is impaired during the first S-phase in tetraploid cells. (a)** Percentage of cells per cell cycle phase in RPE-1 (dark gray) and RPE-1 PCNA^chromo cell lines (light gray). **(b)** Workflow depicting methods used to process and analyze DNA replication by time-lapse. **(c)** Stills of time lapse movies of diploid and tetraploid RPE-1 PCNA^chromo cells. Active replication sites are visualized using PCNA chromobodies (in cyan) and reconstructed using *Imaris* in 3D (in red). **(d)** Total number of active replication sites in S-phase in diploid and tetraploid RPE-1 cells. Mean ± SEM, >20 S-phase cells were analyzed from three independent experiments. **(e)** EdU foci number relative to nuclear area in diploid and tetraploid RPE-1 cells in mid (T5) or late (T9) S-phase. Mean ± SEM, >100 interphase cells were analyzed from at least three independent experiments. **(f)** Volume of active replication sites (in μm³) for diploid and tetraploid RPE-1 PCNA^chromo cells. Mean ± SEM, at least 1000

active replication sites were analyzed from three independent experiments. **(g)** Mean number of active replication sites over time in diploid and tetraploid RPE-1 cells. >20 S-phase cells were analyzed from two independent experiments (see Supplementary Data 1). **(h)** Ratio of early/late S-phase duration in diploid or tetraploid RPE-1 PCNA^chromo cells ± extended G1 duration. Mean ± SEM, > 70 cells from two independent experiments were analyzed. **(i)** S-phase duration in diploid or tetraploid RPE-1 PCNA^chromo cells ± extended G1 duration. Mean ± SEM, > 70 cells from two independent experiments were analyzed. **(j)** Replication fork speed in diploid and tetraploid HCT116 cells. Mean ± SEM, > 250 replication forks were analyzed. **(k)** Proportion of fibers with the indicated inter-origin distance (kb) in diploid or tetraploid HCT116 cells. Mean ± SEM, > 75 replication origins were analyzed. ANOVA test (one-sided) **(a** and **e)**. t-test (two-sided) **(d, f, h, i, j** and **k)**.

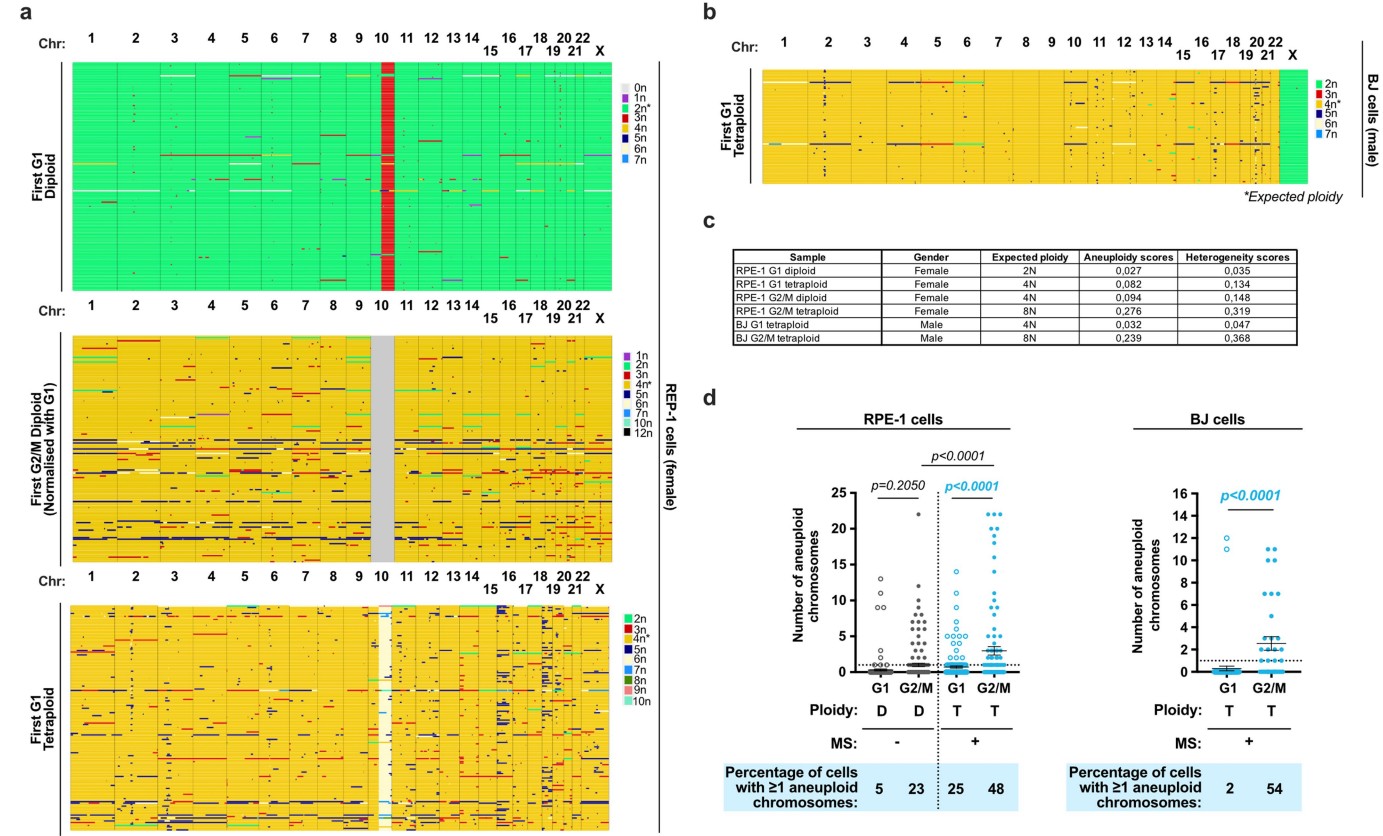

**Extended Data Fig. 7 | Genome wide analysis of RPE-1 and BJ tetraploid cells.** (**a**, **b**) Genome-wide copy number plots of G1 and G2/M diploid RPE-1 cells and G1 tetraploid RPE-1 cells (**a**) were generated using the standard version of the Aneufinder algorithm and genome-wide copy number plots of G1 tetraploid BJ cells (**b**) were generated using a modified version of the Aneufinder algorithm (see methods). G2/M conditions were normalized using G1 cells. Each row represents a cell. The copy number state (in 1-Mb bins) is indicated in color (with aberrations contrasting from green in diploid G1 (2n) or from yellow in diploid G2/M or tetraploid G1 (4n). (**c**) Table showing aneuploidy and heterogeneity scores in the indicated conditions. (**d**) Graph showing the number of aneuploid chromosomes per cell in the diploid G1 and G2/M (in gray) and in tetraploid G1 and G2/M (in blue) cells. The percentage of cells with ≥1 aneuploid chromosome is indicated under the graph. ANOVA test (one-sided) (**d**, left panel). t-test (two-sided) (**d**, right panel).

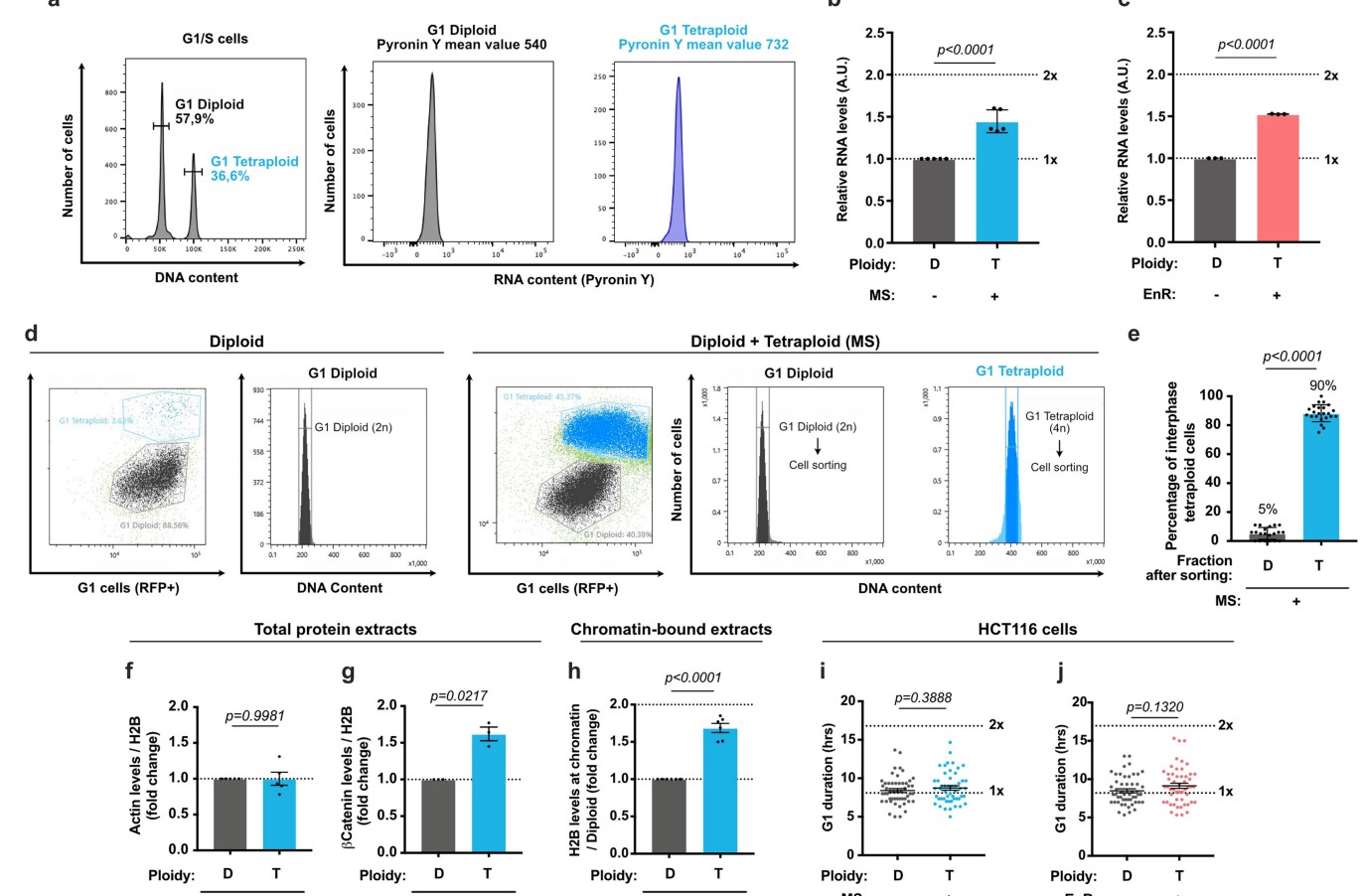

**Extended Data Fig. 8 | DNA damage analysis in 3D cultures. (a)** Left panel - Representative cell cycle distribution of diploid and tetraploid RPE-1 cells. Right panels - RNA content in diploid (in gray) and tetraploid (in blue) populations. **(b, c)** Graphs showing the relative RNA levels in diploid (D, in gray) and tetraploid (T) cells generated through MS (**b**, blue) or EnR (**c**, red). **(d)** Representative images of cell sorting experiments according to cell cycle stage (RFP+ for G1 cells) and DNA content. **(e)** Graph showing the percentage of interphase tetraploid cells in diploid (gray) and tetraploid (blue) RPE-1 cell populations obtained after cell sorting. Mean ± SEM, >100 interphase cells from at least three independent experiments were analyzed. **(f, g)** Graphs

representing actin **(f)** and β-Catenin **(g)** levels relative to H2B levels (fold change) in total protein extracts from diploid (gray) and tetraploid (blue) cells. Mean ± SEM from at least three independent experiments. **(h)** Graph showing H2B levels in the chromatin bound fraction in diploid (gray) and tetraploid (blue) cells. Mean ± SEM from at least three independent experiments. **(i, j)** Graph representing G1 duration in diploid (gray) or tetraploid cells generated through MS (**i**, blue) or EnR (**j**, red). The dotted lines indicate the nuclear area. The white squares correspond to higher magnification. t-test (one-sided) **(b, c, e, f, g, h, i** and **j)**.

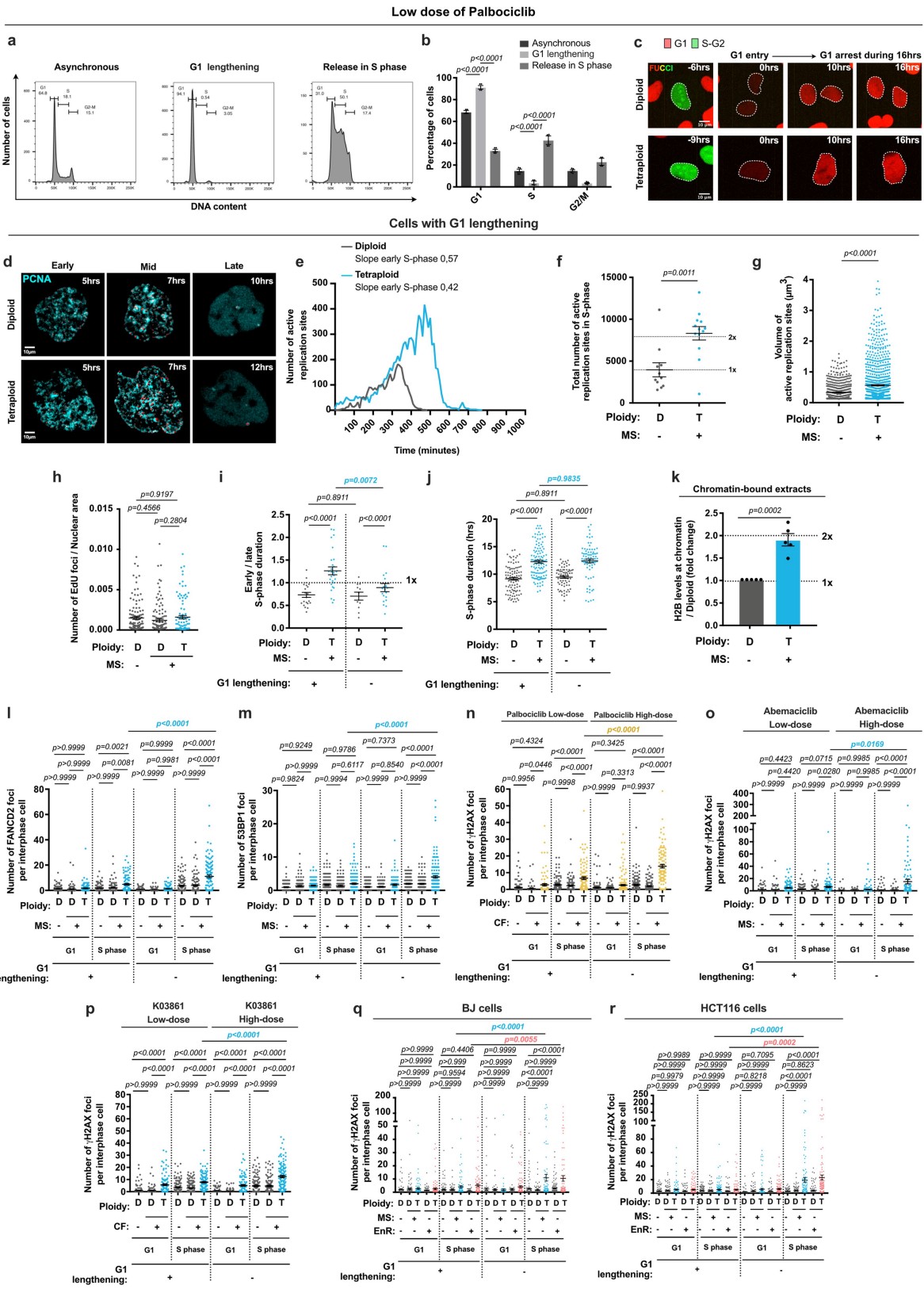

**Extended Data Fig. 9 |** See next page for caption.

**Extended Data Fig. 9 | G1 lengthening restores DNA replication dynamics and results in a decrease in the levels of DNA damage in tetraploid cells.** (**a, b**) RPE-1 cell cycle profile and percentage of cells in the indicated conditions. Mean ± SEM, >30 000 cells from at least three independent experiments. (**c**) RPE FUCCI in diploid and tetraploid cells treated with 160 nM palbociclib. (**d**) Stills of time lapse videos of diploid and tetraploid RPE-1 PCNA[chromo] cells with extended G1. Active replication sites visualized using PCNA chromobodies (cyan) and reconstructed using *Imaris* in 3D (red). (**e**) Active replication sites average number over time with extended G1. Mean ± SEM, >11 S-phase cells analyzed, two independent experiments (see Supplementary Data 1). (**f**) Active replication sites total number with extended G1. Mean ± SEM, >11 S-phase cells were analyzed, two independent experiments. (**g**) Active replication sites volume (μm3) with extended G1. Mean ± SEM, >1000 Active replication sites analyzed, three independent experiments. (**h**) EdU foci number relative to nuclear area with extended G1. Mean ± SEM, >100 interphase cells, at least three independent experiments. (**i**) Ratio of early/late S phase duration ± extended G1. Mean ± SEM, >70 cells, two independent experiments. (**j**) S-phase duration ± extended G1. Mean ± SEM, >70 cells, two independent experiments. (**k**) H2B levels in chromatin bound extracts. Mean ± SEM, four independent experiments. (**l** and **m**) FANCD2 or 53BP1 foci number in cells synchronized in G1 or released in S-phase ± extended G1. (**n**–**p**) γH2AX foci number in cells synchronized in G1 using the indicated treatments or released in S-phase ± extended G1. (**q** and **r**) γH2AX foci number in diploid and tetraploid BJ (**q**) or HCT116 (**r**) cells synchronized in G1 or released in S-phase ± extended G1. (**l**–**r**) Mean ± SEM, >100 interphase cells were analyzed, at least three independent experiments. The dotted lines indicate the nuclear area. ANOVA test (one-sided) (**b, h, i, j, l, m, n, o, p, q** and **r**). T-test (two-sided) (**f, g** and **k**).

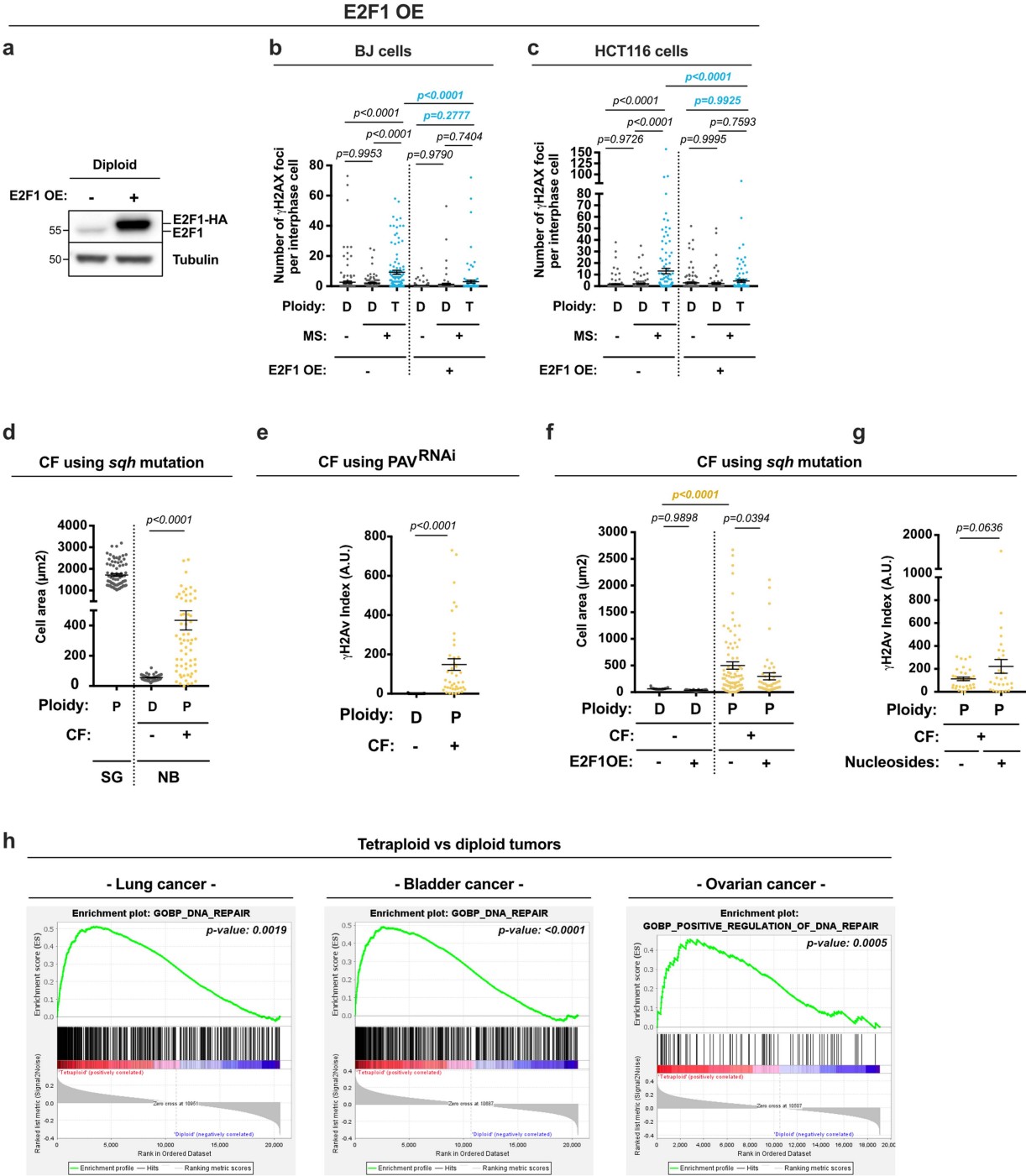

**Extended Data Fig. 10 | E2F1OE decreases DNA damage levels in tetraploid human cell lines and in *Drosophila* NBs. (a)** Western blot documenting the levels of E2F1 and tubulin from cell lysates obtained from diploid RPE-1 cells ± E2F1-HA over-expression (OE). **(b–c)** Graphs representing the number of γH2AX foci per interphase cell in diploid (D) and tetraploid (T) BJ **(b)** or HCT116 **(c)** cells released in S-phase ± E2F1 OE. Mean ± SEM, >100 interphase cells were analyzed from at least three independent experiments. **(d)** Graph showing wild type salivary gland cell (in gray), diploid (in gray) and polyploid NBs (in yellow) area (in μm²). Mean ± SEM, >60 cells were analyzed per condition. **(e)** Graph showing γH2Av indexes in diploid (in gray) or polyploid NBs (in yellow) induced through

CF by depleting Pavarotti. Mean ± SEM, >40 cells were analyzed per condition. **(f)** Graph showing the cell area (μm²) of diploid (gray) and polyploid NBs (yellow) ± E2F1OE. Mean ± SEM, >30 cells were analyzed per condition. **(g)** Graph representing the γH2Av index in polyploid NBs ± 10μM nucleosides. Mean ± SEM, >28 cells were analyzed per condition. **(h)** Gene set enrichment analysis from GSEA. Plots show significant enrichment of DNA repair genes in near-tetraploid tumors when compared to near-diploid tumors in lung, bladder and ovarian cancers (TCGA pan cancer data set). **(h)** p value from false discovery rate (FDR; methods). ANOVA test (one-sided) **(b, c, d, f)**. t-test (two-sided) **(e** and **g)**.

Simon Gemble

# Reporting Summary

## Statistics

For all statistical analyses, confirm that the following items are present in the figure legend, table legend, main text, or Methods section.

| n/a | Confirmed | |
|---|---|---|
| ☐ | ☒ | The exact sample size (*n*) for each experimental group/condition, given as a discrete number and unit of measurement |
| ☐ | ☒ | A statement on whether measurements were taken from distinct samples or whether the same sample was measured repeatedly |
| ☐ | ☒ | The statistical test(s) used AND whether they are one- or two-sided *Only common tests should be described solely by name; describe more complex techniques in the Methods section.* |
| ☒ | ☐ | A description of all covariates tested |
| ☒ | ☐ | A description of any assumptions or corrections, such as tests of normality and adjustment for multiple comparisons |
| ☐ | ☒ | A full description of the statistical parameters including central tendency (e.g. means) or other basic estimates (e.g. regression coefficient) AND variation (e.g. standard deviation) or associated estimates of uncertainty (e.g. confidence intervals) |
| ☐ | ☒ | For null hypothesis testing, the test statistic (e.g. *F*, *t*, *r*) with confidence intervals, effect sizes, degrees of freedom and *P* value noted *Give P values as exact values whenever suitable.* |
| ☒ | ☐ | For Bayesian analysis, information on the choice of priors and Markov chain Monte Carlo settings |
| ☒ | ☐ | For hierarchical and complex designs, identification of the appropriate level for tests and full reporting of outcomes |
| ☐ | ☒ | Estimates of effect sizes (e.g. Cohen's *d*, Pearson's *r*), indicating how they were calculated |

*Our web collection on statistics for biologists contains articles on many of the points above.*

## Software and code

Policy information about availability of computer code

| Data collection | Flow cytometry acquisitions were performed using BD FACSDiva Software Version 8.0.1 Microscopy acquisitions were performed using Metamorph 7.10.1 software (Molecular Devices, USA) |
|---|---|
| Data analysis | Image J software V2.1.0/1.53c was used to analyze most of the data. Custom made plugins were used to quantify DNA damage in cells and in tissues. After manual segmentation of the nuclei, a thresholding operation was used to determine the percentage of gamma H2Av positive pixels (coverage) and their average intensity in a single z plane in the center of the nucleus. Coverage and intensity were multiplied to obtain the gamma H2Av. For human cells gamma H2AX signals were measured using z-projection stacks after thresholding. Both FI and the percentage of nuclear coverage was obtained for each nucleus. Gamma H2AX index was obtained multiplying FI by the coverage. Statistical tests were performed using GraphPad Prism version 7.00 for Mac, GraphPad Software. Quantitative 4D live imaging of endogenous DNA replication, 3D reconstruction and analysis were done using Imaris Software v.9.6.0. Flow cytometry data were analyzed using FlowJo software 10.6.0 |

For manuscripts utilizing custom algorithms or software that are central to the research but not yet described in published literature, software must be made available to editors and reviewers. We strongly encourage code deposition in a community repository (e.g. GitHub). See the Nature Portfolio guidelines for submitting code & software for further information.

## Data

Policy information about availability of data

All manuscripts must include a data availability statement. This statement should provide the following information, where applicable:

- Accession codes, unique identifiers, or web links for publicly available datasets
- A description of any restrictions on data availability
- For clinical datasets or third party data, please ensure that the statement adheres to our policy

The datasets generated during and/or analysed during the current study are available from the corresponding author.

# Field-specific reporting

Please select the one below that is the best fit for your research. If you are not sure, read the appropriate sections before making your selection.

☒ Life sciences  ☐ Behavioural & social sciences  ☐ Ecological, evolutionary & environmental sciences

For a reference copy of the document with all sections, see nature.com/documents/nr-reporting-summary-flat.pdf

# Life sciences study design

All studies must disclose on these points even when the disclosure is negative.

| | |
|---|---|
| Sample size | At least 100 interphase cells were analyzed to determine DNA damage levels. The size of the sample was chosen to offer sufficient statistical power. |
| Data exclusions | We did not exclude any data. |
| Replication | All experiments were considered as replicates. |
| Randomization | Randomization was not relevant in this study. |
| Blinding | We tried to analyze DNA damage and quantify the phenotypes and behaviors described in this article in a blind manner initially. However, this was not possible as tetraploid cells can be easily distinguished from diploid cells. In any case, in terms of immunostaining experiments, the distinction between both cell types was solely based on the characteristics mentioned in the paper- cell and nuclear size and centrosome number. |

# Reporting for specific materials, systems and methods

We require information from authors about some types of materials, experimental systems and methods used in many studies. Here, indicate whether each material, system or method listed is relevant to your study. If you are not sure if a list item applies to your research, read the appropriate section before selecting a response.

### Materials & experimental systems

| n/a | Involved in the study |
|---|---|
| ☐ | ☒ Antibodies |
| ☐ | ☒ Eukaryotic cell lines |
| ☒ | ☐ Palaeontology and archaeology |
| ☐ | ☒ Animals and other organisms |
| ☒ | ☐ Human research participants |
| ☒ | ☐ Clinical data |
| ☒ | ☐ Dual use research of concern |

### Methods

| n/a | Involved in the study |
|---|---|
| ☒ | ☐ ChIP-seq |
| ☐ | ☒ Flow cytometry |
| ☒ | ☐ MRI-based neuroimaging |

## Antibodies

| | |
|---|---|
| Antibodies used | For Immunofluorescence:<br>Primary and secondary antibodies were used at the following concentrations: Guinea pig anti CEP192 antibody (1/500; Basto lab)60, rabbit anti-beta catenin (1/250; C2206 from Sigma-Aldrich, RRID:AB_476831), mouse anti-gamma H2A.X phospho S139 (1/1000; ab22551 from Abcam, RRID:AB_447150), mouse anti-XRCC1 (1/500; ab1838 from Abcam, RRID:AB_302636), rabbit anti-Rad51 (1/500; ab133534 from Abcam, RRID:AB_2722613), mouse anti-KU80 (1/200; MA5-12933 from ThermoFisher, RRID:AB_10983840), rabbit anti-FANCD2 (1/150; NB100-182SS from Novusbio, RRID:AB_1108397), mouse anti-53BP1 (1/250; MAB3802 from Millipore, RRID:AB_2206767), rabbit anti-ΞH2Av (1/500; 600-401-914 from Rockland; RRID: AB_11183655), Alexa Fluor® 647 Phalloidin (1/250; A22287 from ThermoFisher Scientific, RRID:AB_2620155), goat anti-Rabbit IgG (H+L) Highly Cross-Adsorbed Secondary Antibody, Alexa Fluor 647 (1/250; A21245 from ThermoFisher, RRID:AB_2535813), Goat anti-Guinea Pig IgG (H+L) Highly Cross-Adsorbed |

Secondary Antibody, Alexa Fluor 488 (1/250; A11073 from ThermoFisher, RRID:AB_253411), Goat anti-Mouse IgG (H+L) Cross-Adsorbed Secondary Antibody, Alexa Fluor 546 (1/250, A11003 from ThermoFisher, RRID:AB_2534071), Goat anti-Rabbit IgG (H+L) Highly Cross-Adsorbed Secondary Antibody, Alexa Fluor 546 (1/250; A-11035 from Thermo Fisher Scientific, RRID:AB_2534093).

For Western Blot:
- Primary and secondary antibodies were used at the following concentrations
Mouse anti-Etubulin (1/5000; T9026 from Sigma, RRID:AB_477593), mouse anti- CDC45 (1/100; sc-55569 from Santa Cruz Biotechnology, RRID:AB_831146), rabbit anti-PCNA (1/500; sc56 from Santa Cruz, RRID:AB_628110), rabbit anti-Actin (1/2000; A5060 from Sigma-Aldrich, RRID:AB_476738), mouse anti-H2B (1/1000; sc-515808 from Santa Cruz Biotechnology), mouse anti-ORC1 (1/100; sc-398734 from Santa Cruz Biotechnology), mouse anti-MCM2 (1/500; 610701 from BD Biosciences, RRID:AB_398024), mouse anti-E2F1 (1/2000; sc251 from Santa Cruz, RRID:AB_627476), mouse anti-CDC6 (1/500; sc-9964 from Santa Cruz, RRID:AB_627236), rabbit anti-CDT1 (1/500; 8064S from Cell Signaling, RRID:AB_10896851), rabbit anti-Treslin (1/500; A303-472A from Betyl, RRID:AB_10953949), Goat anti-Rabbit IgG (H+L) Cross-Adsorbed Secondary Antibody, HRP (1/2500; G21234 from ThermoFisher, RRID:AB_2536530), Peroxidase AffiniPure Goat Anti-Mouse IgG (H+L) (1/2500; 115-035-003 from Jackson ImmunoResearch, RRID:AB_10015289).

For DNA combing:
Antibodies were used at the following concentrations:
Rabbit anti ssDNA (1/5; 18731 from IBL International, RRID:AB_494649), Rat anti CldU (1/10; Ab6326 from Abcam, RRID:AB_2313786), Mouse anti IdU (1/10; 555627 from BD Biosciences, RRID:AB_10015222), mouse Alexa Fluor 647 Donkey (1/25; JIM-715-605-151 from Biozol), Rat Alexa Fluor 594 Donkey (1/25; JIM-712-585-153 from Biozol), Rabbit Brilliant Violet 480 Donkey (1/25; 711-685-152 from Jakcson Immuno Research, RRID:AB_2651109).

| | |
|---|---|
| Validation | Commercial antibodies were initially considered using the website information and by testing conditions that should increase or decrease the signals expected. For example- anti-gamma H2A.X phospho S139 signals should be increased upon the generation of DNA damage with external agents such as Aphidicoline, which we used. The single non- commercial antibody used was CEP192 antibody (1/500; Basto lab) (Vargas et al., 2019, Current Biology, PMID: 31495584; previously characterised with CEP192 depletion and by western blot on the size of the expected band). All secondary antibodies have been tested on multiple occasions in different projects from the lab. Different combinations of secondary antibodies have already been tested. |

# Eukaryotic cell lines

Policy information about cell lines

| | |
|---|---|
| Cell line source(s) | hTERT RPE-1 cells (ATCC Cat# CRL-4000, RRID:CVCL_4388; HEK293 cells (ATCC Cat# CRL-1573, RRID:CVCL_0045); BJ cells (ATCC Cat# CRL-4001, RRID:CVCL_6573) and HCT116 cells (ATCC Cat# CCL-247, RRID:CVCL_0291) |
| Authentication | Cells were authenticated using the Institut Curie genotype validation. |
| Mycoplasma contamination | All cells were routinely tested for Mycoplasm contamination. All cells used in this paper were mycoplasm-free. |
| Commonly misidentified lines (See ICLAC register) | We did not use any mis-identified cell line. |

# Animals and other organisms

Policy information about studies involving animals; ARRIVE guidelines recommended for reporting animal research

| | |
|---|---|
| Laboratory animals | Drosophila melanogaster- w background. <br> Mutants analyzed were L3 male with matched controls. <br> UAS E2F1 OE: M{UAS-E2f1.ORF}ZH-86Fb <br> sqh mutant: y w sqh1/FM7 <br> sqh mutant + wrnGal4: y w sqh1/FM7;WornGal4/Cyo <br> PAV RNAi: y1 v1; P{TRiP.HMJ02232}attP40 |
| Wild animals | This study did not involved wild animals. |
| Field-collected samples | This study did not involved samples collected in the field. |
| Ethics oversight | Ethical approval is not required for work with Drosophila melanogaster (invertebrate animal- fruit fly). European research guidelines in terms of disposal and transgenic reporting were followed. |

Note that full information on the approval of the study protocol must also be provided in the manuscript.

# Flow Cytometry

## Plots

Confirm that:

☒ The axis labels state the marker and fluorochrome used (e.g. CD4-FITC).

☒ The axis scales are clearly visible. Include numbers along axes only for bottom left plot of group (a 'group' is an analysis of identical markers).

☒ All plots are contour plots with outliers or pseudocolor plots.

☒ A numerical value for number of cells or percentage (with statistics) is provided.

## Methodology

| | |
|---|---|
| Sample preparation | A mix of diploid and tetraploid cells (see "generation of tetraploid cells" section) were incubated with 2μg/ml Hoescht (94403 from Sigma Aldrich) for 1 hour at 37°C, 5% CO2. Then, a single cell suspension was generated. Cells were washed using PBS 1X, the supernatant was removed and cells were resuspended in cold cell culture medium at 1x107 cell per ml and kept at 4° C during all the experiment. FACS sorting was performed using Sony SH800 FACS (BD FACSDiva Software Version 8.0.1). |
| Instrument | Sony SH800 and BD LSRII |
| Software | BD FACSDiva Software Version 8.0.1 |
| Cell population abundance | Post-sort analysis was performed to determine the purity of the sorted populations (see extended data Fig 8d-e) |
| Gating strategy | Compensation was performed using the appropriate negative control samples. Experimental samples were then recorded and sorted using gating tools to select the populations of interest. RFP+ / GFP- negative cells (G1 cells) were first selected. Then, in this population, DNA content was used to segregate diploid (2n) and tetraploid (4n) G1 cells. Once gates have been determined, diploid and tetraploid G1 cells were sorted into external collection tubes. |

☒ Tick this box to confirm that a figure exemplifying the gating strategy is provided in the Supplementary Information.

