## [Peer Review File · Nature]

Manuscript Title: Genetic instability from a single S-phase after whole genome duplication

Reviewer Comments & Author Rebuttals

Reviewer Reports on the Initial Version:

Referees' comments:

Referee #1 (Remarks to the Author):

In this manuscript, Gemble et al. analyzed the first cell cycle following whole-genome duplication (WGD) in cell lines in culture. The authors found high levels of DNA damage accumulating in the first interphase following WGD. They then showed that this DNA damage is (at least partly) dependent on DNA replication, and that tetraploid cells fail to properly scale the number of active replication forks and replication timing. General characterization of protein extracts from the cells demonstrated that the tetraploid cells enter the S-phase with insufficient amounts of DNA replication factors. Both the lengthening of G1 and overexpression of E2F1 significantly reduced DNA damage following WGD. The main results were reproduced in an in vivo system of fly neural stem cells.

Overall, this is an interesting study that addresses an important question, namely the immediate consequence of WGD. The experiments are technically-challenging yet the findings seem robust, appropriate controls are in place, and data analysis is also sound. However, a few improvements and additions should be made in order to examine how broadly applicable the study's findings actually are (i.e., beyond RPE1 cells cultured in 2D).

Major comments:

- 1) The findings in Fig. 1 are convincingly repeated in BJ and HCT116 cells, demonstrating that they are not specific to RPE1. However, the majority of the study was performed using RPE1 cells as a sole model system. These additional analyses (especially those shown in Fig. 2) should be repeated with the BJ and HCT116 cell lines, to ensure that the proposed model indeed goes beyond RPE1. This is of particular importance given that DNA combing was done in HCT116 cells and not in RPE1 due to technical challenges.
- 2) The experiments were performed in 2D cell lines, which may affect the ability of the cells to scale size, DNA content and protein production. Can the authors perform key experiments in a 3D system as well? This may be more physiologically relevant.
- 3) The focus of the study is on the first S-phase following WGD. However, what do the authors expect to happen in the following cell cycle? Would adaptation (referred to in the Discussion) already occur? An analysis of the 2nd and 3rd S-phases will be highly interesting.
- 4) DNA damage induction should be assessed using additional approaches (on top of γ H2AX)
- 5) The authors demonstrate some in vivo relevance using their system of drosophila neural stem cells. Arguably, cancer is a much more relevant system, as tetraploid tumors have undergone unscheduled WGD at some point during their life history, and were able to cope with the consequences. The WGD status of tumors has been determined in many cancer genomic studies, including TCGA (see Taylor et al. Cancer Cell 2018 for example). Do aneuploid tumors exhibit higher

levels of DNA damage? Or elevated gene expression signatures associated with DDR? This would be a key prediction of the findings, despite the presumed adaptation of the cancer cells. If adaptation prevents the detection of ongoing DNA damage in advanced tumors, then the genetic lesions that arose during the first cell cycle following WGD should still be evident in the $\sim 4N$ tumors, when compared to $\sim 2N$ tumors. Is there any evidence for that?

6) "We reasoned that increased levels of E2F1 might override the G1 lengthening defect" – doesn't E2F1 overexpression mimic the G1 lengthening rescue effect, rather than override it? This sentence should be clarified.

7) The model in Fig. 4H is somewhat over-complicated, and requires some more work

Minor comments:

1) A few grammatical errors exist throughout the text

2) In Fig. 2D, the comparison of interest is between the tetraploid cells with and without the APH.

3) Fig. 2J right plot – is this difference significant? If not, this should be mentioned in the text.

Referee #2 (Remarks to the Author):

Mechanisms of genetic instability in a single S-phase following whole genome doubling –

In this manuscript by Renata Basto and colleagues the authors describe their new observations on how tetraploidization induces genome instability. Specifically, the authors focus on the first cell cycle after tetraploidization, which they experimentally induced by mitotic slippage, cytokinesis inhibition or induction of endo-reduplication. Specifically, they find a strong induction of DNA damage in the first S phase following tetraploidization that coincides with abnormal DNA replication. They correlate these findings with a disproportionately short G1 phase that does give a sufficient rise in replication proteins. The authors therefore conclude that cells cannot sufficiently prepare for replication of a tetraploid chromosome set. Consistently, experimentally lengthening G1 or overexpressing E2F1 partly suppresses the DNA damage phenotype.

Overall, this paper describes intriguing insights into the relation between chromosomal instability and genomic instability. Some parts of the paper are however premature, so that I can only support publication once the following points have been addressed.

(1) The paper investigates the immediate effects of tetraploidization. To induce tetraploidization the authors need to substantially intervene with cellular physiology, cell cycle regulation in particular. This opens the question in how far the observed phenotypes are direct or indirect consequences of the experimental manipulation. I think the authors are fully aware of this and therefore start their paper with several systems for tetraploidization. Unfortunately, not all systems are utilized equally during the course of the study. In particular, I would urge the authors to use to as well use their endo-reduplication system for the key experiments of the paper.

(2) DNA replication is not only the key of event of S phase, it starts much earlier, with pre-RC's assembling as early as late M phase, i.e. when tetraploidization is induced. To show that pre-RCs form nonetheless normally would seem like a necessary control for the authors' model.

(3) Throughout the text the author's very much stress the importance of the first cell cycle/S phase. While this makes for a good selling point, I wonder what is the actual basis for highlighting the first cell cycle that much. The authors should carefully compare differences between first and second (or later) cell cycles.

(4) Much of the paper relies on the formation of gammaH2AX foci as a experimental readout. The authors conclude that the presence of a gammaH2AX focus is equivalent to the presence of a DNA damage. Is this actually true? Also stalled replication forks can yield a gammaH2AX signal. How about a more direct read-out of DNA breaks (e.g. comet assay)? Also, digging into the exact nature of the DNA damage signal might be informative!

(5) The suppressive effect of lengthening G1 is interesting. Unfortunately, the effect on expression levels of some (few) replication factors is less convincing. Analysis of proteomes of tetraploid cells by quantitative mass spectrometry will give a much clearer picture.

Referee #3 (Remarks to the Author):

Mechanisms of genetic instability in a single S-phase following whole genome doubling

Nature Manuscript 2021-07-11495

Gemble et al. have elegantly demonstrated the cause of the high levels of DNA damage in the first S phase after unscheduled tetraploidy and its impact on genome instability. The key observations were that the levels of DNA damage were reduced when DNA replication fork progression was inhibited with low doses of aphidicolin and strikingly a lack of scaling up of active replication sites was observed. This elevated damage could be relieved by lengthening G1 allowing protein mass scaling up to be achieved. The study was performed meticulously throughout, however due to the somewhat unexpected findings, some additional experiments below would need to be performed for the manuscript to be acceptable for publication in Nature.

Major comments

1. Firstly, to fully understand the nature of the increased DNA damage in the unscheduled tetraploid cells, it would be advisable to establish whether in addition to H2aX, other DNA damage markers are observed with all three strategies applied. In light of the elevated damage observed in response to endo replication, fancd2 foci formation should at least be quantified in all panels of Figure 1 to determine whether all 3 methods generate replication stress or is it just mitotic slippage. As an additional control to determine the generality of these findings, additional stimuli should be applied to further explore the differential susceptibility of diploid and tetraploid cells.

2. The DNA combing assays were restricted to HCT116 cells following cytokinesis inhibition whereas the majority of the manuscript was performed using mitotic slippage in RPE cells. Although the technical challenges of the combing assay are appreciated, the surprising result of accelerated fork progression should be confirmed in at least one additional p53 wildtype cell line.

3. The main conclusion is that the cells which enter unscheduled tetraploidy have difficulty with DNA

replication due to failure to scale up the replication factors required to deal with their doubled genome. As a result, DNA damage is observed in the first interphase after tetraploidy. The authors are able to rescue the defect by lengthening G1. Conceptually it is still slightly unclear when this defect is conceived. Can the authors deduce at what stage after induction of tetraploidy, the replication factors are not transcribed or not translated? Furthermore, to more convincingly demonstrate the consequence of this failed scaling up of S phase replication factors, it would be more informative to assess the chromatin association of these proteins by western blot rather than just measuring the total levels present at G1/S as this binding could impact the observed fork stability.

4. On a similar note, the authors conclude that E2F1 overexpression can override the defect in G1 lengthening to prevent damage in the first S phase. However, their E2F1 overexpression was performed in diploid cells prior to unscheduled tetraploidy, so although it's an interesting experiment, it doesn't fully address their hypothesis. Furthermore, the rescued DNA damage observed in the tetraploid neuroblasts following E2F1 overexpression would be more compelling if the interphase damage was scored in S phase specifically.

5. In figure 2, the suggestion of delayed replication timing in tetraploid cells compared to diploid cells is a key finding. This is in contrast to longer term cultures which have actually identified scaling up in tetraploid lines. Although perhaps challenging, to solidify this finding, it would be more acceptable to directly measure replication timing more accurately as opposed to inferring it from patterns of PCNA.

6. The accelerated fork progression rate following unscheduled tetraploidy is intriguing and the reasons for this should be explored in more detail. The authors should disentangle the role played by replication stress and identify which phenotypes can be rescued with nucleoside treatment. The authors should also perform fork symmetry assays to identify whether mechanistically, the accelerated fork progression rates in tetraploid cells induced by mitotic slippage is caused by an actual defect in fork stalling similar to what has been reported for PARP inhibition.

7. The elevated level of copy number variability in tetraploid cells is quite compelling and likely underlies the heterogeneity in G2/M tetraploid cells. Could the authors estimate/calculate the actual proportion of the genome that is subject to these aberrations to get a better appreciation of the severity of the phenotype?

Minor comments

1. Several typos and grammatical errors were noted throughout the manuscript
2. Extended data Figure 1K – x-axis labelling is incorrect

Referee #4 (Remarks to the Author):

This manuscript from Gemble et al, addresses a very interesting question regarding the mechanism of genome instability (GIN) following a whole genome duplication event. This is an exciting topic for several reasons, primarily because it is known that genome instability following WGDs is responsible for the aneuploidy that is observed in cancer cells and GIN following WGDs has broad implication for evolution. To tackle this question, the authors artificially induce tetraploidization of RPE cells through multiple mechanisms. And, through a number of assays, determine that DNA damage and GIN in these cells occurs during the first S phase following tetraploidization. This is true across multiple cell types and regardless of the mechanism used to generate the tetraploid cells. They found that tetraploid cells have rapid fork progression, but forks are generally unstable in the first S phase following tetraploidization. They then attempt to determine the underlying molecular mechanism responsible for GIN in these tetraploid cells. Here, the authors found that these artificially induced polyploid cells don't scale up protein production in G1 phase, which results in less replication factors per unit chromatin. Therefore cells have a hard time executing a faithful replication cycle. They can partially rescue the GIN in tetraploid cells by extending G1 prior to the first S phase, which supports their proposed model of GIN.

While I think this is really elegant system to study the consequences of WGDs on GIN, I think there are some concerns about the molecular mechanism underlying GIN in the tetraploid cells. Assuming the data in the paper to be accurate (concerns outlined below), it appears that the mechanism proposed only partially explains the GIN. For example, extending G1 or over expressing E2F1 reduces the DNA damage phenotype by half. While this interesting, clearly there must be something else contributing to the GIN in the first S phase following tetraploidization. In addition to this, the proposed mechanism of GIN in these tetraploid cells remains unclear. These cells don't have enough ORC and MCMs, which suggest a problem with helicase loading. However, there is no change in inter-origin distance when comparing tetraploid and diploid cells. The authors are dancing around this in the discussion ln336-337 suggesting that decreased MCM levels could explain the increased rate of fork progress. This is well established that when there are fewer forks competing for limiting resources they can go faster. However, one would expect an increase in the inter-origin distance. The major questions remains, if you can restore protein levels in G1 by extending G1 or E2F1OE, what is driving the remaining (and substantial) GIN in the tetraploid cells?

What might be the most major concern I have with this paper, however, is that the majority of the statistical comparisons were done incorrectly. Therefore, until the correct tests are used, any conclusions drawn on these incorrect tests are not supported by the data. Rather than describing the statistical test in figure legends, the authors devote a section in the methods to say they used Student t-test for all comparisons. Given that most of the analysis focus on more than two groups, they should have used an ANOVA test. While I think that at least some of the claims of significance will stand when the correct tests are performed, I fear that many of the claims of significance will change. I would suggest that the authors think more carefully about the correct statistical analysis for each data set/comparison. It is unlikely one test will be used throughout an entire study. Also, error bars go both above and below the bar graph. Bar graphs are no longer the standard when presenting data like this. At the very least, the data points should be included in the bar graph so the n is clear for each experiment and readers can get a sense of the distribution of the data.

Significant concerns:

1) Figure 2J: How do the authors define 'unstable forks'? It doesn't appear to be described in the methods or legend. I'm assuming they measuring some aspect of asymmetry, but this should be made clear. Also, I'm suspicious that this result is statistically significant since no statistical test was used to compare the data sets. Until that is done, they can't claim that forks go faster in the tetraploid cells, which seems to be important for their ideas of GIN following tetraploidization.

2) For Figure 3D: What would be the most useful comparison is the amount of chromatin-bound ORC, MCM and PCNA, not the amount of total ORC, MCM, etc. per cell. For example, you could have a slight decrease in the total amount of a given protein without affecting the amount loaded on chromatin, which I would argue is more relevant for this study. Additionally, does O/E of E2F1 result an increase in the amount of ORC/MCM on chromatin?

3) Extended Figure 3H: Given the increase in gammaH2Ax staining throughout the nucleus, is the increase in colocalization with PCNA above what you would expect by chance? The same would be true for Extended 4 figures. However, admittedly the colocalization between Rad51 and gH2Ax is much more convincing.

4) Ln 180-183: While I agree that the signal lingers for a longer period of time, the increase in the number of active sites being more gradual and the dissociation of PCNA happening more progressively would require the authors to compare the slopes of these curves. By eye, I don't see a huge difference that would support these claims. These are also hard claims to validate based on four cells.

5) It would be helpful to demonstrate that treating cells with a low level of CDK4/6 and CDK2 inhibitors causes a lengthening of G1. Right now, the authors are assuming that G1 is lengthened. Also, it would be good to know how much G1 lengthens and generally show more cell cycle profiles to confirm arrests, etc. throughout the paper.

6) I don't understand the logic of Rb over expression in Drosophila. While this would theoretically extend G1, it would do so by antagonizing E2F1 and thereby give the opposite phenotype than E2F1OE. In the reference cited, I don't believe there is direct evidence of G1 lengthening in RbO/E

7) One paradox I see with this work is that dampening of the E2F system in polyploid cells. While trying to compare between naturally polyploid cells and artificially induced tetraploid cells, the assumption is that artificially-induced cells don't have enough time to scale up E2F-dependent expression in G1, but natural polyploid cells do. However, at least in Drosophila, It seems natural instances of polyploidy have adapted to reduced E2F capacity. (Maqbool et al., 2010 J. of Cell Science)

Minor concerns:

Ln 88: gammaH2Ax is a well-known marker of DSBs and stalled forks (Ward and Chen JBC 2001). Therefore, gammaH2Ax cannot be assumed to be a marker of only DSBs and doesn't provide evidence that the tetraploid cells have increased levels of DSBs. In fact, based on the data presented

in this work (Extended Figure 1M for example), it seems likely that there could be a significant number of stalled forks associated with tetraploid cells.

Ln 179 decreased -> decrease

Extended figure 3B legend says 53BP1 is in green, however in the figure it is red.

Extended 5(A) legend: several -> Four

Ln 239 - enables us to sort

Ln 262 - dependent of cell mass -> dependent on cell mass

Ln 313: They don't have 2000 chromosomes. They have a ploidy or C value of 2000, but since they are polytene are only 1n. Diploids are paired in somatic cells so they have 4 chromosomes (2n) and are 2C.

Ln 320: Is this striking given that it is the same result you showed in Figure 3?

Ln 454 - cell -> cells

Ln 586 - define FI (assume it's fluorescent intensity)

Ln 655 - 1/2 500 -> 1/2,500

Ln 697 - Labelling -> labeling

Ln 724 - define NEBD

Extended data figure 2D and E: It is not clear what method was used to generate the tetraploid cells. This should be made clear in the text and/or figure legend.

The pattern of dots is the same for the diploid controls in Extended 3E and F. While I think it is okay to use the same reference control the authors should make this clear in the legend.

In Ext. Fig 4C and 4F: Why is there such an increase in Rad51 and FANCD2 when comparing the diploid cells +/- realize into S phase? Is the arrest causing damage itself that must be resolved in S phase?

Figure 4D: Why don't you see DNA damage in the salivary gland chromosomes? These cells are known to have constitutive DNA damage due to under replication of pericentric heterochromatin and other sites throughout euchromatin. Not seeing this damage raises concerns about the sensitivity of the antibody staining.

In 4G I think the CF +/- signs correspond to each of the three D or Ps. It would be easier to follow if all the spaces were filled. Took this reviewer a few minutes to figure out why cells are P but don't have a + for the CF

Author Rebuttals to Initial Comments:

We think that we can address most comments with additional experiments that will be initiated shortly. But in order to ensure that we fully understand and interpret correctly these comments, I am listing a few points. It would be great if the reviewers could let us know if our interpretation is meeting your expectations.

Reviewer#1

Point 5- "The authors demonstrate some in vivo relevance using their system of drosophila neural stem cells. Arguably, cancer is a much more relevant system, as tetraploid tumors have undergone unscheduled WGD at some point during their life history, and were able to cope with the consequences. The WGD status of tumors has been determined in many cancer genomic studies, including TCGA (see Taylor et al. Cancer Cell 2018 for example). Do aneuploid tumors exhibit higher levels of DNA damage? Or elevated gene expression signatures associated with DDR? This would be a key prediction of the findings, despite the presumed adaptation of the cancer cells. If adaptation prevents the detection of ongoing DNA damage in advanced tumors, then the genetic lesions that arose during the first cell cycle following WGD should still be evident in the $\sim 4N$ tumors, when compared to $\sim 2N$ tumors. Is there any evidence for that?"

We wonder if the underline text concerns polyploid tumors (not aneuploid as it is written)? And more importantly, whether this reviewer is asking for bioinformatics analysis of $4N$ tumors to reveal signatures of DNA damage response (DDR) or more genetic instability as shown in the literature?

Reviewer#2

Point 4- "Much of the paper relies on the formation of gammaH2AX foci as a experimental readout. The authors conclude that the presence of a gammaH2AX focus is equivalent to the presence of a DNA damage. Is this actually true? Also stalled replication forks can yield a

gammaH2AX signal. How about a more direct read-out of DNA breaks (e.g. comet assay)? Also, digging into the exact nature of the DNA damage signal might be informative!”

We understand this point and we will perform the comet assay experiments and we will also include additional DNA damage markers. What is less obvious to us is the underline sentence concerning the exact nature of the damage. Can the reviewer be more specific about what he/she is expecting? Or in other words, he/she is asking (just as the point below) to include more markers and the quantification of other markers?

Reviewer#3

Pont 1- “. Firstly, to fully understand the nature of the increased DNA damage in the unscheduled tetraploid cells, it would be advisable to establish whether in addition to H2aX, other DNA damage markers are observed with all three strategies applied. In light of the elevated damage observed in response to endo replication, fancd2 foci formation should at least be quantified in all panels of Figure 1 to determine whether all 3 methods generate replication stress or is it just mitotic slippage. As an additional control to determine the generality of these findings, additional stimuli should be applied to further explore the differential susceptibility of diploid and tetraploid cells.”

Here what we do not fully understand is the underlined – “additional stimuli”. Is the reviewer asking for the quantification of DNA damage using several markers in tetraploid cells induced in different ways? Or alternatively, a comparison of DNA damage levels generated by other agents in diploid cells.

Point 5. “ In figure 2, the suggestion of delayed replication timing in tetraploid cells compared to diploid cells is a key finding. This is in contrast to longer term cultures which have actually identified scaling up in tetraploid lines. Although perhaps challenging, to solidify this finding, it would be more acceptable to directly measure replication timing more accurately as opposed to inferring it from patterns of PCNA. ”

We wonder if the reviewer is asking us to measure replication timing using EdU pulses in diploid and tetraploid cells? Because we infer the replication timing not by the pattern of PCNA, but by the simple appearance and release of PCNA fluorescence signals on the chromatin (Extended Figure 3P). In other words, the replication timing is provided independently of the way PCNA behaves once on the chromatin.

Reviewer Reports on the First Revision:

Reviewer#1

Point 5- “The authors demonstrate some in vivo relevance using their system of drosophila neural stem cells. Arguably, cancer is a much more relevant system, as tetraploid tumors have undergone unscheduled WGD at some point during their life history, and were able to cope with the consequences. The WGD status of tumors has been determined in many cancer genomic studies, including TCGA (see Taylor et al. Cancer Cell 2018 for example). Do aneuploid tumors exhibit higher levels of DNA damage? Or elevated gene expression signatures associated with DDR? This would be a key prediction of the findings, despite the presumed adaptation of the cancer cells. If adaptation prevents the detection of ongoing DNA damage in advanced tumors, then the genetic lesions that arose during the first cell cycle following WGD should still be evident in the ~4N tumors, when compared to ~2N tumors. Is there any evidence for that?”

We wonder if the underline text concerns polyploid tumors (not aneuploid as it is written)? Yes

And more importantly, whether this reviewer is asking for bioinformatics analysis of 4N tumors to reveal signatures of DNA damage response (DDR) or more genetic instability as shown in the literature?

Yes, indeed. The genomic correlates of whole-genome doubling have been assessed (most notably by Bielski et al. Nat Genet 2018; and Quinton et al. Nature 2021), and the link between WGD and ongoing genomic instability is well established. Nonetheless, as far as I know no published paper includes a comprehensive assessment of the potential link between 4N tumors and the genetic and transcriptional signatures of DNA damage response (and the various ‘flavours’ of GIN). This may be a nice addition to the current manuscript, to support the in vivo link between polyploidy, DNA damage and GIN.

Reviewer#2

Sorry, I admit this sentence was rather colloquial. Yes, comet assay and quantification of further DNA damage markers will be fine.

Reviewer#3

Point 1-

The reviewer is asking for the former –quantification of DNA damage using several markers in tetraploid cells induced in different ways.

Point 5.

The reviewer suggests that the term replication kinetics (or similar) is used in the manuscript text to avoid any misunderstanding. Replication timing assays give more temporal information relating to when in S phase specific loci are duplicated and is often measured using more sensitive protocols. Some re-wording would be sufficient to address this point.

Author Rebuttals to First Revision:

Response to reviewers

Manuscript: 2021-07-11495A-Z from Gemble et al: Mechanisms of genetic instability in a single S-phase following whole genome doubling

Referee #1 (Remarks to the Author):

In this manuscript, Gemble et al. analyzed the first cell cycle following whole-genome duplication (WGD) in cell lines in culture. The authors found high levels of DNA damage accumulating in the first interphase following WGD. They then showed that this DNA damage is (at least partly) dependent on DNA replication, and that tetraploid cells fail to properly scale the number of active replication forks and replication timing. General characterization of protein extracts from the cells demonstrated that the tetraploid cells enter the S-phase with insufficient amounts of DNA replication factors. Both the lengthening of G1 and overexpression of E2F1 significantly reduced DNA damage following WGD. The main results were reproduced in an *in vivo* system of fly neural stem cells.

Overall, this is an interesting study that addresses an important question, namely the immediate consequence of WGD. The experiments are technically-challenging yet the findings seem robust, appropriate controls are in place, and data analysis is also sound. However, a few improvements and additions should be made in order to examine how broadly applicable the study's findings actually are (i.e., beyond RPE1 cells cultured in 2D).

Major comments:

1) The findings in Fig. 1 are convincingly repeated in BJ and HCT116 cells, demonstrating that they are not specific to RPE1. However, the majority of the study was performed using RPE1 cells as a sole model system. These additional analyses (especially those shown in Fig. 2) should be repeated with the BJ and HCT116 cell lines, to ensure that the proposed model indeed goes beyond RPE1. This is of particular importance given that DNA combing was done in HCT116 cells and not in RPE1 due to technical challenges.

We understand the reviewer's concerns about the importance of showing reproducibility among different cell lines. We have thus performed several experiments in either BJ or HCT116 cells or even in both cell lines. A list of the additional experiments can be found below:

1) *The different means to generate tetraploidy - mitotic slippage (MS), cytokinesis failure (CF) and endoreplication (ENR) - were induced in RPE-1, BJ and HCT116 cell lines- Figure 1 and Extended data Fig.2g and h.*

2) *Comet assays in RPE-1 and BJ cells - Extended data Fig.2j and k.*

3) *DNA combing in RPE-1, BJ and HCT116 cells- Fig.2e-g and Extended data Fig. 6j-k.*

4) Single-cell DNA sequencing in RPE-1 and BJ cells- Fig.2h-i and Extended data Fig.7.

5) Rescue of DNA damage levels after APH treatment in RPE-1, BJ and HCT116 cells - Figure 2d and Extended data Fig. 5e and f.

6) 3D spheroids in RPE-1, BJ and HCT116 cells Extended data Fig.3a-d.

7) Rescue of DNA damage levels after G1 lengthening in RPE-1, BJ and HCT116 cells- Figure 3p and Extended data Fig. 9q and r.

8) Rescue of DNA damage levels after E2F over-expression in RPE-1, BJ and HCT116 cells- Figure 4b and Extended data Fig. 10b and c.

The use of additional cell lines confirmed the results initially obtained with RPE-1 cells.

2) The experiments were performed in 2D cell lines, which may affect the ability of the cells to scale size, DNA content and protein production. Can the authors perform key experiments in a 3D system as well? This may be more physiologically relevant.

To follow this reviewer's suggestion, we have generated spheroids of RPE-1, BJ and HCT116 cells. Importantly, in 3D, the levels of DNA damage in recently born tetraploid cells are quite high and show similar differences to controls when compared to the cell lines in 2D. We have included these new data in Extended Figure 3a-d.

3) The focus of the present study is on the first S-phase following WGD. However, what do the authors expect to happen in the following cell cycle? Would adaptation (referred to in the Discussion) already occur? An analysis of the 2nd and 3rd S-phases will be highly interesting.

There are several reasons why we decided to address specifically the first cycle. First, the large majority of previous publications on early consequences of WGD (even if not on the first cell cycle) focused on proliferation and viability potential of these cells¹⁻³ and so did not address the consequences in terms of genetic instability. Second, it has been convincingly shown that the large majority of cells after WGD undergo abnormal multipolar mitosis, which leads often to cell death or the generation of highly aneuploid cells. Indeed, several labs in the field have focused on the consequences of tetraploidy as a promoter of aneuploidy through mitotic errors and on the adaptation and evolution of tetraploid/aneuploid karyotypes^{1,2,4-7}. Third, no study so far has specifically addressed what happens mechanistically in the first S-phase of newly born tetraploid cells. To our knowledge this is the first study focusing on the first cell cycle and on the characterization of DNA replication in the first S-phase that follows tetraploidization. Hence a major focus on the first interphase.

We nevertheless understand this reviewer's concern. And to provide a comparison between the first and subsequent cell cycles is important. We have to take into consideration the following: in the first cell cycle following tetraploidization (independently of the way tetraploidization is generated), the large majority of cells continues to cycle and proliferate as described here (our results). This is not the case afterwards, where the proliferative capacity of tetraploid cells is highly decreased. We

know from data in the literature that p53 is involved in the arrest of tetraploid cells⁸. To be able to analyze the 2nd and 3rd cycles, we have depleted p53 using stable single hairpin (sh) RNAs. Indeed, in this condition, tetraploid RPE-1 cells now continue to proliferate over many cell cycles. We have analyzed the levels of DNA damage in the 2nd and 3rd cycles and found that they decrease, suggesting indeed a capacity for adaptation. We have included these data in Extended data Fig. 2l-o.

4) DNA damage induction should be assessed using additional approaches (on top of γ H2AX).

In the revised version of this manuscript we have included other markers such as 53BP1 and FANCD2 in tetraploid cells after MS, CF or EnR. This analysis shows that all these markers are increased in tetraploid cells independently of the means used to induce tetraploidy. Moreover, similarly to what we observed for γ H2AX, G1 lengthening prevents the accumulation of 53BP1 and FANCD2 foci in newly born tetraploid S-phase cells. This can be found in Figure 1i-l and Extended data Fig. 9l and m.

5) The authors demonstrate some in vivo relevance using their system of drosophila neural stem cells. Arguably, cancer is a much more relevant system, as tetraploid tumors have undergone unscheduled WGD at some point during their life history, and were able to cope with the consequences. The WGD status of tumors has been determined in many cancer genomic studies, including TCGA (see Taylor et al. Cancer Cell 2018 for example). Do “polyploid” tumors exhibit higher levels of DNA damage? Or elevated gene expression signatures associated with DDR? This would be a key prediction of the findings, despite the presumed adaptation of the cancer cells. If adaptation prevents the detection of ongoing DNA damage in advanced tumors, then the genetic lesions that arose during the first cell cycle following WGD should still be evident in the ~4N tumors, when compared to ~2N tumors. Is there any evidence for that?

To answer to this point, we have performed a bioinformatic analysis of TCGA ovarian cancer data. We chose a cohort comprising 482 samples containing near diploid (162 tumors) and near tetraploid (320) tumors. We analyzed transcriptomic signatures related with DNA damage response (DDR). Interestingly, out of 17 genes, 5 appeared upregulated in near tetraploid tumors, when compared to near diploid tumors. These results suggest that in near tetraploid tumors, certain DDR components are upregulated, which might be explained by high levels of genetic instability typical of ovarian tumors. These data are included in the Extended data Fig. 10h of the revised manuscript.

6) "We reasoned that increased levels of E2F1 might override the G1 lengthening defect" – doesn't E2F1 overexpression mimic the G1 lengthening rescue effect, rather than override it? This sentence should be clarified.

The reviewer is absolutely right. We have changed this sentence to “we reasoned that increased levels of E2F1 might compensate for defects in G1 length scaling up”.

7) The model in Fig. 4H is somewhat over-complicated, and requires some more work

We have attempted to simplify the model, we hope the reviewer finds it more straightforward.

Minor comments:

1) A few grammatical errors exist throughout the text

We thank the reviewer for pointing this out. We have corrected all the typos and grammatical errors.

2) In Fig. 2D, the comparison of interest is between the tetraploid cells with and without the APH.

This is exactly what we have done, but we think the reviewer's point comes from the fact that the line with the statistical text is not long enough to illustrate the two points being compared. We have corrected this.

3) Fig. 2J right plot – is this difference significant? If not, this should be mentioned in the text.

We have changed the graphs to express the percentage of fibers that fall into different categories of CldU/IdU ratios, which measures fork asymmetry. We have included the statistical analysis (Fig. 2f-g). We report fork asymmetry in both RPE-1 tetraploid cells induced through MS or EnR or in BJ tetraploid cells induced through EnR, suggesting the presence of collapsed or stalled forks. These results can explain the high levels of DNA damage observed in tetraploid cells.

Referee #2 (Remarks to the Author):

Mechanisms of genetic instability in a single S-phase following whole genome doubling

In this manuscript by Renata Basto and colleagues the authors describe their new observations on how tetraploidization induces genome instability. Specifically, the authors focus on the first cell cycle after tetraploidization, which they experimentally induced by mitotic slippage, cytokinesis inhibition or induction of endo-reduplication. Specifically, they find a strong induction of DNA damage in the first S phase following tetraploidization that coincides with abnormal DNA replication. They correlate these findings with a disproportionately short G1 phase that does not give a sufficient rise in replication proteins. The authors therefore conclude that cells cannot sufficiently prepare for replication of a tetraploid chromosome set. Consistently, experimentally lengthening G1 or overexpressing E2F1 partly suppresses the DNA damage phenotype.

Overall, this paper describes intriguing insights into the relation between chromosomal instability and genomic instability. Some parts of the paper are however premature, so that I can only support publication once the following points have been addressed.

(1) The paper investigates the immediate effects of tetraploidization. To induce tetraploidization the authors need to substantially intervene with cellular physiology, cell cycle regulation in particular. This opens the question in how far the observed phenotypes are direct or indirect consequences of the experimental manipulation. I think the authors are fully aware of this and therefore start their paper with several systems for tetraploidization. Unfortunately, not all systems are utilized equally during the course of the study. In particular, I would urge the authors to use to as well use their endo-reduplication system for the key experiments of the paper.

As requested by the reviewer, we have now included extra data performed in tetraploid cells induced through endoreplication (EnR). This includes:

- 1) Levels of additional DNA damage markers- Fig. 1i-l.*
- 2) Quantification of DNA damage levels along the first cell cycle- Extended Data Fig. 3f.*
- 3) Quantification of DNA damage levels in conditions with low APH concentration to inhibit DNA replication in RPE-1, BJ and HCT116 cells induced through EnR- Extended Data Fig. 5d-f.*
- 4) Lack of rescue of DNA damage levels in tetraploid cells treated with nucleosides- Extended Data Fig. 5j.*
- 5) Comet assays- Extended Data Fig. 2j and k.*
- 6) Rescue of DNA damage levels in tetraploid cells after G1 lengthening – Fig. 3q.*

7) Measures of relative RNA levels in tetraploid cells- Extended Data Fig. 8c.

8) Measures of G1 lengthening in RPE-1 and HCT116 cells- Figure 3i and Extended Data Fig. 8j.

9) Analysis of the 2nd and 3rd cell cycles after EnR- Extended Data Fig. 2o.

(2) DNA replication is not only the key of event of S phase, it starts much earlier, with pre-RC's assembling as early as late M phase, i.e. when tetraploidization is induced. To show that pre-RCs form nonetheless normally would seem like a necessary control for the authors' model.

We agree with the reviewer that it is important to characterize earlier steps of DNA replication in G1 to assess the recruitment and assembly of pre-replication complexes (pre-RCs). To answer to this question, we analyzed extracts from chromatin-bound fractions in diploid and tetraploid cells during G1 and probe for ORC1, MCM2 and CDT1, which are members of pre-RCs. Importantly, comparison between diploid and tetraploid showed a lack of scaling up. The data are included in Fig.3g and h and Extended data Fig. 8h. Importantly, we also probe the levels of these proteins after G1 lengthening and show that the chromatin-bound levels are now comparable- Fig.3n and o and Extended Data Fig. 9k.

(3) Throughout the text the author's very much stress the importance of the first cell cycle/S phase. While this makes for a good selling point, I wonder what is the actual basis for highlighting the first cell cycle that much. The authors should carefully compare differences between first and second (or later) cell cycles.

We stressed the importance of the first cell cycle for two reasons. The first one is that the immediate consequences of tetraploidy were not known before this study. Many studies have shown that WGDs are a hallmark of a high number of human cancers and it has been associated with genetic instability. However, whether this results from the accumulation of abnormal interphase and/or mitosis over several cell cycles or from a single cell cycle was not known. And we show here that within the first cell cycle these cells generate severe DNA damage due to defects in replication fork progression and highly abnormal karyotypes. The proliferation and adaptation related with evolved tetraploid karyotypes has been addressed previously by different labs such as Storchova, Ganem, Cimini or Swanton labs ^{1,2,4-7,9}. The second reason is related with mitosis and aneuploidy. As shown by different labs, cell division in the presence of extra centrosomes and extra chromosomes has the capacity to generate chromosome aneuploidies through either multipolar divisions or chromosome mis-segregation. Further, as shown by us recently ^{10,11}, cell cycle asynchrony can generate DNA damage at mitotic entry. So the analysis of later cell cycles will reveal consequences of DNA damage generated in the 1st cell cycle and the subsequent mitosis and interphase.

It is also important to mention that as tetraploid cells progress through successive cell cycles, p53 is activated in order to arrest these cells ⁸. This type of analysis has already been made by several other labs, and it is presumed that it will select for the 'winner' karyotypes that can bypass the p53 control.

To address this point, as other reviewers have also raised it, we have determined the levels of DNA damage in the first, second and third cell cycles in tetraploid cells induced by mitotic slippage or endoreplication, in p53 knockdown conditions. Importantly, we show that indeed cells adapt and decrease DNA damage levels. However, they still present higher DNA damage levels than controls. These data argue that analyzing the first cell cycle is indeed extremely important and delivers previously unknown information. We have included this analysis in Extended Figure 2l-o.

(4) Much of the paper relies on the formation of gammaH2AX foci as a experimental readout. The authors conclude that the presence of a gammaH2AX focus is equivalent to the presence of a DNA damage. Is this actually true? Also stalled replication forks can yield a gammaH2AX signal. How about a more direct read-out of DNA breaks (e.g. comet assay)? Also, digging into the exact nature of the DNA damage signal might be informative!

As suggested by the reviewer, we have performed comet assays in tetraploid RPE-1, BJ and HCT116 cells. We found that indeed tetraploid conditions showed an increase in comet tail moment, when compared to diploids in all the cell lines analyzed. Concerning other DNA damage markers, we have added the analysis and quantifications of 53BP1 and FANCD2 foci also for different means to generate tetraploidy with or without G1 lengthening. This new set of data can be found in Fig.1 i-l and Extended Data Fig. 2j and k and Extended data Fig. 9l and m.

(5) The suppressive effect of lengthening G1 is interesting. Unfortunately, the effect on expression levels of some (few) replication factors is less convincing. Analysis of proteomes of tetraploid cells by quantitative mass spectrometry will give a much clearer picture.

We agree with this reviewer: a global analysis of newly formed tetraploids will indeed be important. We certainly envisage to perform these experiments in the future, which will complement proteomic analysis already performed in some tetraploid cell lines and in yeast^{6,7,12} or transcriptomic analysis performed in cell lines¹³, but we think these are not reasonable during the revision of this paper. Nevertheless, to increase the number of replication factors analyzed we have also tested the levels of CDT1, CDC6, Treslin and E2F1. Additionally, we have analysed total protein fractions and nuclear fractions. Further, we have also analysed total RNAs levels from both diploid and tetraploid cells. Overall, we found that already at the expression level there is a lack of scaling up, which agrees with the experiments of mass calculation. These results are now presented in Fig. 3e-h and k-o and Extended data Fig. 8a-c.

Referee #3 (Remarks to the Author):

Mechanisms of genetic instability in a single S-phase following whole genome doubling

Nature Manuscript 2021-07-11495

Gemble et al. have elegantly demonstrated the cause of the high levels of DNA damage in the first S phase after unscheduled tetraploidy and its impact on genome instability. The key observations were that the levels of DNA damage were reduced when DNA replication fork progression was inhibited with low doses of aphidicolin and strikingly a lack of scaling up of active replication sites was observed. This elevated damage could be relieved by lengthening G1 allowing protein mass scaling up to be achieved. The study was performed meticulously throughout, however due to the somewhat unexpected findings, some additional experiments below would need to be performed for the manuscript to be acceptable for publication in Nature.

Major comments

1. Firstly, to fully understand the nature of the increased DNA damage in the unscheduled tetraploid cells, it would be advisable to establish whether in addition to H2aX, other DNA damage markers are observed with all three strategies applied. In light of the elevated damage observed in response to endo replication, fancd2 foci formation should at least be quantified in all panels of Figure 1 to determine whether all 3 methods generate replication stress or is it just mitotic slippage. As an additional control to determine the generality of these findings, additional stimuli should be applied to further explore the differential susceptibility of diploid and tetraploid cells.

As suggested by the reviewer, we have quantified the number of FANCD2 foci in RPE-1 tetraploid cells generated by mitotic slippage (MS), cytokinesis failure (CF) and endoreplication (EnR). We have also included other markers such as 53BP1 and performed comet assays in tetraploid RPE-1 and BJ cells induced through MS and EnR. We show that in all conditions, the levels of DNA damage are quite high in tetraploids when compared to diploids. Moreover, similarly to what we observed for γ H2AX, G1 lengthening prevents the accumulation of 53BP1 and FANCD2 foci in newly born tetraploid S-phase cells. Please see: Fig.1i-l, Extended data Fig. 2j and k and Extended data Fig. 9l and m.

2. The DNA combing assays were restricted to HCT116 cells following cytokinesis inhibition whereas the majority of the manuscript was performed using mitotic slippage in RPE cells. Although the technical challenges of the combing assay are appreciated, the surprising result of accelerated fork progression should be confirmed in at least one additional p53 wildtype cell line.

We have performed DNA combing assays in RPE-1 and BJ cells and in agreement with the data obtained with HCT116 cells, replication forks appeared to be accelerated in tetraploid cells when compared to controls. Importantly, this new set of results show that CF, MS or EnR behave in the same manner. In the new version of the manuscript, the data concerning DNA combing can be found in Fig. 2e-g and Extended data Fig.6j-

k.

3. The main conclusion is that the cells which enter unscheduled tetraploidy have difficulty with DNA replication due to failure to scale up the replication factors required to deal with their doubled genome. As a result, DNA damage is observed in the first interphase after tetraploidy. The authors are able to rescue the defect by lengthening G1. Conceptually it is still slightly unclear when this defect is conceived. Can the authors deduce at what stage after induction of tetraploidy, the replication factors are not transcribed or not translated? Furthermore, to more convincingly demonstrate the consequence of this failed scaling up of S phase replication factors, it would be more informative to assess the chromatin association of these proteins by western blot rather than just measuring the total levels present at G1/S as this binding could impact the observed fork stability.

We fully agree with this assessment and that it is key to determine if there are defects in the chromatin-bound fraction of replication factors. Indeed, a decrease in total amount of a certain replication factor does not mean obligatorily that the same factor is poorly loaded on the chromatin. To address this question, we have compared the protein levels between diploid and tetraploid cells in total cell extracts or in chromatin bound extracts. Our data also show a lack of scaling up of replication proteins in the chromatin fraction. Importantly, G1 lengthening allows scaling up the levels of replication proteins with DNA content at the chromatin in tetraploid cells.

To investigate if there are defects at the transcriptional level, we have compared total RNA levels in tetraploid cells induced through MS or EnR. Normalization shows that these are increased in tetraploid cells, but not to the expected level of a cell that has double the amount of DNA.

These new set of data can be found in Fig.3e-h and k-o, and Extended data Fig.8a-c in the revised version of the manuscript.

4. On a similar note, the authors conclude that E2F1 overexpression can override the defect in G1 lengthening to prevent damage in the first S phase. However, their E2F1 overexpression was performed in diploid cells prior to unscheduled tetraploidy, so although it's an interesting experiment, it doesn't fully address their hypothesis. Furthermore, the rescued DNA damage observed in the tetraploid neuroblasts following E2F1 overexpression would be more compelling if the interphase damage was scored in S phase specifically.

We understand this reviewer's concern, but to analyze the first interphase following unscheduled tetraploidization we do not have an alternative way. If we were to induce first tetraploidization and then over-express E2F1, by the time its levels would be increased, cells will be already in the 2nd cell cycle. Importantly, we tested the levels of E2F1 in tetraploid cells and also found that they do not scale up. However, after G1 lengthening, E2F1 levels show comparable levels to diploid cells (no fold change)- Fig.3e-h and Fig.3k-o. These results not only show the importance of G1 extension but also a lack of scaling up for E2F1 in newly born tetraploid cells.

To address the comment related to neuroblasts (NBs), we have repeated these experiments in the presence of EdU to exclusively identify nuclei in S-phase. First, our results clearly show that DNA damage is occurring in nuclei that are in S-phase (EdU +). Moreover, there is less damage in S-phase nuclei in the presence of E2F1. We

have described these findings and added a new Figure- Fig4i-j in the revised version of the manuscript.

5. In figure 2, the suggestion of delayed replication timing in tetraploid cells compared to diploid cells is a key finding. This is in contrast to longer term cultures which have actually identified scaling up in tetraploid lines. Although perhaps challenging, to solidify this finding, it would be more acceptable to directly measure replication timing more accurately as opposed to inferring it from patterns of PCNA.

We realized after consulting with this reviewer, through the editor- Dr Barbara Martea a second time that we did not explain the way replication timing was ascertained. We included DNA replication timing quantification as the signal of PCNA fluorescence intensity (FI) in the nucleus. This replication timing was characterized independently of any particular behavior of PCNA. As soon as PCNA FI was detected in the nucleus, $t=0$ was defined. We have explained this important point better.

6. The accelerated fork progression rate following unscheduled tetraploidy is intriguing and the reasons for this should be explored in more detail. The authors should disentangle the role played by replication stress and identify which phenotypes can be rescued with nucleoside treatment. The authors should also perform fork symmetry assays to identify whether mechanistically, the accelerated fork progression rates in tetraploid cells induced by mitotic slippage is caused by an actual defect in fork stalling similar to what has been reported for PARP inhibition.

We have attempt to rescue the defects in DNA replication with nucleoside treatment as suggested by the reviewer in tetraploid cells generated either by mitotic slippage or endoreplication both in RPE-1 cells and in Drosophila NBs (after CF). Importantly, the levels of DNA damage were not decreased in any of the different conditions- Extended data Fig.5i-j and Extended data Fig.10g.

Concerning accelerated fork progression and fork symmetry analysis, we now show that independently of the way we induce tetraploidy and of the cell line used, tetraploid cells have faster forks. To take into account the reviewer's comments concerning fork symmetry, we have measured the symmetry of the tracks of replication forks that initiated at the same origin. We found that in tetraploid cells there is a highly significant enrichment of asymmetric forks consistent with defects in fork stalling or collapse.

7. The elevated level of copy number variability in tetraploid cells is quite compelling and likely underlies the heterogeneity in G2/M tetraploid cells. Could the authors estimate/calculate the actual proportion of the genome that is subject to these aberrations to get a better appreciation of the severity of the phenotype?

As suggested by the reviewer, we measured the number of aneuploid chromosomes in diploid and tetraploid G1 and G2/M RPE-1 cells and in tetraploid G1 and G2/M BJ cells. We observed a significant increase in the number of aneuploid chromosomes in G2/M tetraploid RPE-1 and BJ cells compared to G1 tetraploid cells. Moreover, the

percentage of cells with ≥ 1 aneuploid chromosome is also increased in G2/M tetraploid cells compared to G1 tetraploid cells- Extended data Fig. 7d. Further, we now included a table summarizing the aneuploidy and the heterogeneity scores (Extended data Fig. 7c), which are extremely informative.

Minor comments

1. Several typos and grammatical errors were noted throughout the manuscript

We apologize for this and we have extensively revised our manuscript and hope that all these have been corrected.

2. Extended data Figure 1K – x-axis labelling is incorrect

Yes, there were two Ts. We have corrected this. Thank you.

Referee #4 (Remarks to the Author):

This manuscript from Gemble et al, addresses a very interesting question regarding the mechanism of genome instability (GIN) following a whole genome duplication event. This is an exciting topic for several reasons, primarily because it is known that genome instability following WGDs is responsible for the aneuploidy that is observed in cancer cells and GIN following WGDs has broad implication for evolution. To tackle this question, the authors artificially induce tetraploidization of RPE cells through multiple mechanisms. And, through a number of assays, determine that DNA damage and GIN in these cells occurs during the first S phase following tetraploidization. This is true across multiple cell types and regardless of the mechanism used to generate the tetraploid cells. They found that tetraploid cells have rapid fork progression, but forks are generally unstable in the first S phase following tetraploidization. They then attempt to determine the underlying molecular mechanism responsible for GIN in these tetraploid cells. Here, the authors found that these artificially induced polyploid cells don't scale up protein production in G1 phase, which results in less replication factors per unit chromatin. Therefore cells have a hard time executing a faithful replication cycle. They can partially rescue the GIN in tetraploid cells by extending G1 prior to the first S phase, which supports their proposed model of GIN.

While I think this is really elegant system to study the consequences of WGDs on GIN, I think there are some concerns about the molecular mechanism underlying GIN in the tetraploid cells. Assuming the data in the paper to be accurate (concerns outlined below), it appears that the mechanism proposed only partially explains the GIN. For example, extending G1 or over expressing E2F1 reduces the DNA damage phenotype by half. While this interesting, clearly there must be something else contributing to the GIN in the first S phase following tetraploidization. In addition to this, the proposed mechanism of GIN in these tetraploid cells remains unclear. These cells don't have enough ORC and MCMs, which suggest a problem with helicase loading. However, there is no change in inter-origin distance when comparing tetraploid and diploid cells. The authors are dancing around this in the discussion In336-337 suggesting that decreased MCM levels could explain the increased rate of fork progress. This is well established that when there are fewer forks competing for limiting resources they can go faster. However, one would expect an increase in the inter-origin distance. The major questions remains, if you can restore protein levels in G1 by extending G1 or E2F1OE, what is driving the remaining (and substantial) GIN in the tetraploid cells?

We understand the point raised by the reviewer. We wrote in the manuscript that G1 lengthening was partially rescuing DNA damage and so G1 duration and lack of scaling in terms of time and protein content seems to be a very likely contributor to GIN. We did not defend a point of view where this was the only contributor. It is also important to mention here that we are extending G1, but maybe the conditions are not the ones that allow for perfect stoichiometry of DNA replication complexes. Indeed, after G1 lengthening some factors are more expressed than a perfect scaling up with DNA content (protein levels which are more than 1, please see Figure 3k and m). Importantly, we now provide evidence that there is a decrease of chromatin-bound

replication factors in newly born tetraploid cells and that this decrease is rescued by G1 lengthening.

In the revised version of our manuscript, we have included an extensive DNA combing analysis of three cell lines, where tetraploidy has been induced by either CF, MS or EnR. In all of these, a higher fork speed has been noticed in tetraploid cells when compared to diploid cells (Figure 2e-g and Extended Data Fig 6j-k). These results suggest that this is a common characteristic of newly born tetraploid cells. Concerning inter origin distance (IOD), in HCT116, a trend for a higher IOD has been observed (Extended Data Fig. 6k), even if not statistically significant. But, we would like to point out that the combing was performed on non-sorted cells - a mix of diploid and tetraploid cells. This might obscure the results somewhat. We have tried several times to perform DNA combing analysis on sorted diploid and tetraploid S-phase cells, however, the sorting seemed to perturb DNA combing experiments. Another important observation is the clear reduction in Pre-RCs in chromatin-bound fractions in tetraploid extracts compare to diploid extract (Figure 3g-h). All together, these data suggest that fewer Pre-RCs result in faster replication rates and most likely in longer IODs.

What might be the most major concern I have with this paper, however, is that the majority of the statistical comparisons were done incorrectly. Therefore, until the correct tests are used, any conclusions drawn on these incorrect tests are not supported by the data. Rather than describing the statistical test in figure legends, the authors devote a section in the methods to say they used Student t-test for all comparisons. Given that most of the analysis focus on more than two groups, they should have used an ANOVA test. While I think that at least some of the claims of significance will stand when the correct tests are performed, I fear that many of the claims of significance will change. I would suggest that the authors think more carefully about the correct statistical analysis for each data set/comparison. It is unlikely one test will be used throughout an entire study. Also, error bars go both above and below the bar graph. Bar graphs are no longer the standard when presenting data like this. At the very least, the data points should be included in the bar graph so the n is clear for each experiment and readers can get a sense of the distribution of the data.

As suggested by this reviewer, we have re-analysed all our previous data and the new set of data with the right statistical tests (including Anova). We have found that all the key results that guided us previously in the conclusions and directions taken were correct. We thank the reviewer for this suggestion.

Also, as suggested by this reviewer, we have included the number of experiments performed in the graph bar. We also included the SEM or SED for the dot plot. Initially we removed these from the graphs to facilitate comprehension.

Significant concerns:

1) Figure 2J: How do the authors define 'unstable forks'? It doesn't appear to be described in the methods or legend. I'm assuming they measuring some aspect of asymmetry, but this should be made clear. Also, I'm suspicious that this result is statistically significant since no statistical test was used to compare the data

sets. Until that is done, they can't claim that forks go faster in the tetraploid cells, which seems to be important for their ideas of GIN following tetraploidization.

We apologize for not having explained this point in the methods. Indeed, in this assay we have measured the symmetry of the tracks of the replication forks that initiated at the same origin. When a fork is unstable, it can stall or even disassemble, which is then reflected by shorter tracks of IdU compare to CldU tracks. We now use the expression Fork asymmetry, which reflects better the basis of the analysis and omits confusion with assays where fork stability is measured directly (Figure 2e-g). For all the different cell lines, independently of the way tetraploidy was induced we observed a significant increase in fork asymmetry in tetraploid cells.

2) For Figure 3D: What would be the most useful comparison is the amount of chromatin-bound ORC, MCM and PCNA, not the amount of total ORC, MCM, etc. per cell. For example, you could have a slight decrease in the total amount of a given protein without affecting the amount loaded on chromatin, which I would argue is more relevant for this study. Additionally, does O/E of E2F1 result an increase in the amount of ORC/MCM on chromatin?

The reviewer is right. This is a really important point. We have performed western blots with chromatin bound fractions in addition to total extracts. We show that indeed the chromatin bound fractions do not scale up in tetraploid cells.

For technical reasons, we did not test the levels of replication factors at the chromatin after E2F1 OE. However, we show that extended G1 duration results in the scaling up of ORC1, MCM2 and CDT1 protein levels at the chromatin.

3) Extended Figure 3H: Given the increase in gammaH2Ax staining throughout the nucleus, is the increase in colocalization with PCNA above what you would expect by chance? The same would be true for Extended 4 figures. However, admittedly the colocalization between Rad51 and gH2Ax is much more convincing.

To respond to this question, we have used Costes randomization using the plugin JACOP in Fiji. We showed that colocalization of γ H2AX with EdU, RAD51 or FANCD2 cannot result from randomness. We have added several examples in Supplementary Data 2a-c.

4) Ln 180-183: While I agree that the signal lingers for a longer period of time, the increase in the number of active sites being more gradual and the dissociation of PCNA happening more progressively would require the authors to compare the slopes of these curves. By eye, I don't see a huge difference that would support these claims. These are also hard claims to validate based on four cells.

We apologize for not explaining the data analyzed more clearly. We have analyzed 25 PCNA expressing cells for each condition (please see Supplementary Data 1). We only included 4 graphs to illustrate more than one cell. But we did not base our conclusions on 4 cells. To follow this reviewer's suggestion, we have determined the slopes of PCNA curves in tetraploid and diploid cells with and without G1 lengthening. These are included in Extended data Fig. 6g and Extended data Fig. 9e.

5) It would be helpful to demonstrate that treating cells with a low level of CDK4/6 and CDK2 inhibitors causes a lengthening of G1. Right now, the authors are assuming that G1 is lengthened. Also, it would be good to know how much G1 lengthens and generally show more cell cycle profiles to confirm arrests, etc. throughout the paper.

To address this request, we have included the FACS profile and time lapse FUCCI data illustrating the effect of the inhibitors- Extended data Fig. 3i and j, Extended data Fig.8d-e and Extended data Fig.9a-c.

6) I don't understand the logic of Rb over expression in Drosophila. While this would theoretically extend G1, it would do so by antagonizing E2F1 and thereby give the opposite phenotype than E2F1OE. In the reference cited, I don't believe there is direct evidence of G1 lengthening in RbOE

The rationale behind RbOE was to extend G1. We understand the reviewer's point but we think that most likely Rb might have other functions in NBs that may overcome E2F1 inhibition. But we think this remains highly speculative at this time. We have thus removed this set of data and only maintained the E2F1OE, which is also consistent with the human cell work.

7) One paradox I see with this work is that dampening of the E2F system in polyploid cells. While trying to compare between naturally polyploid cells and artificially induced tetraploid cells, the assumption is that artificially- induced cells don't have enough time to scale up E2F-dependent expression in G1, but natural polyploid cells do. However, at least in Drosophila, It seems natural instances of polyploidy have adapted to reduced E2F capacity. (Maqbool et al., 2010 J. of Cell Science).

The reviewer bring up an interesting aspect of the differences between naturally occurring polyploid cells and non-physiological polyploidy. As recognized by the reviewer, it is rather intriguing that physiological polyploid cells manage to scale up many parameters while reducing E2F. A likely explanation is that E2F levels have to be maintained very low to ensure that DNA replication is not accelerated in natural polyploid cells. As shown recently by the de Bruin lab ¹⁴, E2F levels determine DNA replication capacity and replication rates impacting even S-phase duration. So a likely possibility is that E2F levels have to be lower during endocycles to ensure the replication of high copy DNA in a correct manner.

Minor concerns:

Ln 88: gammaH2Ax is a well-known marker of DSBs and stalled forks (Ward and Chen JBC 2001). Therefore, gammaH2Ax cannot be assumed to be a marker of only DSBs and doesn't provide evidence that the tetraploid cells have increased levels of DSBs. In fact, based on the data presented in this work (Extended Figure 1M for example), it seems likely that there could be a significant number of stalled forks associated with tetraploid cells.

We have included other DNA damage markers and comet assays- Fig.1i-l and Extended data 2j and k and Extended data Fig. 9l and m.

Ln 179 decreased -> decrease

We have corrected this mistake.

Extended figure 3B legend says 53BP1 is in green, however in the figure it is red.

We have corrected this mistake.

Extended 5(A) legend: As explained below we analysed many more cells than the ones shown in the previous version. We have now included profiles of many of these cells in Supplementary data 1.

Ln 239 - enables us to sort

We have corrected this mistake.

Ln 262 - dependent of cell mass -> dependent on cell mass

We have corrected this mistake.

Ln 313: The don't have 2000 chromosomes. They have a ploidy or C value of 2000, but since they are polytene are only 1n. Diploids are paired in somatic cells to they have 4 chromosomes (2n) and are 2C.

Yes the reviewer is absolutely right. We have corrected this sentence.

Ln 320: Is this striking given that it is the same result you showed in Figure 3?

We have removed the word striking.

Ln 454 - cell -> cells

We have corrected this mistake.

Ln 586 - define FI (assume it's fluorescent intensity)

Yes, we are sorry for not having explained this.

Ln655 - 1/2 500 ->1/2,500

We have corrected this mistake.

Ln 697 - Labelling -> labeling

We have corrected the British way of spelling to the American one.

Ln 724 - define NEBD

We have defined NEBD.

Extended data figure 2D and E: It is not clear what method was used to generate the tetraploid cells. This should be made clear in the text and/or figure legend.

We have included this information in the methods section and in the figure legends.

The pattern of dots is the same for the diploid controls in Extended 3E and F. While I think it is okay to use the same reference control the authors should make this clear in the legend.

The reviewer is absolutely right. We have included this information in the figure legends.

In Ext. Fig 4C and 4F: Why is there such an increase in Rad51 and FANCD2 when comparing the diploid cells +/- realize into S phase? Is the arrest causing damage itself that must be resolved in S phase?

The reviewer is right, we observed more RAD51 and FANCD2 foci in S-phase cells. However, we think that it is due to the fact that both RAD51 and FANCD2 are involved in DNA repair pathways that take place during S-phase and not in G1. So we think that the increase in RAD51/FANCD2 foci spotted by the reviewer actually reflected the fact that RAD51/FANCD2 are mainly recruited in S-phase.

In line with this, we observed fewer KU80 foci in S-phase compared to G1 cells because KU80 / classical NHEJ mainly take place in G1 and not in S-phase.

Figure 4D: Why don't you see DNA damage in the salivary gland chromosomes? These cells are known to have constitutive DNA damage due to under replication of pericentric heterochromatin and other sites throughout euchromatin. Not seeing this damage raises concerns about the sensitivity of the antibody staining.

This reviewer is absolutely right. It has only to do with the sensitivity of the macro- in other words the threshold used to detect DNA damage. Even if there are a few γ -H2Av dots in SGs, they are fewer, much smaller and presenting low FI, when compared to the massive signals we see in the NBs. We have explained this point in the methods section.

References:

1. Dewhurst, S. M. *et al.* Tolerance of whole-genome doubling propagates chromosomal instability and accelerates cancer genome evolution. *Cancer Discov.* **4**, 175–185 (2014).
2. Kuffer, C., Kuznetsova, A. Y. & Storchová, Z. Abnormal mitosis triggers p53-dependent cell cycle arrest in human tetraploid cells. *Chromosoma* **122**, 305–18 (2013).
3. Pedersen, R. S. *et al.* ARTICLE Profiling DNA damage response following mitotic perturbations. *Nat. Commun.* **7**, (2016).
4. López, S. *et al.* Interplay between whole-genome doubling and the accumulation of deleterious alterations in cancer evolution. *Nat. Genet.* **52**, 283–293 (2020).
5. Quinton, R. J. *et al.* Whole-genome doubling confers unique genetic vulnerabilities on tumour cells. *Nature* **590**, 492–497 (2021).
6. Viganó, C. *et al.* Quantitative proteomic and phosphoproteomic comparison of human colon cancer DLD-1 cells differing in ploidy and chromosome stability. *Mol. Biol. Cell* **29**, 1031–1047 (2018).
7. Wangsa, D. *et al.* Near-tetraploid cancer cells show chromosome instability triggered by replication stress and exhibit enhanced invasiveness. *FASEB J.* **32**, 3502–3517 (2018).
8. Ganem, N. J. *et al.* Cytokinesis failure triggers hippo tumor suppressor pathway activation. *Cell* **158**, 833–848 (2014).
9. Baudoin, N. C. *et al.* Asymmetric clustering of centrosomes defines the early evolution of tetraploid cells. *Elife* **9**, (2020).
10. Goupil, A. *et al.* Chromosomes function as a barrier to mitotic spindle bipolarity in polyploid cells. *J. Cell Biol.* **219**, (2020).
11. Nano, M. *et al.* Cell-Cycle Asynchrony Generates DNA Damage at Mitotic Entry in Polyploid Cells. *Curr. Biol.* **29**, (2019).
12. Yahya, G. *et al.* Scaling of cellular proteome with ploidy. *bioRxiv* 2021.05.06.442919 (2021) doi:10.1101/2021.05.06.442919.
13. Potapova, T. A., Seidel, C. W., Box, A. C., Rancati, G. & Li, R. Transcriptome analysis of tetraploid cells identifies cyclin D2 as a facilitator of adaptation to genome doubling in the presence of p53. *Mol. Biol. Cell* **27**, 3065–3084 (2016).
14. Pennycook, B. R. *et al.* E2F-dependent transcription determines replication capacity and S phase length. *Nat. Commun.* 2020 111 **11**, 1–10 (2020).

Reviewer Reports on the Second Revision:

Referees' comments:

Referee #1 (Remarks to the Author):

The authors have adequately addressed most of my concerns.

One remaining point that should be better addressed (and that can be addressed rather easily in my opinion): the analysis of the TCGA ovarian cancer data (ED Fig. 10h) is very limited and not so informative. First, given the high degree of DNA damage and genomic instability in HGSOc, I think that this is not an ideal choice of tumor type to look at. There are multiple additional tumor types in TCGA with a fair amount of both $\sim 2n$ and $\sim 4n$ tumors, so doing the analysis across a few (3-4) of those would be better. Second, it is hard to interpret the upregulation of a subset of the DDR genes, as shown in Fig. 10h. Gene set enrichment analyses should be performed in order to determine if DNA repair pathways are indeed upregulated in the $\sim 4n$ tumors. Third, it would be interesting to try to compare mutational signatures associated with replication stress or DDR-deficiency between $\sim 2n$ and $\sim 4n$ tumors, looking for genetic 'scars' that are unique to the tetraploid tumors (and that might be related to the findings reported in this study).

Referee #2 (Remarks to the Author):

Mechanisms of genetic instability in a single S-phase following whole genome doubling –

The revised version of this manuscript by Renata Basto and colleagues is an impressive improvement of the first version of the paper. It addressed my points of criticism and further expanded on the initial findings.

Intriguingly, it now also shows that origin licensing, the first step of DNA replication, is largely deficient in the first G1 phase after tetraploidization. This gives the authors two potential hypotheses why DNA replication during the first S phase may be deficient. The first is a lack of available, licensed origins. The second is a lack of S phase replication factors required for initiation of those origins. Unfortunately, the manuscript in its current form does not sufficiently discuss which of these two mechanisms (or both) is primarily responsible for the observed replication stress. This could be easily addressed by repeating the analysis of Fig. 3g to the scenarios of extended G1 or E2F1 overexpression. E2F1 overexpression in particular may be useful in this regard as it will likely not affect pre-RC formation, yet it does rescue the replication stress/DNA damage phenotype. I think clarification of this final point will be very useful for the broad readership of Nature.

Lastly, I would also suggest to strengthen the manuscript text by including quantitative statements wherever possible and replacing statements such as "... a certain proportion of γ H2AX foci in S-phase tetraploid cells partially co-localized with..." (l. 156-157

Referee #3 (Remarks to the Author):

The revised version of this manuscript has been suitably revised. The authors have adequately addressed the comments in full. Once some minor details are clarified, this manuscript will be suitable for publication in Nature.

The comments in question are outlined below.

Point 4

The reviewer accepts the explanation that using transient transfection of E2F1 after the induction of tetraploidy would mean that the levels would not be increased until the cells are in the second cell cycle. Did the authors also consider using an type of inducible model system to overexpress E2F1.

Point 6

The observation that nucleoside supplementation did not rescue the DNA damage in tetraploid cells is somewhat surprising. Can the authors offer an interpretation of this finding. Is the suggestion that DNA replication stress is not involved?

Referee #4 (Remarks to the Author):

The authors did a very nice job addressing my concerns and the concerns of other reviewers. I think there are details of the underlying mechanism that still need to be addressed, but that will likely take years and I'm sure be the subject of much investigation by this group. I only have minor comments that should be corrected prior to publication. I congratulate the authors on a really impressive body of work.

Ln 43: is - should be are

Ln 84: Using an early marker of DNA damage (rather than DSBs) - gamma H2Ax marks DSBs and fork stalls among other types of damage.

Ln121-123 and Lns132-134 are nearly identical. I would suggest just saying once.

Ln223-226: It would be helpful to call out extended fig 6k. Took me a minute to find this data.

Figure legends: Anova -> ANOVA

Author Rebuttals to Second Revision:

Referees' comments:

Referee #1 (Remarks to the Author):

The authors have adequately addressed most of my concerns.

One remaining point that should be better addressed (and that can be addressed rather easily in my opinion): the analysis of the TCGA ovarian cancer data (Extended data Fig. 10h) is very limited and not so informative.

First, given the high degree of DNA damage and genomic instability in HGSOV, I think that this is not an ideal choice of tumor type to look at. There are multiple additional tumor types in TCGA with a fair amount of both $\sim 2n$ and $\sim 4n$ tumors, so doing the analysis across a few (3-4) of those would be better.

Second, it is hard to interpret the upregulation of a subset of the DDR genes, as shown in Fig. 10h. Gene set enrichment analyses should be performed in order to determine if DNA repair pathways are indeed upregulated in the $\sim 4n$ tumors.

To follow this new suggestion, we have analyzed the data available from ¹ in which lung, bladder and ovarian cancers have been considered. We chose this reference since $2n$ and $4n$ tumors have been annotated, which is not the case for several other cohorts and would then imply a more in depth and timely cost analysis. Importantly, as suggested and predicted by this reviewer, gene set enrichment analysis revealed a clear enrichment for DNA repair signatures in $4n$ tumors when compared to $2n$ tumors. This new set of data has been included in new Extended data Fig 10h.

Third, it would be interesting to try to compare mutational signatures associated with replication stress or DDR-deficiency between $\sim 2n$ and $\sim 4n$ tumors, looking for genetic 'scars' that are unique to the tetraploid tumors (and that might be related to the findings reported in this study).

The reviewer is absolutely right. This is indeed an extremely important point. We hope the reviewer understands that the characterization and validation of mutational signatures in $4n$ tumors would represent a study on its own and so it does not fit within the scope of this revision. It will indeed require a highly in depth and sophisticated bioinformatic analysis. It is also important to mention that several labs are currently working on this type of question, which has even been addressed by certain studies. These have explored the mutational landscape in diploid and tetraploid tumors. Interestingly, in medulloblastoma, a positive correlation between genome-wide mutation rate and patient age was lost in tetraploid tumors compared to diploid tumors ². In ovarian tumors, it has been reported that genome doubling influences the pathways of tumor progression, with recessive inactivation being less common after genome doubling ³. Finally, in a more recent study which has compared multiple cancers, it has been shown that tetraploid tumors tend to have a higher mutational burden. In particular, a significant enrichment for mutations in spindle assembly checkpoint genes and Kif18 in tumors with WGD ⁴.

We would like to point out that mutational landscapes from evolved tetraploids most likely are also influenced by cell of origin, tissue identity, etc... Therefore, this type of analysis

even if highly important and relevant might generate very broad views on which type of mutations favor tetraploid adaptations. We propose to mention this analysis in the discussion in supplementary information.

Referee #2 (Remarks to the Author):

Mechanisms of genetic instability in a single S-phase following whole genome doubling –

The revised version of this manuscript by Renata Basto and colleagues is an impressive improvement of the first version of the paper. It addressed my points of criticism and further expanded on the initial findings.

Intriguingly, it now also shows that origin licensing, the first step of DNA replication, is largely deficient in the first G1 phase after tetraploidization. This gives the authors two potential hypotheses why DNA replication during the first S phase may be deficient. The first is a lack of available, licensed origins. The second is a lack of S phase replication factors required for initiation of those origins. Unfortunately, the manuscript in its current form does not sufficiently discuss which of these two mechanisms (or both) is primarily responsible for the observed replication stress. This could be easily addressed by repeating the analysis of Fig. 3g to the scenarios of extended G1 or E2F1 overexpression. E2F1 overexpression in particular may be useful in this regard as it will likely not affect pre-RC formation, yet it does rescue the replication stress/DNA damage phenotype. I think clarification of this final point will be very useful for the broad readership of Nature.

We thank this reviewer for acknowledging the improvement of our manuscript and the effort in addressing all the initial comments. We fully agree with the interpretation of this reviewer that both pre-RC loading and DNA replication seem to be impaired in newly born tetraploids. This is precisely why we tested the effect of G1 lengthening in the levels of chromatin bound factors (Figure 3K and n). We apologize if in the revised version we did not explain these findings correctly. Indeed, we think that both events are impaired in tetraploid cells. And since both conditions, G1 lengthening and E2F rescue the high levels of DNA damage we favor the possibility that all the defects may be explained by a lack of scaling up factors that are normally loaded in G1 (pre-RCs) and in S phase (initiation and elongation). We have explained this point in the text and in the supplementary discussion.

Lastly, I would also suggest to strengthen the manuscript text by including quantitative statements wherever possible and replacing statements such as “... a certain proportion of γ H2AX foci in S-phase tetraploid cells partially co-localized with...” (l. 156-157)

We fully understand this point. We have included quantitative information in the new version of the manuscript.

Referee #3 (Remarks to the Author):

The revised version of this manuscript has been suitably revised. The authors have adequately addressed the comments in full. Once some minor details are clarified, this

manuscript will be suitable for publication in Nature. The comments in question are outlined below.

Point 4

The reviewer accepts the explanation that using transient transfection of E2F1 after the induction of tetraploidy would mean that the levels would not be increased until the cells are in the second cell cycle. Did the authors also consider using an type of inducible model system to overexpress E2F1.

We thank the reviewer for his/her nice words and acknowledging our efforts. Concerning the inducible system, we have tried to use such a system to overexpress E2F1, but unfortunately we were not able to obtain stable clones.

Point 6

The observation that nucleoside supplementation did not rescue the DNA damage in tetraploid cells is somewhat surprising. Can the authors offer an interpretation of this finding. Is the suggestion that DNA replication stress is not involved?

We understand the reviewer's concern. The fact that nucleoside supplementation did not rescue DNA damage in tetraploid cells was also a surprise for us. We can explained these findings into two ways: (1) nucleoside concentration in tetraploid cells is not a limiting factor for DNA replication and thus does not explain DNA damage generation in this context or (2) nucleoside supplementation by itself is not sufficient to restore genetic stability because other factors are still limiting and impaired DNA replication generating DNA damage. We have included this information in the new version of the manuscript (Supplementary discussion.) It is also important to mention that nucleoside addition rescue of DNA damage initially reported in ⁵ was described in conditions of replicative stress induced by HPV-16 E6/E7 or cyclin E oncogenes. In this condition, replication forks actually progressed slowly. Which is different from tetraploid cells, where these actually progress faster. As suggested by the reviewer we have discussed the two possibilities in supplemental discussion.

Referee #4 (Remarks to the Author):

The authors did a very nice job addressing my concerns and the concerns of other reviewers. I think there are details of the underlying mechanism that still need to be addressed, but that will likely take years and I'm sure be the subject of much investigation by this group. I only have minor comments that should be corrected prior to publication. I congratulate the authors on a really impressive body of work.

We thank this reviewer for his/her nice appraisal of our efforts and work.

Ln 43: is - should be are

The reviewer is absolutely right. We have modified the text.

Ln 84: Using an early marker of DNA damage (rather than DSBs) - gamma H2Ax marks DSBs and fork stalls among other types of damage.

We agree with the reviewer and we have modified this sentence.

Ln121-123 and Lns132-134 are nearly identical. I would suggest just saying once.

We have modified the text.

Ln223-226: It would be helpful to call out extended fig 6k. Took me a minute to find this data.

We have mentioned this figure.

Figure legends: Anova -> ANOVA

We apologize for this mistake, we have changed all Anova to ANOVA.

REFERENCES:

1. Taylor, A. M. *et al.* Genomic and Functional Approaches to Understanding Cancer Aneuploidy. *Cancer Cell* **33**, 676-689.e3 (2018).
2. Jones, D. T. W. *et al.* Dissecting the genomic complexity underlying medulloblastoma. *Nature* **488**, 100–105 (2012).
3. Carter, S. L. *et al.* Absolute quantification of somatic DNA alterations in human cancer. *Nat. Biotechnol.* **30**, 413–421 (2012).
4. Quinton, R. J. *et al.* Whole-genome doubling confers unique genetic vulnerabilities on tumour cells. *Nature* **590**, 492–497 (2021).
5. Bester, A. C. *et al.* Nucleotide Deficiency Promotes Genomic Instability in Early Stages of Cancer Development. *Cell* **145**, (2011).

Reviewer Reports on the Third Revision:

Referees' comments:

Referee #1 (Remarks to the Author):

I have no further comments. I find the paper suitable for publication in Nature, and I congratulate the authors for the completion of this elegant study.

Referee #2 (Remarks to the Author):

The authors have fully addressed my concerns and the supplementary discussion regarding the cause of replication stress is appreciated.